# IRX3 controls a SUMOylation-dependent differentiation switch in adipocyte precursor cells

Jan-Inge Bjune [1,2,3], Samantha Laber[4], Laurence Lawrence-Archer[1,3], Patrizia M. C. Nothnagel[5], Shuntaro Yamada [6], Xu Zhao[5], Pouda Panahandeh Strømland[3], Niyaz Al-Sharabi[6], Kamal Mustafa [6], Pål R. Njølstad [1,7], Melina Claussnitzer [1,8,9,10], Roger D. Cox[4], Pierre Chymkowitch [5] ✉, Gunnar Mellgren [1,2,3,11] ✉ & Simon N. Dankel [1,2,3,11] ✉

IRX3 is linked to predisposition to obesity through the *FTO* locus and is upregulated during early adipogenesis in risk-allele carriers, shifting adipocyte fate toward fat storage. However, how this elevated *IRX3* expression influences later developmental stages remains unclear. Here we show that IRX3 regulates adipocyte fate by modulating epigenetic reprogramming. ChIP-sequencing in preadipocytes identifies over 300 IRX3 binding sites, predominantly at promoters of genes involved in SUMOylation and chromatin remodeling. IRX3 knockout alters expression of SUMO pathway genes, increases global SUMOylation, and inhibits PPARγ activity and adipogenesis. Pharmacological SUMOylation inhibition rescues these effects. IRX3 KO also reduces SUMO occupancy at Wnt-related genes, enhancing Wnt signaling and promoting osteogenic fate in 3D cultures. This fate switch is partially reversible by SUMOylation inhibition. We identify IRX3 as a key transcriptional regulator of epigenetic programs, acting upstream of SUMOylation to maintain mesenchymal identity and support adipogenesis while suppressing osteogenesis in mouse embryonic fibroblasts.

Common obesity has a strong genetic component that accounts for 40-80% of observed variations in body-mass index (BMI)[1,2]. Among more than 1,100 independent loci known today to be associated with BMI[3], the *FTO* locus was the first to be discovered[4–6] and has since consistently shown the strongest association with BMI across ages and ethnicities[7]. Individuals homozygous for the risk haplotype have 1.7-fold increased odds of developing obesity and gain on average 3 kg more fat than homozygous protective carriers[4]. For a single, common

[1]Mohn Research Center for Diabetes Precision Medicine, Department of Clinical Science, University of Bergen, Bergen, Norway. [2]Mohn Nutrition Research Laboratory, Department of Clinical Science, University of Bergen, Bergen, Norway. [3]Hormone Laboratory, Department of Medical Biochemistry and Pharmacology, Haukeland University Hospital, Bergen, Norway. [4]Medical Research Council Harwell Institute, Mammalian Genetics Unit, Harwell Campus, Oxfordshire, UK. [5]Department of Biosciences, Faculty of Mathematics and Natural Sciences, University of Oslo, Blindern, Oslo, Norway. [6]Center of Translational Oral Research-Tissue engineering, Department of Clinical Dentistry, University of Bergen, Bergen, Norway. [7]Department of Pediatrics and Adolescents, Haukeland University Hospital, Bergen, Norway. [8]The Novo Nordisk Foundation Center for Genomic Mechanisms of Disease, Broad Institute of MIT and Harvard, Cambridge, MA, USA. [9]Diabetes Unit and Center for Genomic Medicine, Massachusetts General Hospital, Boston, MA, USA. [10]Department of Medicine, Harvard Medical School, Boston, MA, USA. [11]These authors contributed equally: Gunnar Mellgren, Simon N. Dankel. ✉ e-mail: pierre.chymkowitch@ibv.uio.no; gunnar.mellgren@uib.no; simon.dankel@uib.no

obesity-associated variant locus, this comprises a huge effect size[3]. However, despite the impact of the *FTO* locus, the underlying mechanisms have long been debated as the associated variants are located in the introns of the gene, thereby lacking a clear function[3,8,9]. While early analyses suggested *FTO* itself could be the target gene (as reviewed in ref. 8), subsequent analyses by us and others showed that the haploblock is situated in a super-enhancer that regulates expression of the neighboring gene IRX3 through a 500 kb long chromatin loop[10–13].

Mechanistically, the variant-harboring enhancer was found to act on IRX3 specifically in adipose-derived mesenchymal stem cells, and only during a short time window during early stages of adipogenesis, in which the risk variant promotes a transient increase in IRX3 expression[12]. Intriguingly, this brief elevation in IRX3 levels in adipocyte precursor cells was sufficient to cause reduced beiging and diminished thermogenesis in the mature adipocytes[12]. Moreover, we recently found that complete knockout (KO) of IRX3 in mouse embryonic fibroblasts resulted in loss of preadipocyte identity, inhibition of adipogenesis, and activation of chondrogenesis[14]. These data suggest that IRX3 may have an epigenetic effect on adipocyte identity, adipogenesis, and/or adipocyte function, although this has never been investigated. The IRX3 protein does not possess known chromatin remodeling domains. However, IRX3 is a homeobox transcription factor involved in a range of developmental processes[15,16], so we reasoned that it may act upstream of chromatin remodeling or histone modifying enzymes to control their expression. However, very few direct IRX3 target genes have been described, and to our knowledge, none related to epigenetic regulation.

The epigenetic landscape and resulting gene expression profile of any given cell largely determines the identity of that particular cell[17]. Well-studied examples include the generation of induced pluripotent stem cells (iPSCs) from adult fibroblasts[18] and vice versa; the commitment and differentiation of pluripotent stem cells into specialized somatic cell types[19,20]. More recently, obesity was linked to an obesogenic epigenetic fate imprinted in adipocytes[21]. Changes in the epigenetic status, and thereby cell identity, is tightly controlled by an array of histone modifying enzymes[22,23] and chromatin remodelers[19], whose activity and function in turn can be modulated by post-translational modifications (PTMs)[24].

Posttranslational modification by the Small ubiquitin-like modifier (SUMO) is particularly involved in modulation of chromatin remodeling, epigenetic regulation, and transcriptional activity[25–27]. SUMO consists of three main members: SUMO1, SUMO2, and SUMO3. The latter two share 97% sequence identity and are often referred to as SUMO2/3[28]. All SUMO isoforms are expressed as immature prepeptides that require enzymatic maturation followed by a series of enzymatic transfer steps before being conjugated to the target proteins (reviewed in ref. 28). While these enzymes, which collectively are termed the SUMO machinery, have been described, and the critical importance of SUMO in determining cellular identities such as discriminating totipotent versus pluripotent stem cells[29,30] is well established, the upstream regulation of this machinery as well as of SUMO itself is completely unknown.

In this study, we aimed to gain insight into the molecular mechanisms underlying the obesity-associated enhancer in the *FTO* locus by uncovering genes that are under direct transcriptional control of IRX3 in developing preadipocytes. We identified both *Sumo1, Sumo3*, and most of the SUMOylation machinery to be directly controlled by IRX3, and show how SUMOylation mediates an IRX3-dependent switch between adipogenic and osteogenic cell fate in MEFs.

## Results

### IRX3 binds to the promoters of the SUMO machinery and chromatin remodelers

To identify direct target genes of IRX3 in differentiating adipocytes, we performed ChIP-seq experiments in preadipocytes isolated from mouse inguinal white adipose tissue (iWAT) one day before (day −1) and one day after (day 1) initiation of differentiation, as well as from gonadal WAT (gWAT) on day 1 of differentiation (Fig. 1). We observed a trend of increased IRX3 binding to the chromatin following initiation of differentiation in iWAT, and more binding in gWAT than iWAT on day 1 among the raw peaks (Fig. 1a, b). We next applied a stringent filter in which only peaks with q-value below 10e-4 and enrichment above 10-fold were considered, resulting in the identification of 310 high-confidence peaks across both depots and timepoints (Fig. 1c and Supplementary Data 1).

Strikingly, more than 95% of the ChIP-seq peaks were located to proximal promoter regions, with a clear enrichment on transcription start sites (TSSs) (Fig. 1d). Thus, virtually all IRX3 binding events could be directly mapped to a target gene (Supplementary Data 1). Reactome pathway analyses revealed chromatin organization and chromatin modifying enzymes to be the most consistently enriched pathways among the top 10 most significantly enriched pathways across all depots and timepoints (Fig. 1e and Supplementary Data 1). Of note, the promoters of genes encoding both histone acetyl transferases (HATs), deacetylases (HDACs), methyltransferases (HMTs) and demethylases (KDMs), as well as SWI/SNF components, were bound by IRX3 (Fig. 1f), suggesting a broad, master-regulatory role of IRX3 in epigenetic regulation of preadipocytes. The second most consistently enriched pathway was related to activation of SUMO, which was significant at both timepoints in iWAT, but not in gWAT (Fig. 1e, g and Supplementary Data 1). Still, IRX3 ChIP-seq tracks show clear binding of IRX3 to SUMO-related promoters in gWAT as well (Fig. 1h). Thus, genes related to processing of SUMO and chromatin modifications were the most consistent IRX3 target genes in primary preadipocytes from both visceral and subcutaneous white adipose tissues.

Since there is some discrepancy regarding the IRX3 binding motif, we performed Motif discovery analyses under the IRX3 peaks. In vitro studies have found Irx orthologs in *Drosophila* to preferably bind to the minimal dimeric binding site ACAnnTGT[31,32], although the inverted repeat (TGTnnACA) is also recognized[31–34]. In contrast, human IRX3 is proposed to bind to $ACA(n)_{10}TGT$ according to JASPAR[35]. Through MEME motif discovery analyses of the IRX3 peaks, we observed the inverted *Drosophila* motif, although it did not reach a significant *E*-value (Fig. 1i). Instead, the motif analysis revealed a highly significant binding site for a large cluster of ETS transcription factor members (Fig. 1j and Supplementary Data 2–4). Among these, the most significant hits were members of the ternary complex factor (TCF) subfamily which are involved in MAPK/ERK-stimulated activation of immediate early response genes, such as *Fos, Jun*, and *Myc*. GO analysis of all the significantly matching TFs showed enrichment of proteins involved in cell differentiation and regulation of transcription (Fig. 1j and Supplementary Data 2–4). The second most significant motif matched another more diverse group of TFs also involved in transcription, but also specifically with adipogenesis (Fig. 1k and Supplementary Data 2–4). These results suggest that IRX3, at least in the context of differentiating preadipocytes, primarily binds indirectly to the chromatin via interactions with other TFs.

### IRX3 binds to open chromatin

We next sought to investigate the chromatin accessibility in IRX3 binding sites during adipogenesis. To this end, we performed ATAC-sequencing at seven timepoints during differentiation of preadipocytes from gWAT (Supplementary Fig. 1 and Supplementary Data 5). While the total number of open regions was stable until day 7 of differentiation, followed by a 50% reduction in the mature adipocytes, there were dynamic changes in chromatin accessibility in ~20,000 loci across the different days, which included more open chromatin in regions related to fat cell differentiation, energy homeostasis and a wide range of other processes (Supplementary Fig. 1a, d

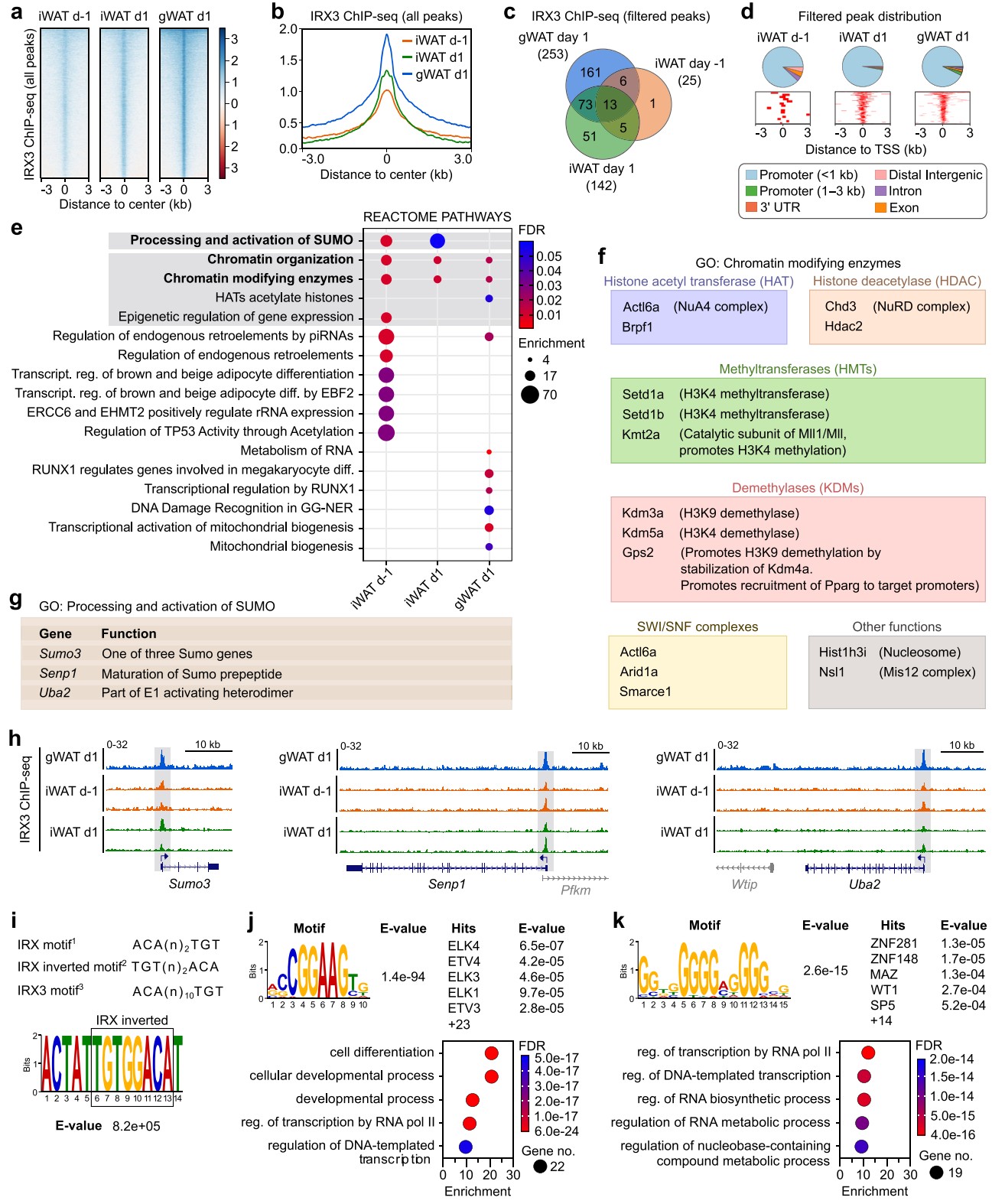

and Supplementary Data 5). Integration of ATAC-seq data with the IRX3 ChIP-seq data revealed that 97% and 80% of IRX3 binding events on day 1 occur in regions with >2-fold and >10-fold ATAC enrichment, respectively. Furthermore, these loci remained open throughout differentiation (Supplementary Fig. 1e, f). While 18% of all ATAC-peaks changed significantly over time, this number was 43% for IRX3-bound ATAC-peaks peak, representing more than a 2-fold enrichment. Moreover, while 58% of changing ATAC-peaks were more open at the

end of differentiation, 94% of IRX3-bound ATAC-peaks showed the same pattern (Supplementary Fig. 1d, g, h). Taken together, IRX3 binds to moderately open chromatin during early stages of white adipose differentiation, and these loci remain accessible until day 7 before fully opening up by the end of the differentiation.

Since IRX3 is important for regulation of white-versus-beige adipocyte identities[12], we further investigated whether the IRX3-bound loci were also accessible in beige adipocytes. We therefore performed

**Fig. 1 | Genome-wide mapping of direct IRX3 target genes in preadipocytes.**
Chromatin immunoprecipitation followed by sequencing (ChIP-seq) was performed for IRX3 in preadipocytes isolated from inguinal white adipose tissue (iWAT) one day before (day −1) and one day after (day 1) induction of differentiation ($n = 2$ per condition), and from gonadal WAT (gWAT) on day 1 ($n = 1$). **a** Heatmap showing normalized IRX3 peak intensities across all called peaks in iWAT and gWAT in preadipocytes. **b** Metaprofile plot of aggregate IRX3 binding across all peaks. **c** Venn diagram of high-confidence IRX3 peaks ($q < 0.001$, ≥ 10-fold enrichment). See also Supplementary Data 1. **d** Genomic distribution of filtered IRX3 peaks relative to gene features (top) and nearest transcription start sites (TSS) (bottom). **e** Top 10 significantly enriched REACTOME pathways among IRX3-bound genes per condition. Full list in Supplementary Data 1. **f** IRX3 target genes involved in chromatin modification. **g** IRX3 target genes involved in SUMOylation. **h** Genome browser tracks of IRX3 occupancy at loci encoding SUMO pathway components. Shaded boxes mark peak locations. **i** Schematic of known and proposed IRX-family DNA binding motifs: preferred minimal motif in *Drosophila* (1)[31,32], inverted (2)[31–34], and predicted in humans (3) (JASPAR, unvalidated[35]). MEME-ChIP motif analysis under filtered peaks reveals a low-significance match to motif 2. **j** Most significantly enriched motif identified by MEME-ChIP and its top five TOMTOM matches (top), with associated Panther GO terms for all significant ($E < 0.05$) matches (bottom). See also Supplementary Data 2–4. **k** Second most significant MEME-ChIP motif, top five TOMTOM matches, and corresponding Panther GOs for all matches ($E < 0.05$). See also Supplementary Data 2–4. Source data are provided in the Source Data file.

ATAC-seq on days 0, 1 and 7 of differentiation in the ME3 beige cell line. While the total number of ATAC-seq peaks was much lower in the beige compared to white cells, most of the IRX3-bound loci identified in white cells were also accessible in the beige cells (Supplementary Fig. 2). Moreover, many of these loci displayed increased openness in the mature adipocytes and were related to stem cell differentiation, histone methylation, transcription and a wide array of other processes (Supplementary Data 5). Overall, chromatin accessibility at IRX3-bound promoters was similar between the white and beige cells.

## Differentially expressed IRX3 target genes relate to histone modifications and chromatin remodeling

Having identified direct IRX3 target genes by ChIP-seq in WAT, and found similar patterns of chromatin accessibility at the promoters of these genes in both WAT and ME3 cells, we next sought to assess whether these genes are affected by IRX3 depletion. We therefore compared the list of IRX3-bound genes in WAT with our previously published list[14] of differentially expressed genes between control and IRX3-KO on days 1 and 7 of adipogenic differentiation in ME3 cells to identify robust IRX3 regulation across similar, but not identical cell types. As expected, we found most of the IRX3-bound genes in WAT to be sensitive to IRX3 ablation in ME3 cells (Fig. 2a and Supplementary Fig. 3). Intriguingly, the direct IRX3-target genes only constituted about 2% of the IRX3-responsive genes (Fig. 2a), suggesting that the direct IRX3 target genes act upstream of vast gene networks. In line with this hypothesis, we previously demonstrated that IRX3-KO results in loss of adipogenic identity, and this effect was prominent already on day 1 of differentiation[14]. Moreover, in the present work, we found that more than half of the direct IRX3 target genes that were differentially expressed following IRX3-KO were affected already on day 1 of differentiation (Fig. 2b). These data support an important and early role of IRX3 in determining adipogenic identity.

The top enriched GO categories for day 1 direct target genes were related to histone methylation and chromatin organization, in addition to several other processes (Fig. 2c, and Supplementary Data 6). While the genes encoding histone methyltransferases were consistently upregulated (Fig. 2d), the genes involved in chromatin organization were both up- and downregulated (Fig. 2e) in IRX3-KO cells on day 1. Interestingly, direct IRX3 target genes that were differentially expressed on day 7 were also enriched for histone modifications and chromatin modification, as well as DNA repair and mitochondrial gene expression (Fig. 2f–j, Supplementary Fig. 3 and Supplementary Data 6). Taken together, these data indicate that IRX3 controls the expression of multiple epigenetic factors during both early and late stages of adipogenic differentiation.

## IRX3 represses SUMOylation

Genes involved in SUMOylation were amongst the most consistent IRX3 target genes in primary preadipocytes (Fig. 1c, e). Given that SUMOylation may affect the function of epigenetic regulators, and has been shown to be critical for epigenetic maintenance of cell identities, we decided to investigate a potential functional link between IRX3 and the SUMOylation pathway. While SUMOylation came up as one of the most enriched GO terms in iWAT, it did not reach significance in gWAT (Fig. 1c). However, IRX3 did bind to the same three SUMO-related genes in gWAT as in iWAT (Fig. 1h), suggesting a regulatory role of IRX3 on SUMO in gWAT as well. Indeed, manual curation of the ChIP-seq tracks revealed that, particularly in gWAT, seven additional genes, representing all parts of the SUMOylation cycle were actually bound to some degree by IRX3 (Fig. 3a). Importantly, most of these genes were also bound by IRX3 in ME3 cells as determined by ChIP-qPCR (Supplementary Fig. 4). Moreover, these genes were differentially expressed in response to IRX3-KO in ME3 cells, and this effect was largely independent of chromatin accessibility (Fig. 3a).

Specifically, SENP1 and SENP5, required for maturation of SUMO, the SAE1 and UBA2 components of the E1 activating enzyme, and RANBP2, an E3 ligase, were all upregulated in the KO cells, both at the mRNA (Fig. 3a) and protein levels (Fig. 3b, c). Conversely, the desumoylase *Uspl1* was downregulated in IRX3-KO cells. These data suggest a repressive role of IRX3 on SUMOylation and thereby increased conjugation of SUMO in the IRX3-KO cells. We further interrogated the IRX3-KO vs control RNA-seq dataset for changes in additional regulators of SUMOylation that were not identified by the ChIP-seq in WAT. Indeed, the SUMO ligases *Pias1*, *Pias2*, *Zmiz1* and *Zmiz2* were all significantly upregulated in the IRX3-KO cells (Fig. 3d). Among these, *Zmiz1* showed both the highest expression levels and the greatest fold increase in response to IRX3 ablation (Fig. 3d). At the same time, the SUMO proteases *Senp2*, *Senp3* and *Senp6* were also significantly upregulated and *Sumo3* itself was downregulated, indicating that a limited number of SUMOylation components responded differentially to IRX3 KO. Overall, however, our data strongly suggested that IRX3 is a global repressor of genes involved in SUMOylation and that in turn, increased conjugation of SUMO may occur in IRX3-KO cells.

Thus, to functionally test SUMOylation levels in response to IRX3 ablation, we performed SUMO Western blots in control and IRX3-KO ME3 cells at different time points during differentiation. These experiments revealed that, at day 0 and day 1, global conjugation of SUMO2/3 significantly increased (1.7-fold) in IRX3-KO cells compared to wild-type cells (Fig. 3e, f). Moreover, there was also increased conjugation of SUMO1 in the IRX3-KO cells, but this effect was most pronounced on day 0 (Fig. 3g, h). Conversely, reintroducing IRX3 in the KO cells reduced the levels of SUMOylation by both SUMO2/3 (Fig. 3i, j) and SUMO1 (Fig. 3k, l). These data confirmed that IRX3 globally suppresses SUMOylation during early differentiation of beige ME3 adipocytes (Fig. 3m).

## Inhibition of SUMOylation restores adipogenesis in IRX3-KO cells

We previously demonstrated that loss of IRX3 in beige cells abolishes adipogenesis[14], and we show here that IRX3 ablation increases SUMOylation in the same cell type. Because SUMOylation has been reported to repress adipogenesis[36,37], we hypothesized that elevated SUMOylation may mediate the repressive effect of IRX3 ablation on adipogenesis. If so, pharmacological inhibition of SUMOylation should

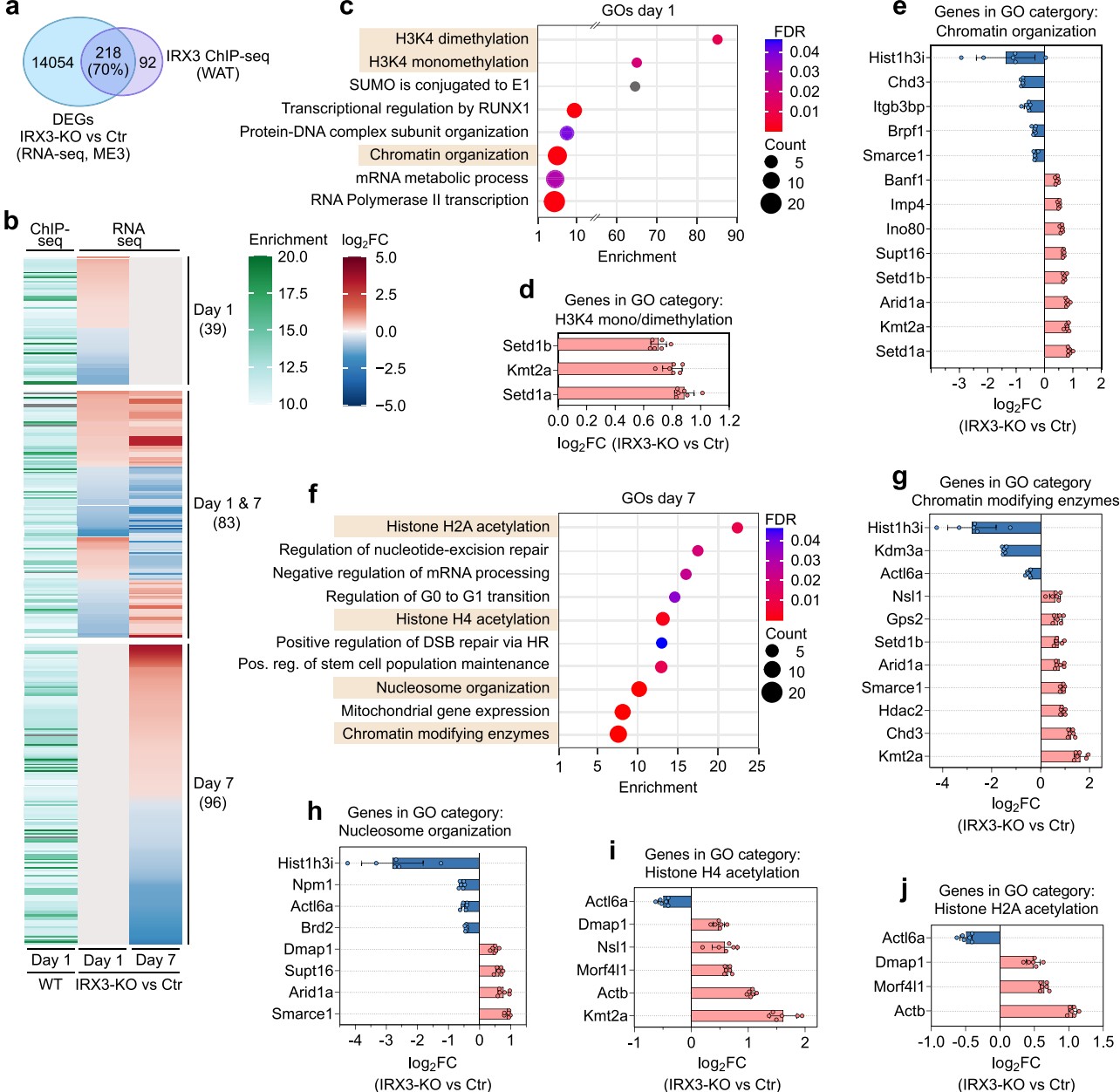

**Fig. 2 | Differentially expressed IRX3 target genes relate to histone modifications and chromatin remodeling.** To identify functionally relevant IRX3 targets, we integrated IRX3 ChIP-seq data from iWAT ($n = 2$) and gWAT ($n = 1$) with transcriptomic profiles of IRX3-knockout (KO) ME3 cells during adipocyte differentiation ($n = 6$; $p_{adj} \leq 0.01$; fold change $\geq 1.2$; data from[14]. **a** Venn diagram showing the overlap between IRX3-bound genes and merged differentially expressed genes (DEGs) on days 1 and 7 of differentiation. Circle sizes are not to scale. **b** Heatmap showing IRX3 binding enrichment at target loci alongside the direction and amplitude of gene expression changes in IRX3 cells on days 1 and 7. **c** Enriched Panther and Reactome ontology terms among the 122 IRX3 target genes that are differentially expressed on day 1. Bar graphs illustrating Log₂ fold

changes in gene expression for IRX3-KO versus control cells within the GO categories "H3K4 mono/demethylation" (**d**) and "Chromosome organization" (**e**) on day 1. **f** Enriched GO terms among the 179 IRX3 target genes with altered expression on day 7. Log₂ fold changes in expression for genes within the GO categories "Chromatin modifying enzymes" (**g**), "Nucleosome organization" (**h**), "Histone H4 acetylation" (**i**), and "Histone H2A acetylation" (**j**) on day 7. Bars indicate mean expression changes ($n = 6$), with standard deviation shown as error bars. Upregulated genes are displayed in red; downregulated genes in blue. See Supplementary Fig. S3 for additional categories. Source data are provided in the Source Data file.

restore adipogenesis in the IRX3-KO cells. To test this, we treated control and IRX3-KO cells with the SUMOylation inhibitor ML-792 from days −2 to 9 of differentiation and assessed the effect on differentiation (Fig. 4). Strikingly, ML-792 treatment had a profound impact on the differentiation capacity of the IRX3-KO cells (Fig. 4a, b and Supplementary Fig. 5a), and was able to completely restore the total lipid levels (Fig. 4c). This effect was attributable to both a partial restoration

of the number of differentiating cells, as well as increased lipid droplet size in each cell (Fig. 4c). In accordance with this, the expression of both the adipogenic master regulators *Pparg* and *Cebpb*, as well as other markers of mature adipocytes and lipid metabolism were increased in response to ML-792 (Fig. 4d). Of note, ML-792 also had a mild proadipogenic role in the control cells, but the effect was more profound in the IRX3-KO cells (Fig. 4e). Despite the stronger effect of

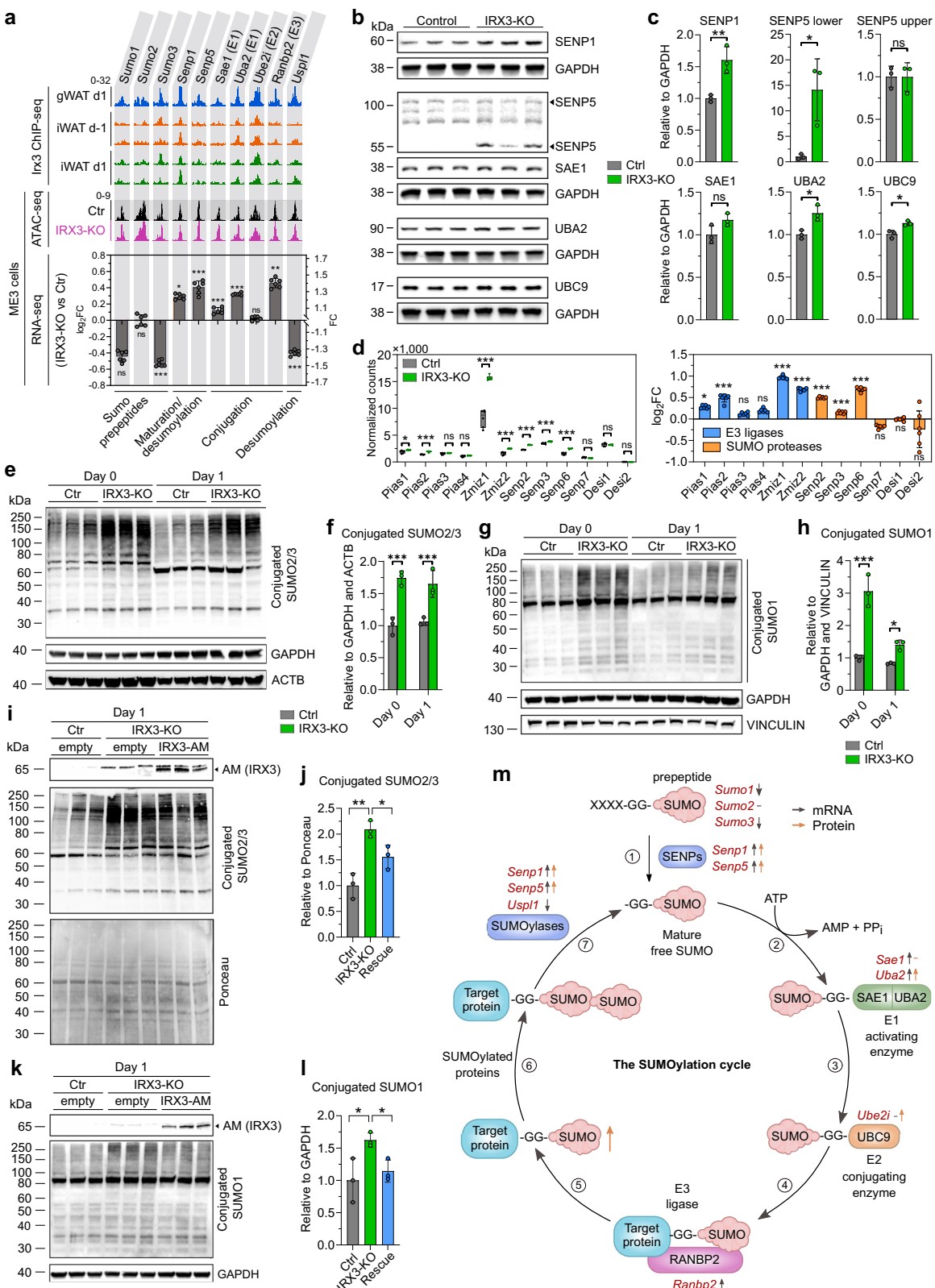

ML-792 in IRX3-KO compared to control cells, the SUMO inhibitor was unable to fully restore the adipogenic gene expression in the KO cells, possibly reflecting our observation that not all KO cells acquired lipid droplets following ML-792 treatment. Intriguingly, *Ucp1* responded to ML-792 in the control cells, but not in the IRX3-KO cells, suggesting that inhibition of SUMOylation restored a white, but not beige adipocyte identity.

ML-792 demonstrated a dose-dependent response effect on lipid accumulation in the IRX3-KO cells (Fig. 4e and Supplementary Fig. 5b), with minimal toxicity at 0.5 μM (Supplementary Fig. 5c).

At both concentrations of ML-792, the pro-adipogenic effect was roughly doubled when administered from day 0 instead of day −2 (Fig. 4f), indicating the importance of inhibiting SUMOylation after the induction of adipogenesis. These results prompted us to test whether

**Fig. 3 | IRX3 represses SUMOylation.** IRX3 ChIP-seq, ATAC-seq, RNA-seq, and immunoblots were used to assess IRX3-dependent regulation of SUMOylation during adipocyte differentiation. **a** IRX3 ChIP-seq signal near TSS ($\pm$1.5 kb) of SUMO pathway genes in iWAT ($n = 2$) and gWAT ($n = 1$) preadipocytes before and after induction of differentiation (top); average ATAC-seq profiles at the same loci in ME3 control and IRX3-KO cells on day 1 ($n = 3$) (middle); log$_2$ fold changes in IRX3-KO vs. control ME3 cells on day 1 ($n = 6$) (bottom). *$p_{adj}$ = 0.01, **$p_{adj}$ = 0.001, ***$p_{adj}$ < 0.001; multiple unpaired, two-sided t-tests with Holm–Sidak correction. **b** Immunoblot of SUMO pathway proteins from **a** in ME3 control and IRX3-KO cells on day 1 ($n = 3$). Representative of two experiments. **c** Quantification of **b**, normalized to GAPDH. *$p < 0.05$, **$p = 0.009$; unpaired, two-sided t-test. **d** RNA-seq of SUMO ligases and proteases not bound by IRX3 in WAT ($n = 6$)[14]; expression

(left) and log$_2$ fold changes (right). *$p_{adj}$ = 0.03, ***$p_{adj}$ < 0.001; multiple unpaired, two-sided t-tests with Holm-Sidak correction. **e–h** Immunoblots and quantifications of global SUMO2/3 (**e–f**) and SUMO1 (**g–h**) conjugation in control and IRX3-KO ME3 cells on days 0 and 1 ($n = 3$). Representative of four (**e-f**) and two (**g-h**) experiments. *$p_{adj}$ = 0.03, ***$p_{adj}$ < 0.001; two-way ANOVA with Holm–Sidak correction. IRX3 rescue in IRX3-KO cells reduces SUMO2/3 and SUMO1 conjugation on day 1. Immunoblots (**i, k**) and quantifications (**j, l**) shown ($n = 3$). Representative of three (**i, j**) and two (**k, l**) experiments. *$p_{adj}$ < 0.05, **$p_{adj}$ = 0.002. **m** Schematic of the SUMOylation cycle. IRX3-bound genes in red; arrows indicate RNA (grey) and protein (orange) changes. Created in BioRender. Bjune, J. (2025) https://BioRender.com/1m0hzx6. Box plots show medians and interquartile ranges; bar graphs show means ± SD. Source data are provided in the Source Data file.

the pro-adipogenic effect of ML-792 was acting during early or late stages of differentiation. Strikingly, delaying the administration of the inhibitor to day 2 or later completely abolished the pro-adipogenic effect, and administration from day 4 actually repressed rather than stimulated differentiation (Fig. 4g). These data indicate that inhibition of SUMOylation plays a particularly important role during days 0-2 of differentiation, i.e., in the window where IRX3 is highly expressed[14] and variants in the *FTO* locus affect IRX3 expression[12]. However, treatment with ML-792 solely on days 0–2 resulted in only a minor stimulatory effect (Fig. 4g), suggesting that inhibition of SUMOylation during the two first days of differentiation is necessary, but not sufficient to promote adipogenesis. Taken together, these data clearly demonstrate that inhibition of SUMOylation during adipogenic stimulation rescues the adipogenic capacity of IRX3-KO cells, but only in the direction of white and not beige adipocytes. This suggests that elevated SUMOylation mediates some, but not all the effects of IRX3 ablation in the ME3 cells (Fig. 4h).

## Inhibition of SUMOylation enhances PPARγ and PGC-1α transcriptional activities

Manipulating IRX3 and SUMOylation has a strong effect on adipogenesis, suggesting that they both control key regulators in adipogenesis, such as PPARγ. Surprisingly, although *Pparg* was strongly downregulated during late stages of adipogenic stimulation in IRX3-KO cells[14], its expression on day 1 was not reduced but rather slightly upregulated (Fig. 5a). However, downstream PPARγ target genes, which themselves are adipogenic activators, such as *Ebf2*, *Ppargc1a* and *Prdm16*, were strongly downregulated already on day 1 (Fig. 5a). This suggests that loss of IRX3 impaired PPARγ activity. Since SUMOylation has been found to have a repressive effect on PPARγ activity[38,39] we hypothesized that the elevated SUMOylation following IRX3-KO could negatively affect PPARγ. To address this, we examined the ability of PPARγ and its co-activator PGC-1α to activate a luciferase reporter under control of an artificial PPAR-response element (PPRE) in ME3 cells with or without ML-792 (Fig. 5b). Indeed, inhibition of SUMOylation increased PPARγ transcriptional activity 4-fold, but only in the presence of the synthetic ligand rosiglitazone (rosi) when PPARγ was overexpressed alone. Interestingly, inhibition of SUMOylation also increased PGC-1α co-activation activity in the presence of rosi. SUMO is known to inhibit the transcriptional co-activity of PGC-1α[40] and is likely recruited to the PPAR-response element by endogenous PPARγ. As expected, co-expression of PGC-1α with PPARγ resulted in strong transcriptional synergy in DMSO-treated cells, and this effect was 3-fold higher in rosi-treated cells. According to a repressive effect of SUMO on PGC-1α, ML-792 treatment resulted in an additional 2.6–2.7-fold increase in transcriptional activity in both DMSO and rosi-treated cells in the presence of overexpressed PGC-1α (Fig. 5b). Thus, blocking SUMOylation enhanced the transactivation activity of PPARγ in a rosi and PGC-1α-dependent manner, with the strongest transcriptional effect achieved when blocking SUMOylation in the presence of both the ligand and the co-activator (Fig. 5b).

## IRX3 ablation alters SUMO occupancy at Wnt and Rho signaling genes

Having found SUMO to play a role in regulating PPARγ-dependent transcriptional activity, we next sought to determine how IRX3 ablation could impact SUMO occupancy at the chromatin using ChIP-seq in control and IRX3-KO ME3 cells on days −1 and 1 of adipogenic differentiation (Fig. 6). PCA of the SUMO peaks ($n = 2$ biological replicates) demonstrated high reproducibility between replicates and considerable differences between the time points and genotypes (Supplementary Fig. 6a). IRX3 ablation resulted in a strong rearrangement of SUMO occupancy both before and after induction of adipogenesis (Fig. 6a and Supplementary Fig. 6b), although the most statistically significant changes in peak intensity between the two genotypes occurred at day 1 (Fig. 6b, c). Adipogenic induction also correlated with significant changes in SUMO occupancy on the chromatin, and this was observed in both control and IRX3-KO cells (Fig. 6b). However, while control cells exhibited a roughly equal number of significantly hyper- and hypoSUMOylated loci in response to the adipogenic induction, the IRX3-KO cells responded almost exclusively by hyperSUMOylation (Fig. 6b, c). Importantly, these loci were enriched with genes related to osteoblast differentiation (Fig. 6d).

While hyperSUMOylated genes in IRX3-KO cells were enriched with genes related to extracellular matrix and synapse organization (Supplementary Fig. 6c), the hypoSUMOylated genes were related to Wnt and Rho signaling (Fig. 6e), which are well-known to inhibit adipogenesis and promote osteogenesis. We next performed motif enrichment analyses to investigate which transcription factors might mediate the shift in SUMO occupancy at Wnt-related genes. STREME analysis identified three enriched motifs with matches in the TOMTOM database, including a motif that matched multiple transcription factors, including several KLF, SP and ZNF family members (Fig. 6f). Of note, many of these, such as SP1, KLF4, KLF5, KLF7 and KLF9, have known roles in regulating osteogenesis and adipogenesis[41–46].

To further analyze which sumoylated transcription factors might be differentially bound to the Wnt-related genes, we interrogated the *Rspo2* gene, a well-known activator of Wnt signaling[47–49], from the list of genes comprising the GO term "Wnt signaling pathway", and searched for overlaps between the SUMO2/3 ChIP-seq peak in our ME3 cells with a library of publicly available ChIP-seq datasets containing multiple transcription factors in various cell types (Fig. 6g). The summit of the SUMO peak in the proximal promoter of *Rspo2* was found right at the transcription start site, and overlapped with peak summits of the key osteogenic and adipogenic master regulators RUNX2, SP1, PPARγ and RXR (Fig. 6g, bottom panel). These factors are all SUMO targets[50,51]. Since Wnt signaling inhibits adipogenesis[52] and promotes osteogenesis[53], we hypothesized that SUMO would repress this pathway and that the reduced chromatin occupancy of SUMO at loci involved in the Wnt pathway would increase Wnt signaling in the IRX3-KO cells. Indeed, *Rspo2* expression was increased in the IRX3-KO cells, and genes known to be activated and repressed by Wnt signaling were up- and downregulated, respectively in the KO cells (Fig. 6h, i), indicating increased Wnt signaling in IRX3-KO compared to control ME3

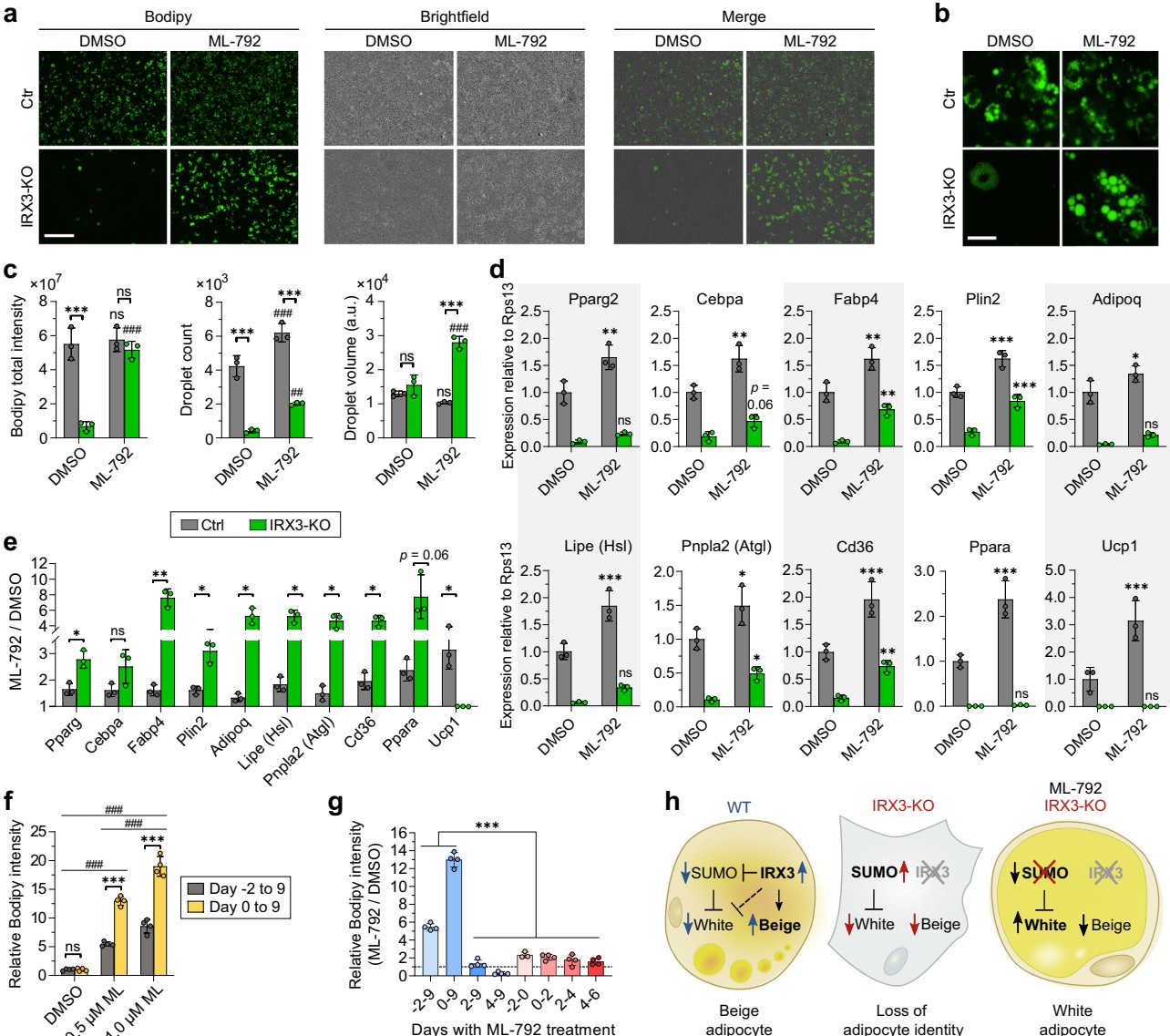

**Fig. 4 | Pharmacological inhibition of SUMOylation restores adipogenesis in IRX3-KO cells.** To assess whether hyperactive SUMOylation contributes to impaired adipogenesis in IRX3-KO cells, ME3 control and IRX3-KO cells were treated with the SUMO E1 inhibitor ML-792 (0.5 μM) from days −2 to 9 of adipocyte differentiation. **a** Fluorescence and brightfield microscopy images of cells stained with Bodipy lipid dye on day 9. Representative fields from biological triplicates (9 fields/well) shown from one of four independent experiments. Scale bar, 400 μm. See Supplementary Fig. S5a for full image sets. **b** Higher magnification of selected fields from **a**. Scale bar, 50 μm. **c** Quantification of Bodipy intensity, lipid droplet number, and average lipid volume per cell. Data from 9 fields per well merged; $n = 3$ wells per group. ***$p_{adj} < 0.001$ (IRX3-KO vs. control); ##$p_{adj} = 0.002$, ###$p_{adj} < 0.001$ (ML-792 vs. DMSO); ns, not significant; two-way ANOVA with Holm−Sidak correction. **d** Adipogenic gene

expression in ME3 control and IRX3-KO cells treated with ML-792; $n = 3$ from one of two experiments. Data normalized to *Rps13* and expressed relative to DMSO-treated control. *$p_{adj} < 0.05$, **$p_{adj} < 0.01$, ***$p_{adj} < 0.001$. **e** Relative effect of ML-792 on gene expression (ML-792/DMSO) in control and IRX3-KO cells. *$p_{adj} < 0.05$, **$p_{adj} = 0.009$; multiple unpaired, two-sided t-tests with Holm−Sidak correction. **f** Dose-response of ML-792 on lipid accumulation in IRX3-KO cells when added from day −2 or day 0 through day 9; $n = 4$ from one of three experiments. *$p_{adj} < 0.05$, ***$p_{adj} < 0.001$ (days), ###$p_{adj} < 0.001$ (dose). **g** Effect of delayed ML-792 applied (blue) or restricted to two-day intervals (red); $n = 4$. ***$p_{adj} < 0.001$; one-way ANOVA with Holm−Sidak correction. **h** Proposed model: IRX3 suppresses SUMOylation to permit adipogenic differentiation. Bar graphs show means ± SD. Source data are provided in the Source Data file.

cells. Taken together, our data demonstrate that IRX3 ablation resulted in reduced SUMO occupancy at genes involved in Wnt signaling, which is associated with increased Wnt signaling and inhibition of adipogenesis. Moreover, our data suggest this may be mediated by altered binding of known regulators of adipogenesis and osteogenesis such as KLF and SP family members, RUNX2, and/or PPARγ.

### IRX3 and SUMO share common target genes involved in adipocyte versus osteoblast development

To better understand which biological processes are under shared control of IRX3 and SUMOylation, we next compared the changes in

global gene expression in response to IRX3-KO and ML-792. We first took advantage of our recently published RNA-seq dataset on the effect of ML-792 on day 1 of differentiation in 3T3-L1 cells[50] and compared it with the effect of IRX3-KO on day 1 of differentiation in ME3 cells (Supplementary Data 11). Of note, ME3 and 3T3-L1 cells exhibit similar gene expression profiles, particularly during early stages of differentiation. We found that more than half of the SUMO responsive genes also showed altered expression with IRX3-KO (Supplementary Fig. 7a), and 40% of these overlapping genes were regulated in opposite directions (Supplementary Fig. 7b), suggesting a repressive effect of IRX3 on the SUMO-dependent regulation of these genes. GO

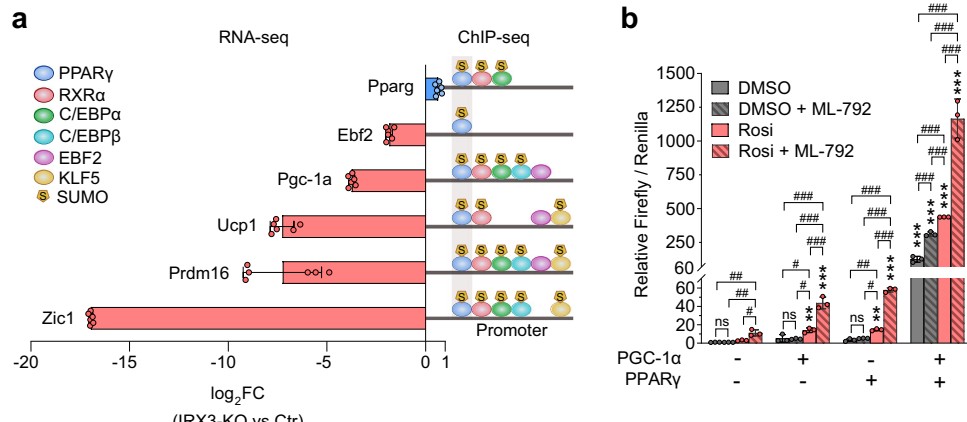

**Fig. 5 | Inhibition of SUMOylation improves PPARγ transcriptional activity and its synergy with PGC-1α.** To investigate how hyperactivated SUMOylation suppresses adipogenesis in IRX3-KO cells, the effect of SUMOylation inhibition on PPARγ activity was assessed. **a** Differential expression of *Pparg* and downstream target genes in IRX3-KO vs. control ME3 cells on day 1 of differentiation ($n = 6$); data from[14]. Experimentally validated binding of upstream transcription factors to the promoters of each adipogenic regulator in WT adipogenic cells is shown to the right; ChIP-seq data collected from the UCSC Genome Browser hub UniBind 2021[143]. Transcription factors experimentally shown to be SUMOylated in WT pre-adipocytes or mature adipocytes are marked with (S), data from[50]. **b** Luciferase

activity of a reporter gene under control of 3 × PPRE sites, co-transfected with PPARγ and/or PGC-1α in ME3 cells. Firefly luciferase units relative to the control group and normalized to constitutive Renilla luciferase is shown; $n = 3$ replicates from one out of two independent experiments. \*\*$p_{adj} = 0.001$, \*\*\*$p_{adj} < 0.001$, overexpression of PGC-1α and/or PPARγ compared to empty plasmid; #$p_{adj} < 0.05$, ##$p_{adj} < 0.01$, ###$p_{adj} < 0.001$, comparison between DMSO, rosi and/or ML-792, two-way ANOVA with Holm-Sidak correction for multiple testing. The data were square-root-transformed prior to statistical analyses. The bar graphs show means ± SD. Source data are provided in the Source Data file.

---

analyses identified genes involved in (brown) fat cell differentiation to be among the top 10 most enriched processes among the overlapping genes and, as expected, these were downregulated in the IRX3-KO cells and upregulated in ML-792 treated cells (Supplementary Fig. 7c). In addition, a diverse set of other processes were also inversely regulated, including catabolism of branched-chain amino acids, actin remodeling, various immune-related processes, extracellular matrix remodeling and histone modification (Supplementary Fig. 7c and Supplementary Data 11).

While we have identified pathways affected by both IRX3 ablation and inhibition of SUMOylation, we next sought to determine whether we could identify the direct SUMO target genes and compare them with IRX3-responsive genes. To this end, we first pooled the significant ML-792 responsive genes identified from the SLAM-seq and RNA-seq datasets, without filtering for fold-change (Supplementary Data 12). We then compared this extended list of genes with our recently published Sumo2/3 ChIP-seq dataset[50] to identify genes that are bound by SUMO and differentially expressed in response to ML-792 (Supplementary Data 12). These direct SUMO target genes were subsequently compared with the IRX3-KO dataset (Supplementary Fig. 7d). Nearly 2/3 of the direct SUMO target genes overlapped with the IRX3-KO responsive genes, and oppositely regulated genes were enriched in GO terms related to differentiation of both adipocytes and osteoblasts, in addition to regulation of histone deacetylation (Supplementary Fig. 7e and Supplementary Data 12). Taken together, our data show that a majority of direct SUMO target genes are regulated by IRX3, strengthening our finding of IRX3 as an upstream regulator of SUMOylation. Moreover, the shared target genes are involved in a range of biological processes, including histone modifications and regulation of adipogenesis and osteogenesis.

We also compared the response in global gene expression to lack of IRX3 or inhibition of SUMOylation on day 7 of differentiation. In the mature adipocytes, 70% of SUMOylation-sensitive genes were also affected by IRX3-KO and 67% of these were regulated in opposite directions (Supplementary Fig. 8). "SUMOylation of transcription factors" was among the top enriched/most significant GO terms for genes repressed upon SUMOylation inhibition, but activated in the absence of IRX3. Therefore, these data corroborate the repressive effect of IRX3 on SUMOylation.

Finally, to directly compare the effect of IRX3 ablation and inhibition of SUMOylation in the same cellular system, we treated ME3 control and IRX3-KO cells with DMSO or ML-792 at three different stages of adipogenic differentiation: just before induction (from day −2 to day −1), during induction (days 0–1) and throughout differentiation (days 0–9) (Fig. 7a and Supplementary Fig. 9a). While IRX3-KO consistently affected a large number of genes, with ~6000 DEGs on both days −1 and 1 and ~9,500 DEGs on day 9, the impact of ML-792 varied significantly depending on the time of treatment and genotype. In line with our previous findings, ML-792 had little impact on day −1 (200 −700 DEGs), but a strong effect on days 1 and 9 (~2500–5500 DEGs) (Supplementary Fig. 9 and Supplementary Data 13). GO analyses showed that IRX3 ablation had a profound effect on the transcriptome already on day −1, reducing the expression of genes involved in fatty acid metabolism, insulin signaling and a wide array of other processes, while increasing the expression of genes involved in chromosome segregation and cell cycle regulation (Supplementary Data 14). On day 1 and 9, genes involved in fat cell differentiation, metabolism, and multiple other processes were downregulated, while cell cycle-related processes were upregulated (Supplementary Data 14), in line with our previous observations[14].

We hypothesized that ML-792 would reverse at least some of the changes in gene expression resulting from IRX3 ablation. Thus, we analyzed the overlap in gene expression between IRX3-KO vs control cells (treated with DMSO) and ML-792 vs DMSO (in IRX3-KO cells) for each of the three timepoints (Fig. 7b). While IRX3-KO affected a large number of genes on day −1, ML-792 only had a modest effect at this timepoint. Still, 75% of the ML-792 responsive genes overlapped the IRX3 sensitive genes. On days 1 and 9, the number of overlapping genes increased 3.3- and 4-fold, respectively. These data are consistent with our previous observation that for ML-792-mediated restoration of adipogenic differentiation in the IRX3-KO cells, days −2 to 0 are dispensable, days 0–2 are essential, and prolonged treatment from day 0 until day 9 are optimal (Fig. 4g).

If IRX3-KO were to alter gene expression at least partially via increased SUMOylation, we reasoned the common target genes should mainly be regulated in opposite directions. While approximately 50% of the genes were inversely regulated on days −1 and 1, this number

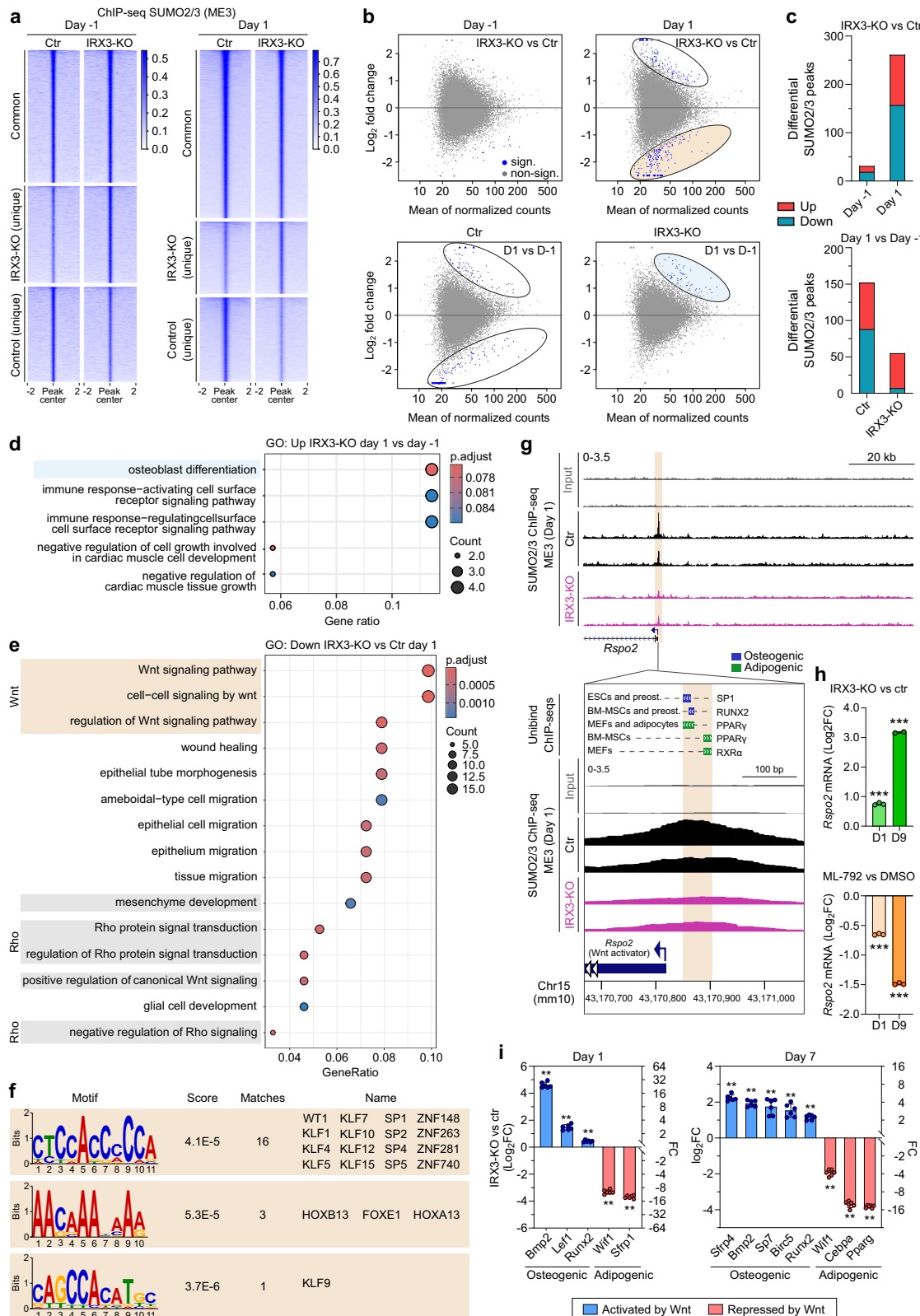

increased to 89% on day 9 (Fig. 7c), supporting a clear functional link between SUMOylation and IRX3 during adipogenesis. The inversely regulated genes were enriched with genes related to a broad range of biological processes (Fig. 7d and Supplementary Data 16). As expected, fat cell differentiation and processes related to fatty acid oxidation/ mitochondrial function were among the top 5 most significant GO terms among genes downregulated with IRX3-KO and upregulated

with ML-792 and on days 1 and 9, respectively (Fig. 7d). Conversely, we observed "response to TGF-beta" among the top 5 most significant processes among genes upregulated with IRX3-KO and downregulated with ML-792 on day 1. Since this signaling pathway is involved in regulating osteogenesis[54] and we previously found other evidence of increased osteogenic gene expression in response to IRX3-KO and a corresponding reduced osteogenic gene expression with ML-792

**Fig. 6 | IRX3 ablation alters SUMO occupancy at Wnt and Rho signaling genes.** ChIP-seq for SUMO2/3 was performed in ME3 control and IRX3-KO cells on days −1 and 1 of adipogenic differentiation ($n = 2$ biological replicates per condition). **a** Heatmap of SUMO2/3 ChIP-seq peak intensities clustered by shared and condition-specific peaks. See Supplementary Fig. 6, Supplementary Data 7. **b** MA plots showing differential SUMO2/3 peak intensity for IRX3-KO vs control (top) and day −1 vs day 1 (bottom). Blue dots, peaks with $p_{adj} < 0.05$; grey, non-significant; DESeq2 analysis with Benjamini-Hochberg correction. Beige ellipse, hypoSUMOy-lated regions in IRX3-KO on day 1; blue ellipse, hyperSUMOylated regions on day 1 vs day −1 in IRX3-KO. See Supplementary Data 7. **c** Number of significantly up- and downregulated SUMO peaks in **b**. **d** ClusterProfiler GO enrichment analysis of genes associated with increased SUMO occupancy in IRX3-KO cells on day 1 versus day −1; Benjamini-Hochberg correction, top 5 terms shown. See Supplementary Data 8. **e** ClusterProfiler GO enrichment analysis of genes with reduced SUMO occupancy in IRX3-KO vs control cells on day 1; Benjamini-Hochberg correction, top 15 terms shown. See also Supplementary Fig. 6 and Supplementary Data 8. **f** STREME motif enrichment analysis of SUMO peaks within Wnt pathway genes. Matching TOM-TOM motifs are shown. See Supplementary Data 9–10. **g** Genome browser view of SUMO2/3 peaks at the *Rspo2* promoter (top). Zoomed-in comparison with Unibind 2021 ChIP-seq tracks[143] from relevant cell types showing peak overlap with transcription factors involved in osteogenesis and adipogenesis (bottom). **h** Relative *Rspo2* mRNA levels in ME3 control vs IRX3-KO cells (top) and in response to ML-792 (bottom) on day 1 and day 9 ($n = 3$). ***$p_{adj} < 0.001$; two-way ANOVA with Holm–Sidak correction. **i** Relative expression of Wnt-responsive genes in IRX3-KO vs control cells on days 1 and 7 ($n = 6$). Genes upregulated and downregulated by Wnt signaling are shown in red and blue, respectively. **$q < 0.01$; multiple two-sided Mann-Whitney tests with FDR correction. Bar graphs show means ± SD. Source data are provided in the Source Data file.

treatment (Supplementary Fig. 7d, e), we interrogated the entire list of enriched GO terms in the gene set that was upregulated with IRX3-KO and downregulated with ML-792 on day 1 (quadrant/gene set iv) for processes related to osteogenesis. We identified multiple significant and enriched osteogenic processes, including osteoblast proliferation and differentiation, bone formation, ossification, and Wnt, Notch, and BMP signaling (Fig. 7e). Finally, we searched for changes in chromatin occupancy on the promoters of the inversely correlated genes. We observed a significant increase in SUMO occupancy at the distal promoter of *Fdx1* (Fig. 7f), a gene involved in lipoylation of enzymes in the TCA cycle[55] and part of the GO term "electron transport chain" (quadrant v). Overall, we found IRX3 and SUMOylation to inversely regulate genes involved in both adipogenesis and osteogenesis.

## IRX3-KO promotes osteogenesis

Our gene expression data suggest that IRX3 ablation and the resulting increase in SUMOylation may promote osteogenesis over adipogenesis. We hypothesized that IRX3-KO cells may have lost their resistance towards the osteogenic lineage due to an underlying alteration of the epigenetic landscape. We therefore performed ATAC-seq to measure differences in chromatin accessibility during adipogenic differentiation in control and IRX3-KO cells. More than 35,000 genomic loci displayed differential openness in response to IRX3 ablation on day 0, and this number was increased to 93,000 and 81,000 on days 1 and 7, respectively (Supplementary Data 17), demonstrating a profound effect of IRX3 on the global chromatin landscape. Indeed, GO analyses of ATAC-seq peaks in more open promoters in IRX3-KO cells on day 0 revealed significant enrichment of several processes related to osteogenesis, including Wnt-signaling, osteoblast differentiation and bone generation (Fig. 8a, b, Supplementary Fig. 10 and Supplementary Data 18), supporting an epigenetic and developmental reprogramming in the IRX3-KO cells. Moreover, similar GO terms were found for more open chromatin on days 1 and 7 (Supplementary Fig. 10 and Supplementary Data 18), suggesting that the pro-osteogenic changes observed on day 0 stably remained and resisted adipogenic stimulation. Finally, we observed strong correlation between the RNA-seq and ATAC-seq data (Supplementary Fig. 11 and Supplementary Data 19). Overall, these data suggest that IRX3 ablation induced a remodeling of the chromatin that steers differentiation towards osteogenesis rather than adipogenesis.

To experimentally test whether IRX3-KO allows ME3 cells to functionally undergo osteogenesis rather than adipogenesis, we performed 3D cultures of ME3 cells and subjected them to either adipogenic (Fig. 8c, d) or osteogenic (Fig. 8e, f) stimulation. As expected, the ME3 control cells readily differentiated into mature adipocytes featuring large lipid droplets after adipogenic stimulation (Fig. 8c) while the IRX3-KO cells did not (Fig. 8d). Conversely, while the control cells showed no or minimal response to osteogenic stimulation, the IRX3-KO cells demonstrated significantly higher basal or induced levels of osteogenic alkaline Phosphatase staining (Fig. 8e) and matrix

mineralization (Fig. 8f). Gene expression data revealed a significant increase in the expression of the key osteogenic transcription factor *Sp7*, on day 7 of osteogenic stimulation in IRX3-KO compared to control cells (Fig. 8g). Furthermore, the intermediate osteogenic marker *Spp1* was elevated in the IRX3-KO cells on day 21 in both growth medium and the osteogenic medium, while the intermediate osteogenic markers *Alpl* and *Col1a1* were strongly upregulated in the IRX3-KO cells at both timepoints and in both media (Fig. 8g). Taken together, the IRX3-KO cells display altered chromatin and transcriptional landscapes that functionally promote differentiation towards an osteogenic lineage.

To assess whether increased SUMOylation levels could mediate the pro-osteogenic effect of IRX3 ablation, we treated 3D cultures of control and IRX3-KO cells grown in osteogenic medium with either vehicle or ML-792 (Fig. 9). As before, the IRX3-KO cells treated with vehicle showed elevated mineralization and expression of osteogenic markers compared to control cells, while inhibition of SUMOylation significantly decreased both mineralization (Fig. 9a) and osteogenic gene expression (Fig. 9b) in the KO cells. These data indicate that SUMOylation at least partially mediates the pro-osteogenic effect of IRX3 ablation.

## Discussion

The present study has identified hundreds of direct target genes of the homeobox transcription factor IRX3 in preadipocytes, providing mechanistic insight into how IRX3 mediates the association between risk variants in the *FTO* locus and increased risk of obesity[12,56]. We previously found a transient, risk genotype-dependent increase in IRX3 during early adipogenesis to promote white over beige adipocyte identity of the mature adipocytes[12,57]. These data suggest an epigenetic effect of IRX3 despite its lack of chromatin or histone-modifying domains. In the current study, we found IRX3 to indirectly affect epigenetics through direct transcriptional regulation of a broad range of genes involved in chromatin remodeling and histone modifications. Moreover, we also found IRX3 to transcriptionally suppress the SUMOylation machinery and thereby reduce global conjugated levels of SUMO, a key PTM involved in modulating the function of chromatin remodelers and histone modifiers[25–28]. Taken together, these data implicate IRX3 as an upstream regulator of SUMOylation and suggest that IRX3 controls both the expression and function of epigenetic regulators.

In agreement with a role of IRX3 in transcriptionally controlling epigenetic factors, we identified profound changes in the open chromatin landscape in IRX3-KO compared to control cells, with more open chromatin in promoters of osteogenic genes and less open chromatin in adipogenic promoters. Consequently, the IRX3-KO cells failed to undergo adipogenesis both in 2D and 3D culture. Conversely, we found the IRX3-KO cells to respond to osteogenic stimulation, while the control cells did not. Finally, pharmacological inhibition of SUMOylation partially reversed the IRX3-KO dependent

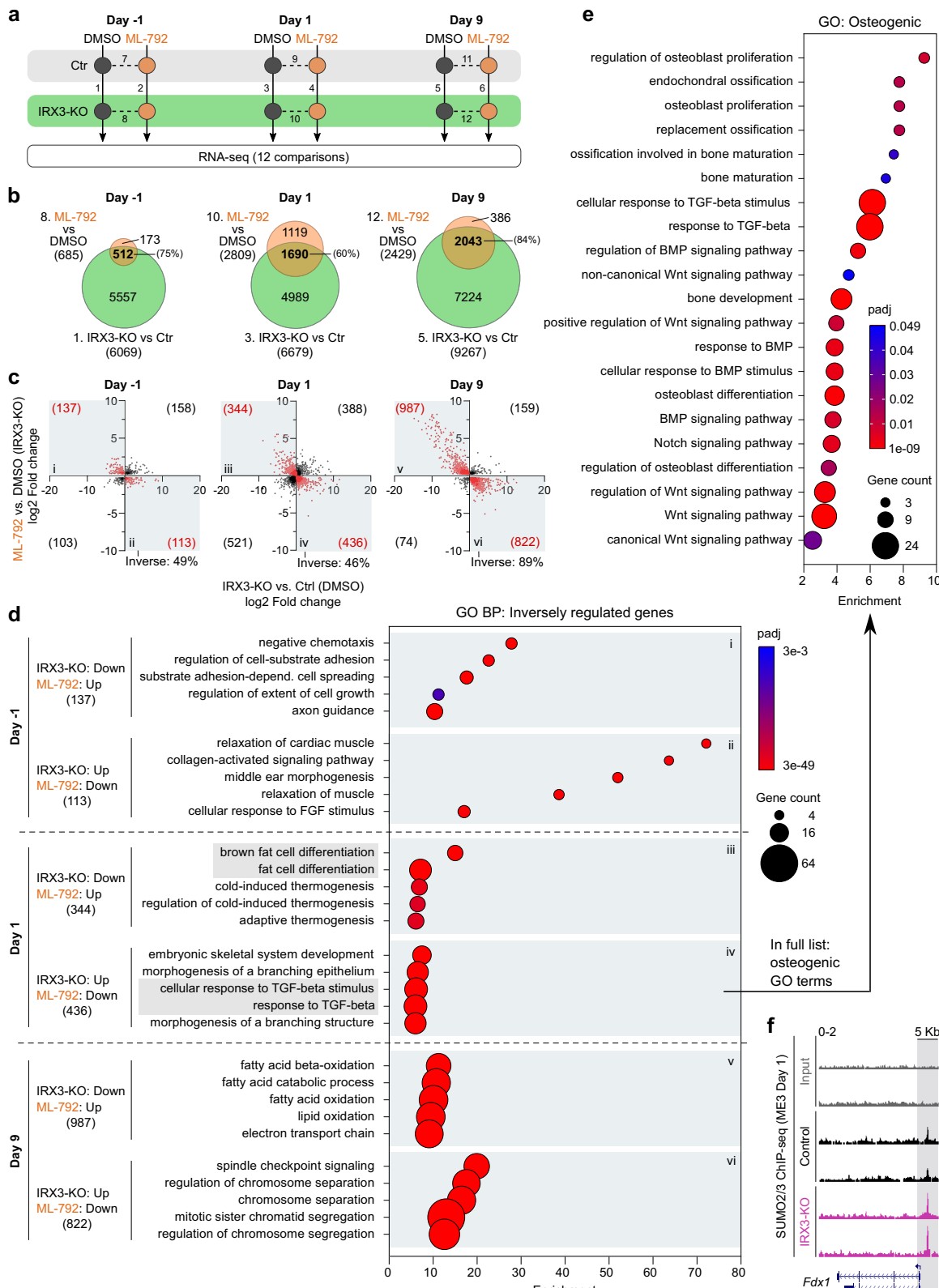

switch from adipogenesis to osteogenesis, demonstrating that this switch is partially mediated via changes in SUMOylation. SUMOylation is well known to control cell fate, and has been shown to be critical to safeguard the identity of somatic cells differentiated from induced pluripotent stem cells, and embryonic stem cells from 2C-like cells[29,30]. In the present study, we found that IRX3 protects MEF identity and preserves adipogenic fate while repressing

osteogenesis, and this effect was at least partially via repression of SUMOylation.

Identification of genes influencing obesity and other complex diseases has long been hampered by the lack of mechanistic information from GWAS data[3,9,58]. While a major leap was achieved by us and others a decade ago by inferring a causal variant in the obesity-associated *FTO* locus, identifying IRX3 and IRX5 as the target genes and

**Fig. 7 | IRX3 and SUMO inversely regulate shared target genes involved in adipogenic and osteogenic differentiation.** ME3 control and IRX3-KO cells were differentiated under adipogenic conditions and treated with either DMSO or 0.5 μM ML-792 from day −2 to −1, day 0 to 1, or continuously from day 0 to 9 (n = 3 replicate wells per condition). RNA was isolated on days −1, 1, and 9 for transcriptomic analysis. **a** Schematic of the experimental design. Days indicate the day of cell harvest; solid lines indicate IRX3-KO vs control comparisons; dashed lines represent ML-792 vs DMSO treatment. Numbers 1-12 denote pairwise comparisons. See also Supplementary Fig. 9. **b** Venn diagrams showing overlap between differentially expressed genes in IRX3-KO vs control cells (comparisons 1, 3, and 5) and in ML-792-treated vs DMSO-treated IRX3-KO cells (comparisons 8, 10, and 12) on days −1, 1, and 9. Percentages indicate the proportion of ML-792 responsive genes also altered in IRX3-KO cells. **c** Scatter plots of log₂ fold changes for overlapping genes

from (b). Red dots, genes inversely regulated by IRX3 and SUMO; black dots, genes regulated in the same direction. Roman numerals indicate quadrants; numbers show gene counts per quadrant. **d** ClusterProfiler GO enrichment analysis of inversely regulated genes from **c**; Benjamini-Hochberg correction, top 5 biological process (BP) terms per quadrant shown. Quadrants i-vi correspond to quadrants in **c**. **e** ClusterProfiler GO terms related to osteogenesis among genes upregulated in IRX3-KO cells and downregulated by ML-792 on day 1 (quadrant iv); Benjamini-Hochberg correction. **f** UCSC Genome Browser tracks of SUMO2/3 ChIP-seq peaks at the *Fdx1* promoter in ME3 control and IRX3-KO cells on day 1. *Fdx1* is part of the "electron transport chain" GO category identified among inversely regulated genes (quadrant v in c-d). Bar graphs show means ± SD. Source data are provided in the Source Data file.

adipocyte precursor cells as the effector cell type[12], the question still remained how these genes mechanistically mediate the effect of the association[3]. Both whole-body ablation[11], as well as adipocyte-specific disruption (adiponectin-IRX3DN)[12] of IRX3 were found to reduce adiposity and body weight, as well as increase thermogenesis and resistance to diet-induced weight gain[11,12], confirming that these genes have a profound impact on adipocyte biology and obesity development. Mechanistically, we identified a transient, genotype-dependent increase in IRX3 and IRX5 expression on days 0-2 of adipogenic differentiation in adipocyte precursor cells, which subsequently inhibited beiging and promoted a white phenotype in mature adipocytes[12]. However, the exact nature of this temporal effect was not investigated. While more recent work by Vámos et al. confirmed reduced thermogenic capacity in human adipose-derived precursor cells from patients homozygous for the *FTO* locus risk variants[57], the authors did not report the expression levels of IRX3. Moreover, Zou et al. reported a conflicting effect of IRX3 on adipocyte beiging and found IRX3 to promote rather than inhibit beiging in primary adipocytes from mice and humans[59]. More recently, the same group used transgenic mouse models to show that constitutive overexpression of human IRX3 in brown and beige adipocytes (*Ucp1*-CRE) treated with chronic cold stimulation resulted in elevated UCP1 protein levels in iWAT, accompanied by increased oxygen consumption and reduced fat mass and body weight in vivo[60]. In contrast, inducible overexpression of IRX3 for 5 days postnatally in 8-week old mice only resulted in weak induction of UCP1 levels[60]. Recently, another group reported that IRX3 levels in iWAT was positively associated with leanness and elevated browning in mice with randomly divergent response to a high-fat diet[61]. Thus, there has been a need for more mechanistic investigations into the role of IRX3 in adipocytes[62]. Zou et al. found reduced expression of thermogenic markers like UCP1 in beige adipocytes following lentiviral-mediated knockdown of IRX3. Therefore, if IRX3 primarily acts through uncoupling, this finding should lead to increased lipid accumulation in the *IRX3*-kd cells. Instead, they found reduced lipid accumulation in the *IRX3*-kd cells[59]. We reasoned the repressive effect on beige gene expression following *IRX3*-kd could be confounded by a suppressive effect on adipogenesis per se. Indeed, using CRISPR/Cas9 to completely knock out IRX3 in beige preadipocyte cells, we previously demonstrated a near complete inhibition of adipogenesis[14], highlighting a strong positive effect of IRX3 on adipogenesis. This finding was subsequently corroborated by the Wang group who showed increased adipogenesis in primary preadipocytes isolated from brown and beige adipose tissues overexpressing IRX3[60]. The current study broadened this finding, showing that IRX3 is vital for maintaining an epigenetic landscape and transcriptional program tuned towards adipogenesis in MEFs, while simultaneously inhibiting chondrogenesis and osteogenesis. This apparently positions IRX3 as a master regulator of mesenchymal fate, which precedes the white versus beige lineage choice in committed preadipocytes. Therefore, positive effects of IRX3 ablation on thermogenesis may be masked by the inhibitory effect on adipogenesis. Future studies should further

investigate this issue by applying mouse models with inducible KO or KI of IRX3 specifically in adipose tissues, and/or selectively in adipose-derived mesenchymal stem cells (AdMSCs).

Our finding of IRX3 as a guardian of adipogenic fate decision in AdMSC-like cells in vitro underscores its importance in adipose tissue development in vivo, and is supported by KO-studies in mice[11,12], as discussed above. However, it also raises the question of what role IRX3 plays in bone-marrow derived mesenchymal stem cells (BM-MSCs). These cells can also differentiate into adipocytes[63], and increased bone marrow adiposity is closely linked to osteoporosis (as reviewed in ref. 64). This disease is characterized by progressive loss of bone strength with age, and has limited treatment options[65]. Thus, identifying treatment targets that can reduce adipogenesis in bones is of great interest. One could imagine IRX3 and SUMOylation to be such targets, as well as the epigenetic enzymes they regulate. However, while IRX3 represses osteogenesis and promotes adipogenesis in adipose tissue, is appears to have an opposite effect in bones. A recent study showed that mice with global KO of either IRX3 or IRX5 exhibit osteopenia, a precursor state of osteoporosis, with impaired osteoblast differentiation and increased bone marrow adipogenesis[66]. Moreover, in global IRX5-KO mice, additional deletion of IRX3 specifically in hypertrophic chondrocytes (HC) of the bone marrow exacerbated these effects[66]. Thus, in the bone marrow, IRX3 was found to be critical for osteogenic development by inhibiting adipogenesis, and this effect was partially dependent on IRX5. Previous studies have demonstrated cooperative roles between IRX3 and IRX5 in heart development[67], and IRX3/IRX5 double-KO mice do exhibit embryonic skeletal and limb abnormalities, while single IRX3 or IRX5 KO mice do not[68]. Similarly, osteoblast-specific IRX3 depletion in global IRX5-KO mice leads to abnormal craniofacial development, but no limb malfunctions[69], while single, global KO of either IRX3 or IRX5 alone does not affect craniofacial development[69,70]. In contrast, adipose-specific disruption of IRX3 alone is sufficient to reduce body weight and fat storage in adipose depots[12], suggesting a stronger and more autonomous role of IRX3 in adipose tissues compared to the bone marrow. However, in dental mesenchymal stem cells in vitro, siRNA-mediated knockdown of IRX3 impaired both proliferation and osteogenic differentiation in vitro[71]. Taken together, several lines of evidence suggest an inverse role of IRX3 on mesenchymal fate decision depending on the origin of the cells, with an activating role on adipogenesis in fat tissues and an inhibitory role in bone tissues. The reasons for this discrepancy remain to be uncovered, but important clues emerged from a very recent paper by Chen et al, who reported sex-dependent effects on adipogenesis in the bone marrow of mice with double heterozygous deletion of IRX3 and IRX5[72]. Strikingly, while male mice showed the expected increase in bone marrow adipogenesis in the KO mice, female KO mice exhibited reduced bone marrow adipogenesis[72], in line with the observed effects of IRX3 ablation in fat. These results highlight the importance of including both sexes in mice studies in general and when studying IRX3 in particular. Future in vivo studies should aim to investigate

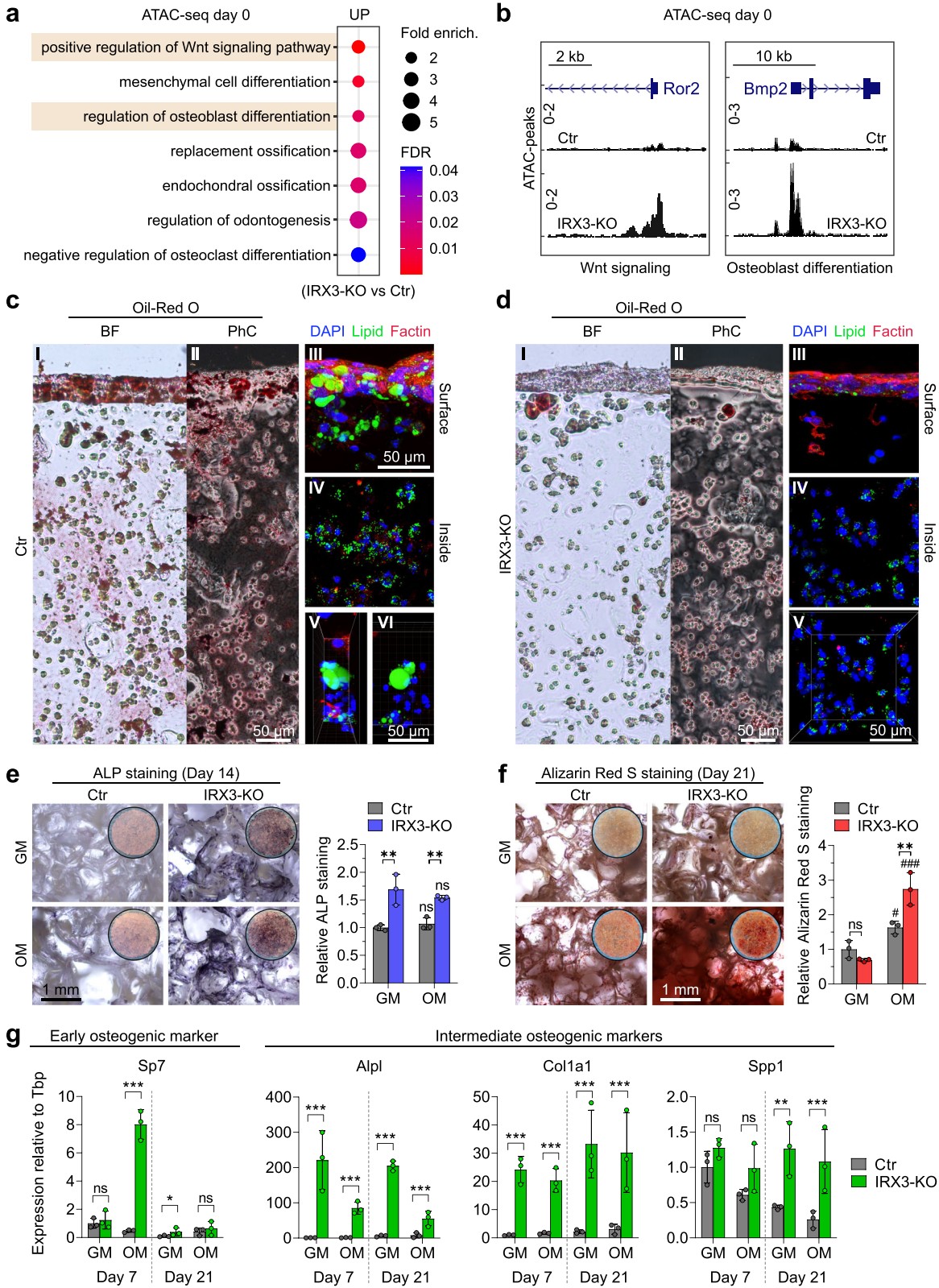

**a** ATAC-seq day 0

UP

positive regulation of Wnt signaling pathway
mesenchymal cell differentiation
regulation of osteoblast differentiation
replacement ossification
endochondral ossification
regulation of odontogenesis
negative regulation of osteoclast differentiation

Fold enrich. (2, 3, 4, 5)
FDR (0.04, 0.03, 0.02, 0.01)

(IRX3-KO vs Ctr)

**b** ATAC-seq day 0

2 kb — Ror2 — Wnt signaling
10 kb — Bmp2 — Osteoblast differentiation
ATAC-peaks — Ctr / IRX3-KO

**c** Oil-Red O (Ctr)
BF | PhC | DAPI Lipid Factin
I | II | III (Surface) 50 μm
IV (Inside)
V | VI — 50 μm / 50 μm

**d** Oil-Red O (IRX3-KO)
BF | PhC | DAPI Lipid Factin
I | II | III (Surface) 50 μm
IV (Inside)
V — 50 μm / 50 μm

**e** ALP staining (Day 14)
Ctr | IRX3-KO
GM / OM — 1 mm
Relative ALP staining (Ctr, IRX3-KO)
GM ** ; OM ns / ** / ns

**f** Alizarin Red S staining (Day 21)
Ctr | IRX3-KO
GM / OM — 1 mm
Relative Alizarin Red S staining (Ctr, IRX3-KO)
GM ns ; OM # / ** / ###

**g** Early osteogenic marker — Sp7
Intermediate osteogenic markers — Alpl, Col1a1, Spp1
Expression relative to Tbp
Ctr / IRX3-KO

both adipose tissues and bones from the same mice of both sexes in response to further clarify the effect of sex on IRX3 function. Moreover, IRX3 manipulation should be tissue specific and restricted to adipocytes or bone marrow to better investigate organ-specific effects of IRX3 manipulations on adipogenesis and osteogenesis. Finally, a particular emphasis should be placed on investigating epigenetic changes in the two tissues.

Although the direct IRX3 target genes may vary between different cell types and tissues, many of the genes identified in preadipocytes are nevertheless involved in biological pathways present in multiple cell types. For example, our data show that IRX3 controls genes in the Wnt/β-catenin and Hippo signaling pathways in preadipocytes, and these pathways have been shown by others to also be regulated by IRX3 and/or IRX5 in kidney development[73]. Moreover, our

**Fig. 8 | IRX3 ablation promotes osteogenesis. a** ATAC−seq was performed in ME3 control and IRX3-KO cells on day 0 of adipogenic differentiation ($n = 3$). Top GO terms among promoter regions with increased chromatin accessibility in IRX3-KO cells are shown. See also Supplementary Fig. S9. **b** UCSC Genome Browser tracks of ATAC−seq peaks at promoters of genes associated with osteoblast-related GO terms in **a**, showing enhanced accessibility in IRX3-KO cells. **c** Adipogenic differentiation of ME3 control cells in 3D culture. I brightfield image of Oil Red O−stained lipid droplets; II, phase contrast image of lipid staining; III, fluorescence image of the hydrogel surface stained with DAPI (blue), Bodipy (green), and F-actin (red); IV, fluorescence image of the hydrogel interior; V–VI, magnified views. Scale bars, 50 µm. **d** Adipogenic differentiation of IRX3-KO cells in 3D culture. I–V as in (**c**). Scale bars, 50 µm. **e** ME3 control and IRX3-KO cells cultured in growth medium (GM) or osteogenic medium (OM), followed by alkaline phosphatase staining. Brightfield

images (left) and quantification (right) shown ($n = 3$). Scale bar, 1 mm. **$p_{adj} < 0.01$, control vs IRX3-KO; ns, not significant, OM vs GM; two-way ANOVA with Holm−Sidak correction. **f** Alizarin Red S staining to assess mineralization after osteogenic differentiation in ME3 control and IRX3-KO cells. Brightfield images (left) and quantification (right) shown ($n = 3$). Scale bar, 1 mm. **$p_{adj} = 0.003$, control vs IRX3-KO; ns, not significant; #$p_{adj} = 0.03$; ###$p_{adj} < 0.001$, osteogenic vs growth medium; two-way ANOVA with Holm-Sidak correction. **g** Expression of early, intermediate, and late osteogenic markers in ME3 control and IRX3-KO cells cultured in GM or OM ($n = 3$). *$p_{adj} = 0.04$, **$p_{adj} = 0.003$, ***$p_{adj} < 0.001$, control vs IRX3-KO; two-way ANOVA with Holm−Sidak correction. Data were ln-transformed before statistical calculations. Bar graphs show means ± SD. Source data are provided in the Source Data file.

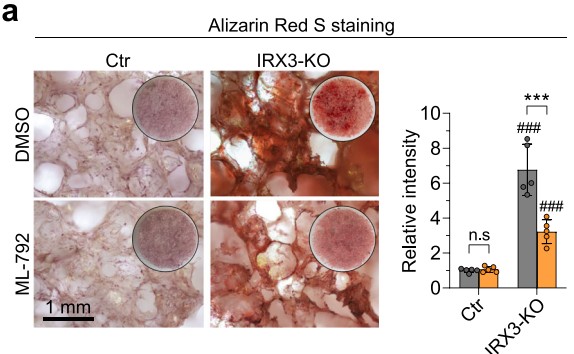

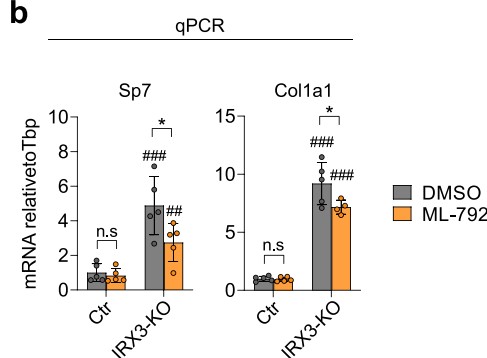

**Fig. 9 | Inhibition of SUMOylation represses IRX3-KO-dependent osteogenesis.** To assess whether SUMOylation mediates the IRX3-KO-dependent adipogenesis, ME3 control and IRX3-KO cells were subjected to osteogenic differentiation in 3D culture for 21 days and treated with either vehicle or 0.5 µM ML-792 throughout the differentiation. **a** Alkaline phosphatase staining. Brightfield view (left) and quantification (right) shown. Data represent $n = 5$ replicate wells from a single experiment. Scale bar, 1 mm. ***$p_{adj} < 0.001$, DMSO vs. ML-792. ns, not significant; ###$p_{adj} < 0.001$, control vs. IRX3-KO. **b** Cells were treated as in (**a**),

except RNA was harvested on day 7 and expression of early and intermediate markers of osteogenesis was measured by qPCR. Data normalized to the reference gene *Tbp* and shown relative to the control group. Data represent $n = 5$ replicate wells from a single experiment. *$p_{adj} < 0.05$, ns, not significant, DMSO vs. ML-792; ##$p_{adj} = 0.002$, ###$p_{adj} < 0.001$, control vs. IRX3-KO; two-way ANOVA with Holm−Sidak correction. Data were square root-transformed before statistical calculations. Bar graphs show means ± SD. Source data are provided in the Source Data file.

comprehensive lists of processes affected by IRX3 ablation reveal myriads of GO terms related to other non-adipocyte cell types, tissues and disease states (Supplementary Data 14). Thus, our data may help understand how IRX3 mediates its effects in several other conditions in which IRX3 has been implicated, including type 2 diabetes[10], metabolic inflammation[74], fertility[75], neurogenesis[76,77], glioblastoma[78], acute myeloid leukemia[79–82], microvascularization[83], and heart development and function[67,70,84–87]. Similarly, we have shown an almost equally pleiotropic effect of inhibition of SUMOylation in both control and IRX3-KO MEFs (Supplementary Data 15), a cell type that usually can differentiate into both mesenchymal and non-mesenchymal lineages[88]. Thus, our work also highlights potential other disorders that may benefit from inhibition of SUMOylation. For example, we observe significant enrichment of genes related to several cancer types among downregulated genes following inhibition of SUMOylation. These data are in line with several phase 1/2 clinical trials for TAK-981, an in vivo compatible inhibitor of SUMOylation, against lymphomas and solid tumors (#NCT03648372, #NCT04074330, #NCT04776018, #NCT04381650 and #NCT04065555).

The master-regulatory role of IRX3 may not be surprising given its classification as a homeobox factor[15,89,90], a highly diverse super-class of transcription factors with critical roles during development, patterning formation, and embryonic stem cell function[15,16]. Interestingly, SUMO has been reported to regulate the function of several homeobox factors, including NKX2.5 via its direct SUMOylation[91] and EGL5 and MAB5 via SUMOylation of the SOP2 Polycomb repressor protein[92]. However, the present report shows a homeobox factor acting

upstream of SUMO, and our discovery fills an important general knowledge gap as no upstream SUMO regulator has been identified previously (at least to our knowledge). This finding should stimulate further studies of the upstream mechanisms controlling SUMO and warrants investigations into the role of IRX3 in a broad range of known SUMO-controlled processes, including stemness and cell identity, chromatin and epigenetic remodeling, stress response, cell division, senescence, protein stability and more, as reviewed in ref. 27. Moreover, potential implications for diseases where SUMO is involved, such as acute promyelocytic leukemia (APL) and neurodegenerative diseases (Alzheimer's, Parkinson's, and Huntington's disease)[27] remains to be examined. Finally, provided the importance of both IRX3 and SUMO in regulating epigenetics, these factors should be considered when studying the etiology of disease states where epigenetics are known to play a role, including obesity[21], cardiovascular disease, cancer, and neurodevelopmental disorders[93].

In the present study, we found SUMOylation to have a strong inhibitory role on adipogenesis in beige ME3 cells, which could be reversed by pharmacological inhibition of SUMOylation. This finding contrasts several other reports suggesting that SUMOylation promotes adiposity. For instance, *Sumo1*-KO mice fed a high-fat diet weigh less and possess smaller and fewer adipocytes compared to WT mice[94]. Similarly, mice lacking the SUMO conjugating enzyme UBC9 specifically in WAT were severely lipodystrophic[95]. Mice treated with the SUMOylation inhibitor TAK-981 failed to fully develop perigonadal WAT[96] and *Ubc9*-kd in 3T3-L1 cells inhibits adipogenesis[97]. Interestingly, mouse embryonic fibroblasts (MEFs) with SUMO1

deficiency demonstrated reduced adipogenesis, but only during late stages of differentiation[94]. These data suggest that SUMO may have temporal effects on adipogenesis of white adipocytes. In support, we recently demonstrated that waves of SUMOylation controls adipogenesis in 3T3-L1 cells by promoting the timely downregulation of preadipocyte genes and activation of mature adipocyte genes[50]. Moreover, in the present study, the pro-adipogenic effect of inhibiting SUMO was completely lost if the inhibitor was administered after day 2 of differentiation, and when administered only on days 4–9 it inhibited the differentiation in line with the abovementioned reports. Still, when the inhibitor was administered from day 0 and throughout differentiation, inhibition of SUMOylation had a clear pro-adipogenic effect.

Other studies support our finding of SUMOylation as overall anti-adipogenic. For example, ablation of the desumoylases SENP1 or SENP2 has also been shown to inhibit adipogenesis via hyperSUMOylation of CEBPB, SETDB1, or SHARP1, resulting in impaired PPARγ activity[98–100]. Moreover, PPARγ itself can be SUMOylated[39] and in vitro mutagenesis of the K107 SUMOylation site clearly demonstrates the inhibitory effect of SUMO on PPARγ transcriptional activity in luciferase assays[38,101]. In the present study, we show a similar repressive effect of SUMO on PPARγ/PGC-1α transcriptional activity. These findings could explain how downstream target genes of PPARγ were downregulated on day 1 of adipogenic differentiation in IRX3-KO cells while PPARγ itself was not. Moreover, while inhibition of SUMOylation stimulated adipogenesis, we found it to repress osteogenesis in the ME3 IRX3-KO cells. Thus, our data suggest that SUMOylation can promote osteogenesis. In agreement with this, Zhang et al. found TGFB-mediated SUMOylation of PPARγ to inhibit adipogenesis and promote osteogenesis[37], and Liu et al. demonstrated a switch from osteogenic to adipogenic differentiation of bone-marrow-derived mesenchymal stem cells following pharmacological inhibition of SUMOylation[36]. Taken together, our results extend our understanding by showing that IRX3 controls an adipocyte-osteocyte epigenetic switch in mesenchymal-like precursor cells, and this process is at least partially mediated via SUMOylation.

Mesenchymal lineage determination is regulated by complexes of Trithorax group proteins, such as the COMPAS and SWI/SNF family members (as reviewed in refs. 102,103. The former includes various SET domain methyltransferases, including SETD1A, SETD1B, and KMT2A (Mll1)[102]. The genes encoding these proteins were all found to be bound by IRX3 and upregulated following IRX3-KO (Fig. 1f and 2d), and we previously found both SETD1A and KMT2A to be SUMOylated in adipocytes[50]. SET domain-specific truncation of KMT2A in mice dramatically reduced mono-methylation of H3K4 on specific *Hox* genes, altering their expression levels and leading to skeletal defects[104]. Interestingly, SENP3-mediated deSUMOylation of KMT2A complexes was found to be necessary for osteogenesis in dental follicle stem cells[105]. Thus, our data point to SETD1A and KMT2A as possible mediators of the anti-osteogenic role of IRX3 in adipocyte precursor cells. However, their exact contribution remains to be determined.

Irx family members have generally been reported to have a repressive effect on target genes during development[31,106–111], however, these studies only investigated a limited number of selected target genes. In the present study, we used ChIP-seq in combination with RNA-seq in control and IRX3-KO cells to systematically investigate genome-wide transcriptional effects of IRX3 in preadipocytes. Here, we identified approximately equal numbers of up- and downregulated direct IRX3 target genes following IRX3-KO, suggesting either that: 1) IRX3 may also have an activating effect on target gene expression, or 2) that IRX3 binding is independent from its effect on gene expression or 3) that the IRX3 binding profile is profoundly different in 3T3-L1 and ME3 cells and that the downregulated genes observed in ME3 cells with IRX3 ablation are downstream effects. While the latter two cannot be excluded, we find it unlikely that IRX3 solely plays a repressive role. The direction of change in gene expression likely depends on

interactions with other transcription factors and/or co-regulators or post-translational changes to IRX3. For example, in the heart, IRX3 has been reported to activate the expression of *Gja5* (encoding the CX40 protein)[84,112], and this was found to occur via physical interactions with TBX5 and NKX.5[70]. Moreover, work by others demonstrates opposing roles of the N- and C-terminal domains of IRX2 and IRX4 on transcriptional regulation[108,111] and that phosphorylation status of the respective domains determines the overall transcriptional effect of the protein[111]. Thus, protein-protein interactions between IRX3 and other factors, as well as the phosphorylation status, may impact the protein conformation, and thereby the transcriptional function of IRX3, although this hypothesis should be addressed in future studies.

While SUMOylation clearly mediates a vital part of IRX3's effect on adipogenesis, as demonstrated by the reversed phenotypes following ML-792 treatment, it likely does not mediate the entire effect. Most notably, we found IRX3 to bind the promoters and alter the expression of a broad range of histone modifying enzymes and chromatin remodelers, which may act independently of SUMO or at least constitute a different substrate pool for SUMOylation. Secondly, while most of the SUMO-responsive genes were also changed by IRX3-KO, IRX3 additionally controlled a large set of genes that were unaffected by inhibition of SUMOylation. For example, we identified *Gas1* as a direct IRX3 target gene that increased 12-fold after IRX3-KO (Supplementary Data 6). The GAS1 protein is a coreceptor that promotes Sonic Hedgehog (SHH) signaling, whose activation is marked by increased *Gli1* expression (reviewed in ref. 113). Since we previously also found *Gli1* expression to increase in response to IRX3-KO[14], our data clearly indicate increased SHH signaling in the IRX3-KO cells. The SHH pathway is well known to inhibit adipogenesis and promote osteogenesis[114,115], and is likely a contributing factor in the pro-osteogenic capacity of the IRX3-KO cells. The GAS1 protein was not found to be a SUMO substrate[50], thus the osteogenic effects of this factor is IRX3-dependent, but SUMO-independent.

In this study, we observed a significant increase in global SUMOylation levels in adipocyte precursor cells with IRX3-KO compared to controls. Moreover, SUMOylation inhibition and rescue experiments established that the changes in global SUMOylation levels affected the gene expression and phenotype of the cells. Therefore, it was surprising to observe that changes in global SUMOylation levels did not translate into abundant changes in SUMO occupancy on the chromatin. Furthermore, among the limited number of differentially SUMOylated loci, more showed reduced rather than increased levels of SUMOylation. Several non-exclusive explanations may account for this apparent discrepancy. First, an increase in global SUMOylation detected by Western blot does not necessarily translate into increased SUMOylation at the chromatin, as previously observed in 3T3-L1 cells[50]. Additionally, the presence or absence of SUMO at target genes can correlate positively or negatively with gene expression, depending on gene subsets, differentiation stage, or cell type (i.e., pre-adipocyte vs. mature adipocyte)[28,50,116]. This context-dependent effect of SUMOylation suggests that multiple dynamic mechanisms regulate SUMOylation simultaneously to fine-tune chromatin-based processes.

Since IRX3 is likely not the sole regulator of SUMOylation, other repressors or activators could compensate or become more prominent in its absence. Moreover, IRX3 regulates multiple genes apart from those directly involved in SUMOylation. Thus, some of the reductions in SUMO occupancy at the chromatin could result from secondary effects of other IRX3 target genes rather than those directly affecting SUMOylation levels. Furthermore, measuring SUMO occupancy by ChIP-seq does not specify which TFs are bound and how the dynamics of TF binding, including those not SUMOylated, might have changed in response to elevated global SUMOylation. Our observations may also reflect compensatory mechanisms counteracting chromatin-associated SUMOylation in IRX3-KO cells[117]. Similar compensatory mechanisms have been described in transcriptional

regulation, where the knockout of general TFs such as SAGA paradoxically leads to mRNA stabilization, counterbalancing otherwise severe transcriptional reductions[118].

Finally, the SUMOylation patterns observed may simply reflect a cell identity conversion. Our IRX3-KO model is stable, and the heterogeneous SUMOylation landscape observed in Fig. 6 (regions of both higher and lower SUMOylation) may be a direct consequence of the stable conversion of preadipocytes into preosteoblast-like cells. Similar SUMOylation shifts have been reported in embryonic stem cells (ESCs) versus 2C-like cells and MEFs[29,30], and during pre-adipocyte-to-mature-adipocyte transitions[50]. To further clarify the effects of Irx3 loss on SUMOylation at the chromatin, future studies could employ inducible or acute degradation systems (e.g., AID degron) instead of stable KO. This approach would allow for a more precise examination of the early consequences of Irx3 depletion on SUMOylation levels on chromatin and help delineate how increased SUMOylation promotes the switch from osteogenesis to adipogenesis.

Our findings clearly implicate IRX3 in epigenetic regulation of adipogenic identity, which paves the way for further investigations into a model where transient changes in IRX3 mediate the effect of *FTO* locus risk variants on human adipocyte beiging via changes in epigenetics. However, our study has limitations. First, our study was performed in mice and not humans. Second, we employed SVF-derived primary cells and ME3 cell lines, which differ from bona fide mesenchymal stem cells. Third, while we have shown an epigenetic effect of complete IRX3 ablation, the modest and transient alterations in IRX3 levels in response to the *FTO* haplotypes may not be sufficient to induce epigenetic changes. Fourth, due to the profound effect on adipocyte differentiation per se obtained in this study, likely due to the complete and constitutive IRX3 ablation, it was not possible to assess the effect on white versus beige development. Future studies should address these shortcomings.

In this study, we found that IRX3 is a master transcriptional regulator of histone and chromatin remodeling enzymes in adipocyte precursor cells, promoting adipogenesis and suppressing osteogenesis. Moreover, we showed that IRX3 is an upstream regulator of SUMOylation and found that repression of SUMOylation is a key mediator of the pro-adipogenic effect of IRX3. More broadly, our study has identified a large set of direct IRX3 target genes and SUMOylation-sensitive genes with pleiotropic effects, which may be relevant for understanding the role of IRX3 and SUMOylation in non-adipocyte conditions and diseases.

## Methods

### Animals

Animal studies were conducted in accordance with the UK Animals (Scientific Procedures) Act (1986). Mouse studies were performed under the Home Office License 30/2642. Procedures were approved by the MRC Harwell Animal Welfare and Ethical Review Board (AWERB) (PPL 30/2642 and 30/3146). C57BL/6NJ (B6N) mice were housed according to UK Home Office welfare guidelines in a 12 hr light / dark cycle at a temperature of $21 \pm 2\,°C$ and humidity of $55 \pm 10\%$. Mice were fed ad libitum and had free access to water (25 ppm chlorine). Mice that were used for primary pre-adipocyte isolation were fed SDS maintenance chow (RM3, 3.6 kcal/g).

### Cells

**Isolation, culturing, and differentiation of adipocyte precursor cells from gWAT and iWAT.** Mouse primary adipocytes were isolated essentially as previously described[119]. Briefly, mouse WAT depots were excised from 6-week-old C57BL/6NJ (B6N) female mice and placed in PBS. Tissues were minced and digested in 5 ml per depot digestion buffer (sterile Hank's Balanced Salt Solution (HBSS; H8264), 0.8 mg/ml collagenase type 2 (Worthington Biochemical Corporation, NJ, USA;

LS004174), 3% bovine serum albumin (BSA) (with fatty acids) and incubated in a 37 °C water bath for 60-75 min with shaking every 10 min by hand. When digestion was complete, tubes were centrifuged for 3 min at 300xg to separate floating mature adipocytes from the supra-vascular fraction (SVF), containing adipocyte precursors. The supernatant containing the floating adipocyte fraction was removed, and the cell pellet re-suspended in pre-warmed growth media consisting of DMEM GlutaMax (# 10569010 DMEM, high glucose, GlutaMAX™ Supplement, pyruvate) supplemented with 10% Fetal Bovine Serum (Gibco, New York, NY, USA 10082-147) and 1% Penicillin-Streptomycin (5,000 U/mL) (Gibco, 15070063) and grown at 37 °C and 5% CO2. The cell solution was subsequently filtered through a 40 μm nylon mesh and plated on a 10 cm dish. Media was replaced the next day. For differentiation experiments, preadipocytes were seeded in density 100 K cells/ml in 12-well or 6-well dishes and grown to confluence. 2 days post-confluence, preadipocytes were induced to differentiate using induction medium (DIFF1) containing growth medium supplemented with 0.5 mM 3-Isobutyl-1-Methylxathine (IBMX) (Sigma-Aldrich, St. Louis, MO, USA I5879), 1 μM Dexamethasone (Sigma-Aldrich, D2915), and 5 μg/ml human insulin (Sigma-Aldrich, I9278). At day 4 of differentiation, media were changed to maintenance media (DIFF2) containing growth medium supplemented with insulin only. Media was changed every 2 days and differentiation was complete at day 7-9 post adipogenic induction. These cells were used for ATAC-seq and ChIP-seq as indicated.

**ME3 WT and IRX3-KO cells.** The wild type (WT) ME3 cell line was previously generated[120]. Cells were grown in AmnioMAX -C100 medium supplemented with 7.5% FBS, 7.5% C100 (all from Thermo Fisher Scientific, Waltham, MA, USA), 1% penicillin-streptomycin (Sigma, St. Louis, MO, USA), and 2 mM L-glutamine (Sigma) at 37 °C and 5% CO2. Cells were initiated to differentiate two days post confluency (day 0) by addition of 5 ug/mL Insulin (Sigma), 1 μM Dexamethasone (Sigma), 0.5 mM isobutyl methylxanthine (IBMX) (Sigma) and 1 μM Rosiglitazone (Cayman Chemical, Ann Arbor, MI, USA) to the basal medium. From day 2 to day 4, only insulin was added to the basal medium,m and from day 4 to 7 cells were grown in the basal medium. Cells were harvested for ATAC-seq on days 0, 1, and 7 of differentiation in two independent experiments. CRISPR-Cas9 control and IRX3-KO cells were generated from WT ME3 cells, and their global gene expression was previously assessed by RNA-seq on days 1 and 7 of differentiation[14]. These cells were also used for adipogenic and osteogenic differentiation, qPCR, Bodipy staining, ATAC-seq, Western blotting (WB), Chromatin Immunoprecipitation (ChIP), and luciferase transactivation assays. Adipogenic differentiation was performed as described for WT ME3 cells. For osteogenic differentiation, the cells were treated with 50 μg/mL L-ascorbic acid 2-phosphate, 10 nM dexamethasone, and 10 mM β-glycerophosphate (all from Sigma-Aldrich-Aldrich) in the complete growth medium. The medium was changed every three days until day 14 for assessment of alkaline phosphatase activity, or day 21 for Alizarin red S staining.

### Cell differentiation in 3D

For adipogenic 3D differentiation, ME3 cells were first encapsulated in Gelatin methacryloyl (GelMA) hydrogel. Briefly, GelMA solutions were reconstituted by dissolving purified GelMA prepolymer powder (X-Pure GelMA 160P60 RG, Rousselot Biomedical, Belgium) in the growth medium in presence of 1% w/v of Lithium Phenyl(2,4,6-trimethylbenzoyl)phosphinate (L0290, Tokyo Chemical Industries, Co., Japan) on a magnetic stirrer at 37 °C under sterile conditions in a dark environment. The ME3 cells were added in the solution with a density of 5 million cells per ml, which was then transferred onto molds in a diameter of 5 mm. Photo-polymerization was conducted using a dental curing lamp set for 1 single cycle of 10 seconds at a light intensity of 1200 mW/cm². Adipogenesis was induced essentially as

for 2D, using the same differentiation components as previously described[14], but the cells were kept two more days in the final medium for a total differentiation time of 9 days.

For osteogenic 3D differentiation, highly porous 3D scaffolds for osteogenic differentiation were fabricated as previously described[121]. Briefly, a co-polymer Poly(L-lactide-co-trimethylene carbonate) lactide was dissolved in chloroform and then mixed with sodium chloride particles ranging in size from 90 to 600 μm. After the complete evaporation of the solvent, the constructs were formed, each possessing a diameter of 12 mm and a thickness of 1.2 mm. To eliminate the presence of sodium chloride, the scaffolds were thoroughly washed with distilled water. The scaffolds were placed in 48 well plates where 500,000 cells in 300 μL of the culture medium were seeded per scaffolds. Osteogenic differentiation in 3D was performed as in 2D for 21 days as described above.

### Lipid staining

Adipogenic differentiation was assessed on formaldehyde fixed cells by Oil-red-O lipid staining as previously described[14] or on live cells using Bodipy 493/503 (Sigma-Aldrich) lipid staining. For Bodipy staining, the culture medium was replaced with a Bodipy solution (1 μg/mL, freshly prepared in OptiMEM by vigorous vortexing) and incubated at dark at 37 °C for 30 min. The staining solution was removed and replaced with regular growth medium before being imaged by fluorescent microscopy using a Leica SP8 confocal microscope (Leica, Germany) for 3D cultures and the Incucyte S3 Live-Cell Analysis System (Essen Biosciences, Ann Arbor, MI, USA) for 2D cultures.

### Immunofluorescence

Samples from adipogenic and osteogenic 3D cultures were fixed in 4% PFA for 15 min followed by permeabilization in 0.1% Triton X-100 in PBS for 15 min at room temperature. The samples were then incubated in a blocking buffer consisting of 10% normal goat serum (NGS: ab7481; Abcam, USA) in 0.05% Tween-20 in PBS (PBSTw) for 60 min at room temperature. After blocking, the samples were incubated with anti-Runx2 antibody (1:250, ab23981; Abcam, USA) in PBSTw at 4 °C overnight. After washing in PBSTw 6 times, the samples were incubated with goat anti-rabbit antibody Alexa Fluor 546 (1:500, A11010; Invitrogen) for 1 h at room temperature simultaneously with 4′,6-diamidino-2-phenylindole (DAPI: 1:2500, D9542; Sigma-Aldrich-Aldrich, USA) and Phalloidin Alexa488 (1:500, A12379; Invitrogen, USA) for nuclear staining and filamentous actin staining, respectively. Image acquisition was performed as z-stack images of 250 μm in depth in 3 separate channels using a Leica SP8 confocal microscope (Leica, Germany).

### Alkaline phosphatase staining

Early osteogenic differentiation was assessed on day 14, and the collected samples were fixed with 4% paraformaldehyde (PFA) for a duration of 1 minutes at room temperature. Subsequently, the samples were incubated in BCIP®/NBT solution (B5655; Sigma-Aldrich-Aldrich) for 60 minutes at room temperature. For quantification, the substrate was extracted by incubating the samples with 100 mM cetylpyridium chloride overnight at room temperature. The absorbance of the extracted solution was measured at 540 nm using a Varioskan™ LUX multimode microplate reader (VLBL00D0; Thermo Scientific, Finland).

### Alizarin red S staining

Late osteogenic differentiation was assessed on Day 21, and the collected samples underwent fixation in 4% PFA for 40 minutes. Subsequently, the samples were gently washed with Milli-Q® water. To visualize mineralized nodes, a 0.2% solution of Alizarin Red S (A5533; Sigma-Aldrich-Aldrich) was applied to the samples for 20 minutes, followed by six rinses with Milli-Q® water. For quantification, the dye was extracted from the samples using 100 mM cetylpyridium chloride,

which was allowed to incubate overnight at room temperature. The dye extract was then subjected to absorbance measurement at 540 nm using the microplate reader.

### Pharmacological inhibition of SUMOylation

Cells were treated with 0.5 μM or 1 μM of the ML-792 SUMOylation inhibitor on indicated days of adipogenic or osteogenic differentiation. The inhibitor specifically targets the E1 SUMO activating enzyme (Sae), leading to loss of global SUMOylation without affecting neddylation or ubiquitination[122].

### ChIP and ChIP-seq

**Sample preparations and chromatin immunoprecipitation for IRX3 ChIP-seq.** Primary pre-adipocytes from gWAT and iWAT of mice were isolated and differentiated as described above. For each depot, cells from 6-12 animals aged 6 weeks were pooled and digested. Cells were collected on days −1 and/or 1 as indicated in one to two independent experiments. Cells were fixed by addition of 1/10 volume of a freshly prepared formaldehyde solution containing 11% formaldehyde, 0.1 M NaCl, 1 mM EDTA pH 8.0, and 50 mM HEPES pH 7.9 (final concentration: 1% formaldehyde, 9 mM NaCl, 90 μM EDTA pH 8.0, and 4.5 mM HEPEs pH 7.9) to the culture media. The fixation reaction was stopped after 15 minutes of agitation at RT by addition of 1/20 volume of 2.5 M glycine (final 120 mM). The cells were thereafter kept on ice and washed twice with 0.5% Igepal in PBS. In the second wash, 1 mM PMSF was added. Pellets were snap-frozen in liquid nitrogen and stored at −80 °C until shipment on dry ice to Active Motif for IRX3 ChIP-sequencing, where in-house protocols based on published methods were used[123–125]. Immunoprecipitation (IP) was performed using 3.2 μg anti-IRX3 antibody (ab25703, lot GR198517-1, Abcam) and 25 μg chromatin pr IP. Library prep was performed, but before deep sequencing, an antibody validation test was performed to ensure the antibody was of ChIP-seq quality. Briefly, the test was performed by shallow sequencing (10 million reads) of the libraries to assess whether the expected number of peaks for a transcription factor was observed. The antibody passed the test, and samples were subjected to deep sequencing (30 million reads). Two pooled DNA input samples were used as negative control.

**IRX3 ChIP-seq data analysis.** Reads were aligned to mouse genome (mm10) using Bowtie2 (Version 2.3.4.3)[126]. Resulting filtered alignment files were first filtered to include only reads that were the primary alignment, with any duplicates and supplementary alignments being removed via samtools (Version 1.7-2)[127]. Reads with a mapping quality below 30 were also removed. Filtered read sets ranged from 25 to 31.5 million reads.

Peak files were then annotated using the R package ChIPseeker (Version 1.22.0)[128] in order to annotate peak location in relation to known genomic features, as well as to generate maps of ChIP peak binding in reference to transcription start sites where peaks were within 3000 base pairs of a TSS. Log2 ChIP-over-input tracks for each alignment file were generated using deepTools bamCompare (Version 3.1.2), filtering out any reads from the Encode blacklist [Encode, mm10, 2014], and then uploaded to a UCSC track hub. Bam files were also submitted to Macs2 (Version 2.1.1) for peak calling, providing input data for each day as control. A q-value filter of 1e-4 was used to filter low-confidence peaks from the resulting Macs2 peak files. A total peak set of all bam files in the first replicate, as well as a merged signal graph track, was created for visualization.

ChIP-seq peaks were overlapped with peaks found in the ATAC-Seq data using the genomicRanges R package (Version 1.38.0)[129] and the enrichment of each overlapping peak, as derived from Macs2, was displayed via a heatmap. Simple correlation tests were performed to identify if the magnitude of enrichment for overlapping peaks was similar in both experiments. These data were then further overlapped

with a previous IRX3-KO RNA-Seq dataset, generated as previously described[14] to highlight the time-dynamic effects of an IRX3 knockout on target regions which featured a gene promoter, an Irx3 ChIP-seq peak, and a region of openness as defined by our ATAC-Seq experiment. Gene ontology enrichment analysis was performed on sets of peaks whose genomic location and annotation overlapped in different experiments via both the Panther[130] and the Reactome[131] pathway databases.

**Sample preparations and chromatin immunoprecipitation for SUMO2/3 ChIP-seq.** ME3 control and IRX3-KO cells were differentiated towards adipocytes as described above in 10 cm dishes. Three dishes were pooled per biological replicate, and two biological replicates were collected on days −1 and 1 for each cell type. Cells were fixed as above, except fixation was performed for 8 min, and 20 nM N-ethyl maleimide (NEM) was added to the Igepal washing steps. In the second wash step, 1X cOmplete ULTRA protease inhibitor (Roche) cocktail was added instead of PMSF. The crosslinked cells were centrifuged at 1250 x $g$ for 3 min at 4 °C and the cell pellet was flash-frozen in liquid nitrogen before storage at −80 °C until chromatin isolation.

The chromatin isolation and immunoprecipitation was adapted from[50]. Frozen cells were thawed on ice and lysed in lysis buffer A (5 mM PIPES pH 7.5, 85 mM KCl, 0.5% NP-40, 20 mM NEM, and 1x cOmplete EDTA-free protease inhibitor cocktail (Roche)) and incubated at 4 °C for 10 min with rotation. Nuclei were centrifuged (1500 rpm for 10 min at 4 °C) and resuspended in lysis buffer B (50 mM Tris-HCl pH 7.5, 1% SDS, 10 mM EDTA, 20 mM NEM and protease inhibitor cocktail) and incubated at 4 °C for 2 h. Nuclear extracts were sonicated for 15 cycles (30 s on/30 s off) at 4 °C using a Bioruptor Pico sonicator (Diagenode). After sonication, lysates were centrifuged at 14,000 rpm for 10 min at 4 °C. Protein concentration was assessed using the Bradford assay, and 200 µg of chromatin were used for each immunoprecipitation. Input samples (12.5 µg) were saved. Samples were diluted 10-fold in immunoprecipitation buffer (1.1% Tri- ton X100, 50 mM Tris-HCl, pH 7.5, 167 mM NaCl, 5 mM NEM, 1 mM EDTA, 0.01% SDS, and protease inhibitor cocktail). Immunoprecipitations were carried out overnight at 4 °C with 4 µg SUMO2/3 antibody (ab3742, Abcam) and protein A and G Dynabeads (Thermo Fisher Scientific). Beads were then washed 2 times in low-salt buffer (50 mM Tris-HCl, pH 7.5, 150 mM NaCl, 1% Triton X100, 0.1% SDS, 1 mM EDTA), 2 times in high-salt buffer (50 mM Tris-HCl, pH 7.5, 500 mM NaCl, 1% Triton X100, 0.1% SDS, 1 mM EDTA), 2 times in LiCl buffer (20 mM Tris-HCl, pH 7.5, 250 mM LiCl, 1% NP-40, 1% deoxy- cholic acid, 1 mM EDTA) and in TE buffer (10 mM Tris- HCl, pH 7.5, 0.2% Tween20, 1 mM EDTA). Elution was done in two times using 50 µl of 100 mM NaHCO3, 1% SDS at 65 °C for 10 min under agitation. Chromatin crosslinking was reversed at 65 °C for 5 h with 280 mM NaCl and 0.2 µg/ml DNase-free RNase (Roche). Proteins were digested using 0.2 µg/ml of Proteinase K (Roche) for 1 h at 65 °C. DNA from immunoprecipitations and inputs were purified using the Qiagen PCR purification kit. DNA concentration was assessed using a Qubit device (Invitrogen) before preparing sequencing libraries. Sequencing was performed by the GenomEast platform, IGBMC.

**SUMO2/3 ChIP-seq library preparation and sequencing.** ChIP DNA samples were purified using Agencourt AMPure XP beads (Beckman Coulter) and quantified with the Qubit (Invitrogen). ChIPseq libraries were prepared from 10 ng of double-stranded purified DNA using the MicroPlex Library Preparation kit v2 (Diagenode s.a., Seraing, Belgium), according to manufacturer's instructions. In the first step, the DNA was repaired and yielded molecules with blunt ends. In the next step, stem-loop adaptors with blocked 5-prime ends were ligated to the 5-prime end of the genomic DNA, leaving a nick at the 3-prime end. The adaptors cannot ligate to each other and do not have single-strand

tails, avoiding non-specific background. In the final step, the 3 prime ends of the genomic DNA were extended to complete library synthesis, and Illumina-compatible indexes were added through a PCR amplification (7 cycles). Amplified libraries were purified and size-selected using Agencourt AMPure XP beads (Beckman Coulter) to remove unincorporated primers and other reagents. Libraries were sequenced on an Illumina NextSeq 2000 sequencer as paired-end 50 base reads. Image analysis and base calling were performed using RTA version 2.7.7 and BCL Convert version 3.8.4.

**SUMO2/3 ChIP-seq data analysis.** Reads were mapped to mouse genome (mm10) using the ENCODE ChIP-seq pipeline (v1.6.1) that uses bowtie2 (v2.3.4.3)[126] with default parameters to align reads. Then, samtools view (v1.9)[127] was used to remove unmapped, low quality (MAQ < 30) and duplicate reads with the following parameters "-F 1804 -q 30" and "-F 1804". Reads in ENCODE blacklisted regions[132] were removed. Then, SUMO peak calling was performed with ssp v1.15.5[133] according to the ENCODE ChIP-seq pipeline v1.6.1. Reproducible peaks were kept after the IDR analysis was run (optimal IDR sets of peaks were kept). Peaks were annotated relative to genomic features using Homer (v4.11)[134]. Normalization and PCA were performed as described before[135,136]. Briefly, normalization was performed against sequencing depth and input DNA. Differential regions were detected using DESeq2 and $p$-values were adjusted for multiple testing using the Benjamini and Hochberg method. Functional enrichment analysis was performed on differential regions using ClusterProfiler (V4.10.1). Heatmaps were generated using Deeptools computeMatrix (v3.5.4) and plotHeatmap v3.5.4. Mean profiles were generated using plotProfile (v3.5.4), and upset plots were made using ComplexHeatmap (v2.18.0). Known or de novo TF motifs were identified using HOMER findMotifsGenome.pl with default parameters.

## ChIP-qPCR

ME3 control and IRX3-KO cells ($n = 2$) were differentiated until day 1 before fixation by formaldehyde and isolation of chromatin as described above for SUMO2/3 ChIP-seq. Immunoprecipitation was performed as described above, except using 6 µg anti-IRX3 antibody (ab25703, Abcam) and 300 µg chromatin. After DNA purification, specific DNA regions were amplified by RT-qPCR using the HOT FIREPol® EvaGreen® qPCR Supermix (Solis Biodyne) on a LightCycler 96 instrument (Roche Diagnostics). RT-qPCR signals from immuno-precipitations were normalized to input samples (% of input). Then the ratio of percent input in control over IRX3-KO was calculated. Primer sequences are shown in Table 1.

## ATAC-seq preparation

**Preadipocytes from iWAT and gWAT.** ATAC-seq was performed by adapting the protocol from ref. 137. Differentiating cells were lysed directly in cell culture plates in triplicates. Lysis buffer was added directly onto cells grown in a 12-well plate. Plates were incubated on ice for 10 min until cells were permeabilized and the nuclei released. Lysis buffer was gently pipetted up and down to wash nuclei off the well and transferred into a chilled 1.5 ml tube to create crude nuclei. Nuclei were spun down at 600 x $g$ for 10 min at 4 °C, nuclei pellets were then re-suspended in 40 µl Tagmentation DNA (TD) Buffer. Transposition reaction was performed as previously described[137]. All tagmented DNA was PCR amplified for 11-13 cycles. Quality was assessed using a DNA1000 Chip (Applied Biosystems) and run on a Bioanalyzer (Applied Biosystems). The profiles showed that all libraries had a mean fragment size of ~190 bp and characteristic nucleosome patterning, indicating good quality of the libraries. Libraries were sequenced at the Wellcome Trust Centre for Human Genetics in Oxford on a HiSeq4000 Illumina generating 50 million reads/sample, 75 bp paired end. To reduce bias due to PCR amplification of libraries, duplicate reads were removed.

**Table 1 | Primer sequences**

| Target gene | Application | Forward primer | Reverse primer |
|---|---|---|---|
| Senp1 | ChIP-qPCR | 5′-GGAACCGGAACCTTCTTTAC-3′ | 5′-CTCGCTCTTTGACCTCTTGTC-3′ |
| Senp5 | ChIP-qPCR | 5′-GCTCTGGACTACAGTTTAAG-3′ | 5′-TCCAGATGAAGGCTCTAC-3′ |
| Senp6 | ChIP-qPCR | 5′-GCTCCCACCTCCATATT-3′ | 5′-CCAAGCTTAGGTCTGAAAC-3′ |
| Ranbp2 | ChIP-qPCR | 5′-CGGATTTATTAGGGCACTG-3′ | 5′-AGACGAACTCCCTTTCC-3′ |
| Sae1 | ChIP-qPCR | 5′-TCCAAGGACACTCTCAATAC-3′ | 5′-GGGTTTCCTAGTGCTTCA-3′ |
| Uba2 | ChIP-qPCR | 5′-CTGAGTCCTGGGATTACTT-3′ | 5′-CCGAGGTCTTTAGAAGGT-3′ |
| Ubc9 | ChIP-qPCR | 5′-AAACTCCTCTTCGGGATG-3′ | 5′-TGACCCTTGGGAATCTATG-3′ |
| Cbx7 | ChIP-qPCR | 5′-CTCAGGATCTCTAGTCCTATC-3′ | 5′-CATCCTGTTCCTCACTTTC-3′ |
| Zmiz1 | ChIP-qPCR | 5′-GGCCCTTTGGAAATAGAATAG-3′ | 5′-AAGGAACTCGGCGTTAG-3′ |
| Ctr1 (Adipoq) | ChIP-qPCR | 5′-CTACTTGGCTTCCCAGACCC-3′ | 5′-ACCCAGTCAAGGCCAATAGC-3′ |
| Ctr2 (gene desert) | ChIP-qPCR | 5′-TACACATTCAGGGTGGAAGCAA-3′ | 5′-TGTCTCTCAGCACTATCTCCCCAA-3′ |
| Pparg2 | qPCR | 5′-TTATAGCTGTCATTATTCTCAGTGGAG-3′ | 5′-GACTCTGGGTGATTCAGCTTG-3′ |
| Cebpa | qPCR | 5′-GGTTTAGGGGATGTTTGGGTTT-3′ | 5′-GCCCACTTCATTTCATTGGT-3′ |
| Fabp4 | qPCR | 5′-ATCACCGCAGACGACAGG-3′ | 5′-TCATAACACATTCCACCACCA-3′ |
| Plin2 | qPCR | 5′-CTCCACTCCACTGTCCACCT-3′ | 5′-ACTCCACCCACGAGACATAGA-3′ |
| Adipoq | qPCR | 5′-ACTCCTGGAGAGAAGGGAGA-3′ | 5′-CGCTGAGCGATACACATAAGC-3′ |
| Lipe (Hsl) | qPCR | 5′-CACAAAGGCTGCTTCTACGG-3′ | 5′-GGAACTGGCGGTCACACT-3′ |
| Pnpla2 (Atgl) | qPCR | 5′-CAACGCCACTCACATCTACGG-3′ | 5′-GGACACCTCAATAATGTTGGCAC-3′ |
| Cd36 | qPCR | 5′-TTGAAAAGTCTCGGACATTGAG-3′ | 5′-TCAGATCCGAACACAGCGTA-3′ |
| Ppara | qPCR | 5′-CGGGAAAGACCAGCAACA-3′ | 5′-GAATCGGACCTCTGCCTCT-3′ |
| Ucp1 | qPCR | 5′-GGGCATTCAGAGGCAAATCAG-3′ | 5′-TTTCCGAGAGAGGCAGGTGTTT-3′ |
| Rps13 | qPCR | 5′-CAGGTCCGTTTTGTGACTG-3′ | 5′-AGCATCCTTATCCTTTCTGTT-3′ |
| Tbp | qPCR | 5′-GGGGAGCTGTGATGTGAAGT-3′ | 5′-CCAGGAAATAATTCTGGCTCA-3′ |
| Sp7 | qPCR | 5′-GGCGTCCTCTCTGCTTGAG-3′ | 5′-TTTGTGCCTCCTTTCCCCAG-3′ |
| Alpl | qPCR | 5′-TGCCCTGAAACTCCAAAAGC-3′ | 5′-GACGCCCATACCATCTCCC-3′ |
| Col1a1 | qPCR | 5′-TCTGACTGGAAGAGCGGAGAG-3′ | 5′-TGAGTAGGGAACACACAGGTC-3′ |
| Spp1 | qPCR | 5′-GAGGAAACCAGCCAAGGACT-3′ | 5′-CGGGAGGGAGGAGGCAAT-3′ |

**WT ME3 cells**. ME3 cells were collected for open chromatin profiling (ATAC-seq) by trypsination on days 0, 1, and 7 of differentiation in two independent experiments. For each sample, technical duplicates of 100,000 cells were frozen at a slow rate (−1°/minute) in 5% DMSO at −80 °C in CoolCell freezing containers (BioCision, San Rafael, CA, USA). Open chromatin was profiled through a commercial service (Active Motif, Carlsbad, CA, USA) based on a method previously described[138]. In brief, chromatin was isolated from the cells followed by Tn5 transposase-mediated insertion of sequencing primers into open chromatin regions. After library preparation, high-throughput sequencing (Illumina) was performed and reads were then aligned to the mm10 genome using Bowtie2 (Version 2.3.4.3) using the "very-sensitive" setting.

**Control and IRX3-KO ME3 cells**. Cells were differentiated for 0, 1, and 7 days in triplicates, pelleted, and snap-frozen in liquid nitrogen. After thawing, cells were treated with DNase to remove DNA released from dead cells before lysis in ice-cold ATAC seq lysis buffer from the ATAC-Seq Kit (Active Motif, Carlsbad, CA, USA) for 2 min. Crude nuclei extracts were spun down at 600xg for 10 min at 4 °C and nuclei were resuspended in 40 µL 1X Tagmentation buffer. The integrity of the nuclei was verified under a microscope, counted, and 50,000 nuclei were transferred pr sample for tagmentation. Tagmentation and indexing were performed according to the instructions of the ATAC-seq Kit (Active Motif). Library quality control was performed with a D5000 ScreenTape assay and sequenced at the Genomics Core Facility (GCF) at the University of Bergen using the HiSeq4000 Illumina and 75 bp paired end sequencing.

**ATAC-seq data analysis**
Reads were first filtered for mapping quality ≥ 30 (samtools Version 1.7-2) and duplicated reads, as well as reads mapping to the mitochondria, were subsequently removed (picardTools Version 2.8.1.1). Peak calling was performed via Macs2 (Version 2.1.1). Peak sets were first filtered using blacklisted regions from the ENCODE project[139], then converted into a simple annotation format for submission to the Rsubread featureCounts (Version 1.6.0) package. Using this consensus peak set as an annotation guide, each unmerged bam file was submitted to featureCounts and the subsequent count matrix was input into RStudio for peak annotation and visualization.

Time-course analysis for each peak was then performed via the R package maSigPro (Version 1.56.0)[140]. A regression fit for each peak was performed, and the p-value associated with the F-statistic of each model was calculated to uncover peaks which change dependent on experimental group. Peaks with an adjusted p-value less than 0.1 were then selected, and stepwise regression was performed to select significant variables for each gene. Peaks which were found to change over time were then submitted to hierarchical clustering and grouped into six clusters with differing time-dependencies and visualized via ggplot2 using a glm smoothing method with polynomial regression.

To detect differential accessibility between control and IRX3-KO samples, individual peaks were first identified using Macs2 as described above, and sequencing reads within each individual peak were then counted using featureCounts[141] and DESeq2[136].

**RNA isolation, cDNA synthesis, and real-time qPCR analysis**
RNA was isolated either alone using the RNeasy mini kit (Qiagen, Hilden, Germany), the RNA/Protein Purification Plus kit (Norgen, Thorold, ON, Canada), or the Maxwell 16 LEV simplyRNA purification kit (Promega, Madison, WI, USA) according to the manufacturers' instructions. RNA quality control, cDNA preparation and qPCR were performed as previously described[14], using the delta-delta Ct method relative to

reference genes *Rps13* or *Hprt*. Primers were designed using either the Universal ProbeLibrary Assay Design Center (Roche, Basel, Switzerland) or Primer-BLAST softwares[142]. Primer sequences are shown in Table 1.

## RNA-seq

ME3 control and IRX3-KO cells were differentiated in under adipogenic conditions, as described above, and treated with either DMSO or 0.5 μM ML-792 from day −2 to −1, day 0 to 1, or continuously from day 0 to 9 (*n* = 3 replicate wells per condition). Cells were harvested by trypsination and snap-frozen in liquid $N_2$ on days −1, 1 and 9. RNA was isolated using phenol/guanidine-based lysis followed by chloroform extraction and silica-based spin column purification of the aqueous phase according to instructions of the RNeasy Lipid Tissue kit (Qiagen). RNA quantity and integrity were assessed by Qubit fluorometric quantification. Libraries were prepared using the Illumina Stranded mRNA Ligation kit and quantified using the Illumina MiSeq Nano sequencing Library QC kit on an Agilent TapeStation. Sequencing was performed on a NovaSeq 6000 sequencer using a NovaSeq S2 flowcell and 2x100bp paired-end reads. Reads were mapped to mouse genome (mm10) using Hisat2 and annotated using featureCounts. Normalization and differential expression analyses were performed using DESeq2.

## Protein isolation and Western Blotting

For isolation of total proteins with intact SUMOylome, fresh cells were washed in PBS and lysed directly in the culture plate with SUMO lysis buffer (2% SDS in PBS, pH 7.4, supplemented with 20 mM NEM and 2X cOmplete ULTRA protease inhibitor cocktail). Lysis was performed in SDS with NEM to immediately block and denature desumoylases, and at room temperature (RT) to avoid precipitation of SDS. Cells were sonicated at RT to reduce viscosity and boiled at 95 °C for 5 min to facilitate protein unfolding and alkylation of free −SH groups through formation of stable thioether bonds. Samples were quantified using the SDS-compatible DC-kit (Biorad), followed by addition SDS-sample buffer containing excess DTT and boiled again at 95 °C for 5 min to reduce disulfide bonds (does not affect the irreversible thioether bonds resulting from NEM treatment).

For Western Blotting of proteins where preservation of SUMO was not relevant, cells were lysed in 1X RIPA buffer (Millipore) supplemented with 1X cOmplete ULTRA protease inhibitor cocktail (Roche). Cell lysates were sonicated at 4 °C to reduce viscosity, and cell debris removed by centrifugation at 13,000 x *g*. Samples were quantified as described above, and 20 μg lysate of each sample was loaded on each gel. All samples were analyzed by WB using 20 μg normalized lysate per sample. The following primary antibodies were used for WB: anti-SUMO2/3 (ab3742, Abcam), anti-SUMO1 (ab32058, Abcam), anti-AM (AB_2793779, Active Motif), anti-ACTB (ab6276, Abcam), anti-GAPDH (MAB374, Millipore), anti-VINCULIN (ab18058, Abcam), (anti-SENP1 (ab236094, Abcam), anti-SENP5 (19529-1-AP, Proteintech), anti-UBA2 (8688S, Cell signaling), anti-UBC9 (4786S, Cell Signaling) and anti-SAE1 (ab185949, Abcam). All primary antibodies were used at 1:1000 dilution except anti-SENP5, anti-SAE1, and anti-VINCULIN, which were used at a 1:2000 dilution. The following secondary antibodies were used: anti-mouse (554002, BD Biosciences) and anti-rabbit (31460, Invitrogen) at 1:7500 or 1:10000 dilutions. Uncropped images are provided in the Source Data file.

## Plasmids

The *Irx3*-AM construct was made by in-fusion cloning. Briefly, mouse *Irx3* (NM_008393) was amplified by PCR from the donor vector pCMV6-entry-*Irx3*-myc-DDK (cat. no MR208149, Origene, Rockville, MD, USA) with the following forward 5′-*CGATCTAAGTAAGCTT*CAC-CATGTCCTTCCCCCAGCTCG-3′ and reverse 5′-*GATCTTGGCAAAGCT*-TAGACGAGGAGAGAGCTGATAAGACC-3′ primers. 5′ extensions (italic)

matching HindIII digested pAM_1C empty destination vector (cat. no 53023, Active Motif) was added to each primer. A consensus Kozak sequence (underlined) was added to the forward primer just upstream of the start codon. The insert was cloned into the pAM_1C vector using the In-Fusion snap assembly master mix according to manufacturer's instructions (Takara Bio, Kyoto, Shimogyō-ku, Japan). The purified plasmid was verified by Sanger sequencing.

## Luciferase reporter assays

ME3 cells were seeded in 96-well plates at 7500 cells/well and grown for 24 h before being transfected with 20 ng of the pGL3-TK-PPRE-firefly-luc reporter construct, 1 ng of the constitutively expressed pRL-SV40-Renilla normalizator construct and 40 ng each of pCDNA-*Pparg2* and/or pCDNA-*Ppargc1a*. The pCDNA-empty vector was used to keep the total plasmid levels constant at 101 ng/well. The plasmids were diluted in OptiMEM and cells were transfected using TransIT-LT1 at a DNA:reagent ratio of 1:5. One day after transfection, the culture medium was replaced with fresh medium containing either DMSO or 0.5 μM ML-792 (MedChemExpress, Monmouth Junction, NJ, USA) in combination with either DMSO or 10 μM rosi. Two days after transfection, the cells were lysed and analyzed using the twinlite kit (PerkinElmer, Waltham, MA, USA) and the FLUOstar OPTIMA luminometer (BMG Labtech, Ortenberg, Germany) according to manufacturers' instructions.

## Statistics & Reproducibility

Animal cohort size was estimated from previous experiments using the same tests and power calculations using G*Power 2. The investigators were not blinded to allocation during experiments and outcome assessment. Outliers were identified by the ROUT test (Q = 1%), but were excluded only when associated with an obvious technical error. Omics data were analyzed in R as described above. Remaining data was analyzed and graphed using GraphPad Prism (9.2.0). Box plots show the median and interquartile range with whiskers indicating the minimum and maximum values; bar graphs show the mean values, with error bars representing SD. Statistically significant differences between group means were investigated using two-sided Student's unpaired t-test, one-way ANOVA, or two-way ANOVA with Holm-Sidak correction for multiple testing as indicated. Data were tested for normality and homogeneity of variance, and transformed, if necessary, prior to statistical analysis to conform with the assumptions of the tests.

## Reporting summary

Further information on research design is available in the Nature Portfolio Reporting Summary linked to this article.

# Data availability

Source data are provided with this paper. Sequencing data generated in this study are available at www.ebi.ac.uk/arrayexpress under accession codes E-MTAB-13524 (ChIP-seq IRX3 in gWAT and iWAT), E-MTAB-13540 (ATAC-seq in gWAT and iWAT), E-MTAB-13520 (ATAC-seq in WT ME3 cells), E-MTAB-13525 (ATAC-seq in control and IRX3-KO ME3 cells) and E-MTAB-14723 (RNA-seq on days −1, 1 and 9 in ME3 control and IRX3-KO treated with ML-792), and at https://www.ncbi.nlm.nih.gov/geo/ under accession code GSE278972 (SUMO2/3 ChIP-seq in ME3 control and IRX3-KO cells). Sequencing data used in this study that was derived from[14] are available at www.ebi.ac.uk/arrayexpress under accession code E-MTAB-8209 (RNA-seq in ME3 control and IRX3-KO on days 1 and 7). Source data are provided with this paper.

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

## Acknowledgements

We thank the engineers at the Harwell Institute, Oxford, UK, and the Hormone Laboratory at Haukeland University Hospital, Bergen, Norway, particularly Michelle Simon, Carol Cook, Margit Solsvik, and Linn Skartveit, for excellent technical assistance. We also thank Anagha Madhusudan Joshi-Michoel for valuable input in interpreting the ChIP-seq data. The WT ME3 cells were kind gifts from Karsten Kristiansen. We thank Rita Holdhus, Dr. Carla P. D. Fernandes, and Aashish Srivastava at the Genomics Core Facility (GCF) at the University of Bergen, which is a part of the NorSeq consortium, for providing services on RNA- and ATAC-sequencing. We thank Dr. Stéphanie Le Gras at the GenomEast platform, which is a part of the Institute of Genetics and Molecular and Cellular Biology (IGBMC) and a member of the 'France Génomique' consortium (ANR-10-INBS-0009), for performing sequencing and parts of the analysis related to the SUMO ChIP-seq data. Figure 3m is adapted from Bjune, J. (2025) https://BioRender.com/1m0hzx6. This work is supported by the Mohn Research Center for Diabetes Precision Medicine, Trond Mohn Research Foundation (TMS2022TMT01, PRN, GM; TMS2021TM08, KM), the Western Norway Regional Health Authority (F-12122, PP, GM), the Research Council of Norway (263124/F20, 245979/F50, GCF; 301268 and 353961, PC), the Norwegian Diabetes Association, Meltzerfondet (JIB), University of Bergen (240413), the UK Medical Research Council (MC_U142661184, RDC), the European Research Council (293574), the Novo Nordisk Foundation (NNF18OC0054741, NNF21SA0072102, NIDDK UM1 557 DK126185 and NIDDK DK102173, MC), Bergen Research Foundation (BFS) (BFS2017TMT04, BFS2017TMT08, and BFS2018TMT10, GCF, KM). PC also received institutional funding from the Dept. of Biosciences at the University of Oslo and UNIFOR.

## Author contributions

J.I.B, S.L, M.C, R.D.C., G.M., and S.N.D. conceived the study. J.I.B., S.L., S.Y., N.A., P.C., and S.N.D. designed the experiments with input from K.M., M.C., G.M., R.D.C., and P.R.N. J.I.B. performed most of the experiments pertaining to ME3 cells and analyzed the data from all experiments. S.L. performed the animal experiments and experiments pertaining to isolated cells from iWAT and gWAT, including ChIP-seq and ATAC-seq. S.Y. and N.A. conducted the osteogenic cell culture experiments. S.Y. performed the 3D culture experiments. P.M.C.N. and P.C. carried out ChIP-seq experiments on SUMO2/3, ChIP-qPCRs on IRX3, and RNA-seq sample preparation in ME3 cells. L.L.A., X.Z., and P.P.S. performed the bioinformatics and J.I.B., P.M.C.N., S.Y., and P.C. conducted additional statistical analyses. J.I.B., L.L.A., and X.Z. handled the data curation. J.I.B. created the figures with contributions of raw figure elements from L.L.A., X.Z., S.Y., P.C., and P.P.S. J.I.B. wrote the manuscript under supervision of P.C., G.M., and S.N.D. Additional supervisory roles were held by K.M., P.R.N., M.C., R.D.C., P.C., G.M., and S.N.D. The project administration was maintained by J.I.B., P.C., G.M., and S.N.D. Financial support was provided by K.M., P.R.N., M.C., R.D.C., P.C., G.M., and S.N.D. All authors contributed to data interpretations, critically reviewed the manuscript, and approved the final version.

## Funding

## Competing interests

The authors declare no competing interests.
