## [Peer Review file · Nature Communications]

IRX3 controls a SUMOylation-dependent differentiation switch in adipocyte precursor cells

Corresponding Author: Professor Simon Dankel

Version 0:

Reviewer comments:

Reviewer #1

(Remarks to the Author)

This manuscript investigates the role played by the *lrx3* protein in adipocyte differentiation. The authors present evidence demonstrating its interaction with DNA, influencing chromatin accessibility, and orchestrating changes in gene expression throughout the differentiation process. Notably, the authors propose a correlation between these effects and the capacity of *lrx3* to suppress genes associated with the SUMO pathway, resulting in a widespread reduction in post-translational modification levels. The study reveals that *lrx3* knockout (KO) leads to a diminished adipogenic differentiation while concurrently promoting osteogenesis.

In essence, this manuscript contributes new findings regarding the chromatin-related functions of *lrx3* and sheds light on its implications in the intricate process of adipogenic differentiation. However, the claims and conclusions are not always supported by the data. In addition, many of the data used in the manuscript were previously published by the authors, often in different cellular model. Their reanalysis is valid but the use of data coming from biological system different from those studied here limits the relevance and the strength of the conclusions. In general, figures lack sufficient explanations of their content and the conclusions are not supported by the data, at least given the provided information. The number of biological replicates should be indicated for every experiment in the figure legend. Statistical analysis are not always provided. For example, for all ATAC-Seq experiments, I could not find the information on how many biological replicates were performed and I had no access to the raw data on ArrayExpress as the identifiers given don't match to any recorded study. At least 2, ideally 3, replicates should be performed.

Figure 1: it is stated that there is a dramatic increase in *lrx3* binding events following initiation of differentiation. When looking at the profiles provided in Fig1E, I don't see much differences in the peak intensities between iWAT (day -1) and iWAT (day 1). For *Uba2*, the peaks are not considered as peaks in the iWAT (day -1) while it is for the iWAT (day 1) although I don't see a difference on the profile. Are these genes not representative of the peaks gained? Other profiles representative of genes with peaks appearing in the iWAT (day1) condition should be presented. A heatmap with all the peaks with iWAT (day -1) as a reference should be provided. A metaprofile for *lrx3* peaks around TSS for all conditions should also be provided.

For the gWAT, only the Day 1 is presented, which makes it difficult to state that there are differences as the basal status of *lrx3* binding might be different to iWAT. Moreover, only one replicate was performed for this condition. This questions the robustness of the results and of the conclusions

Figure 1: The enrichment of SUMO pathway genes is not so obvious. There are 3 genes of the SUMO pathway bound by *lrx3* in iWAT out of 142 peaks. Moreover, in Supplementary File 1, I could not find any GO term linked to SUMO.

Figure 1C: the detailed results of the reactome analysis should be given in the supplementary table.

Figure S1A: The total number of ATAC seq peaks decreases upon differentiation, suggesting that the accessibility of chromatin globally decreases. From the text, we could indeed assume that it increases.

Figure S1B: The numbers of the clusters should be indicated (assuming they are the same as those in the rest of the figure).

Figure S1D: specific GO terms were selected but there are many others in the list, which are more enriched than those

indicated. The sentence line 166 should be modified to take this into account. Moreover the last sentence is, at least at this stage, an overstatement.

Figure S1E: it is not clear what can be concluded from this figure. It seems that there is indeed no strong correlation between the opening of chromatin and *Irx3* binding.

Figure S2: same remarks as SupFig1: the clusters should be indicated on B.

Figure S2I: the ATAC-Seq pic presented on KDM3a seems to decrease with the differentiation. This contradicts the message that *Irx3*-bound peaks get more accessible during differentiation.

Figure 2: based on the ATAC-Seq results, it is assumed that *Irx3* has the same binding profile in WAT and ME3 cells. I don't think this is necessarily true.

Figure 2A: how are DEG selected? Is there a fold change selection? This should be indicated in the figure legend

Figure 2B: *Irx3* binding sites are equally abundant on genes that are up- and down-regulated upon differentiation. This makes it difficult to conclude on the role of *Irx3* in their expression and suggests that the binding of *Irx3* could be independent from its effect on gene expression

Figure 3: the effect of *Irx3* on SUMO pathway genes binding and expression is not really convincing. I don't understand the sentence line 232-233: the authors state that the SUMO pathway genes are enriched in the « differentially expressed *Irx3*-bound genes » but this is not significant. Then I would conclude they are not enriched in the differentially expressed genes if it is not significant. This contradicts the subsequent claim that these genes are up- or down-regulated (line 240-241) as well as the sentence in the abstract (line 56-57). Considering the low level of changes, I am anyway not convinced this translates in significant changes in protein abundance. This should be checked by immunoblotting, at least for the most dysregulated SUMO pathway proteins.

Figure 3C-F: the increase in SUMOylation in the KO is not really convincing. Whilst in Fig3C, I can see an increase, this is not that clear in Fig3E. The experiments in Fig3C, F should be performed at least 3 times, which seems not the case. In Fig3F, no statistical significance is provided, which questions the validity of the presented results. The levels of SUMOylation should also be analyzed in iWAT and gWAT upon induction of differentiation

There is no Supplementary Figure S4

Figure 4: Although the effect of *Irx3*-KO on SUMOylation is not really convincing (Figure 3), the effect of ML792 on bodipy staining is impressive. The effect on the genes is also very clear. It might therefore be that *Irx3* primarily affects the SUMOylation on the chromatin. It would thus be interesting to see if there are changes in the level of SUMO on the adipogenic genes in wt vs *Irx3* KO cells by ChIP-Seq or ChIP-qPCR.

Figure 5B: it is stated that ML792 increases PPAR γ activity 2-4 fold. It is not really obvious from the graph provided. Moreover, no statistical analysis is provided on the graph. I agree that the effect becomes clear in the presence of Rosiglitazone and PGC1 α but the ligand is not present in the context of the differentiation model, suggesting that the desumoylation of PPAR might not be the main driver of ML792 induced adipogenesis in *Irx3* KO cells

Figure 6: I don't think that the comparison of the RNA-Seq in *Irx3*-KO in ME3 to the SLAM-Seq in 3T3L1 is very relevant. Moreover, as around 1/3 of the protein-coding genes expressed in mouse genome are changing expression in *Irf3* KO, it is somehow normal that there is some overlap with ML792 regulated genes. It would be more informative to perform RNA-Seq (or SLAM seq) in ME3 wt and *Irf3* KO $-/+$ ML792

Figure 6: *Irx3* KO increases SUMOylation so should have an opposite effect to ML792, which decreases SUMOylation. However, when looking at the overlapping DEG, this is not the case. Genes up-regulated in *Irx3* KO are not necessarily down-regulated in ML792 and vice versa. This questions the relevance of the conclusions.

Reviewer #2

(Remarks to the Author)

In this manuscript, Bjune et al demonstrated that *IRX3*, known to be a causal gene of obesity-associated FTO variants, plays a critical epigenetic role in the regulating adipogenic and osteogenic potentials, particularly via sumoylation. To reach this conclusion, the authors utilized multi-omic approaches, including ChIP-seq, ATAC-seq and RNA-seq, in combination with cell biological experiments. The manuscript is interesting, but below are points that I hope will strengthen the study.

1. One of the novel results in this manuscript is *Irx3* ChIP-seq data in preadipocytes, as its direct targets have not been well studied. However, the manuscript lacks clarity regarding the *Irx3* ChIP-seq experiment. The details about ChIP antibodies against *Irx3* and the control antibodies were not described in the methods section. Please provide information about the antibody (cat#, log#, concentration). Especially, a clear validation of the *Irx3* antibody specificity will be required and shown (e.g. IP-WB, cross-reactivity testing etc). Although IgG is conventionally used as a control in ChIP-seq experiments, I suggest a more specific approach given that the authors have *Irx3*KO ME3 cells. It would be strongly suggested to perform *Irx3*-ChIP using the same anti-*Irx3* antibody in

both WT and *Irx3*-KO ME3 cells, and then compare the data to the IgG control ChIP-seq data. This method would help in generating *Irx3* ChIP-seq data free from non-specific binding.

Additionally, it appears that motif analysis for *Irx3* binding regions was not performed. Did you observe any putative *Irx3* binding sequence?

2. While *Irx3*-ChIP seq was performed in primary preadipocytes (isolated from SVF), most of the other experiments were performed in ME3 cells. Although ME3 cells are known to be useful for studying beige adipocytes, they are fundamentally mouse embryo fibroblasts, which are developmentally different and more potent than fate-determined preadipocytes. Consequently, the observed phenotypes, such as decreased adipogenic potential and increased osteogenic potential, may reflect developmental changes rather than specific adipocyte and adipogenesis mechanisms. This is supported by the ATAC-seq data at day 0, which showed higher open chromatin in *Bmp2* and *Wnt* in *Irx3*-KO ME3 cells, indicating the developmental role of *Irx3* in the fate determination of mesenchymal cells. Conducting *Irx3* ChIP-seq in ME3 cells would further strengthen these findings.

Furthermore, the authors' main claim that *Irx3* controls a sumoylation-dependent epigenetic switch between adipogenesis and osteogenesis seems less relevant when considering cell type variability. If this claim is relevant, it is important to examine whether *Irx3*KO primary preadipocytes or 3T3-L1 cells also demonstrate a similar propensity to differentiate into the osteogenic pathway. The findings appear to be specifically relevant to ME3 embryonic fibroblast cells, raising questions about the generalizability of the claim to other cell types and in vivo contexts. In vivo, the role of *Irx3* might be more pertinent to bone health or bone marrow dynamics rather than to adipose tissue function or obesity. Therefore, it is crucial for the authors to explore and clarify how the observed mechanisms and findings in cell culture models translate to in vivo and physiological conditions. Investigating this aspect would not only enhance the understanding of *Irx3*'s role in different tissues but also provide a clearer picture of its potential impact on obesity and related metabolic conditions.

3. Due to the reason above, it seems overly speculative to conclude that these findings are directly related to FTO genetic variants-associated obesity. To establish a more concrete connection, in vivo or in vitro experiments should be conducted using preadipocytes from mice lacking *Irx3* in the context of obesity. This approach would provide more relevant data to understand the role of *Irx3* in the context of FTO variant-associated obesity.

4. In Figure 2A, the criteria used for analyzing the RNA-seq data (*Irx3*KO/CTRL) are not clearly defined and appear to be overly permissive. This is evident from the excessively high number of differentially expressed genes (DEGs), exceeding 11,000. Such a large number of DEGs implies statistically significant risk of overlap, potentially up to a 50% chance for any given gene. Additionally, the differential expression levels of sumoylation genes are relatively modest, with a log₂ fold change of less than 0.6. This indicates a change less than 50%, which may not be substantial enough to draw strong conclusion.

Similarly, the logical analysis pipeline focusing on epigenetics and sumoylation through bioinformatic analysis seems unclear. Considering the nature of GO term analysis, which works with lists of genes regardless of the extent of differential expression, the uncertain criteria for fold change and the exceptionally high number of DEGs could generate a GO list with many available functional terms. Therefore, in the GO term and KEGG pathway analysis, the authors should provide the overall list of terms generated and the actual rank of epigenetics and sumoylation-related pathways. This will clarify whether these pathways were identified as high-ranking hits from an unbiased analysis or if they were specifically selected by the authors based on their interests. Such transparency is crucial to assess the relevance and impact of these pathways in the study.

5. There seems to be an inconsistency in the data comparison between different sets of omics data. In Figure 2A, the number of DEG on day 1 (*Irx3*KO/WT ME3) was reported as 11486, while in Figure 6, this number is reduced to 8013. Furthermore, the cutoff criteria for these analyses should be clearly stated. When they are mentioned, as in Figure 6A, the authors have used a fold change of 1.2 (not log₂FC) as the threshold, which could be considered overly lenient for bulk RNA-seq. A more stringent FC threshold is preferred to reduce the likelihood of identifying false positives in such analyses.

6. Figure 4D appears to be confusing as it does not accurately reflect the changes occurring in *Irx3*-KO ME3 cells, potentially leading to misinterpretation. For example, *Fabp4* appeared to be shown as 8-fold higher in *Irx3*-KO cells treated with ML-792. However, Supplementary Figure S5 presents more direct and comparable data. It would be helpful if the authors could ensure that the figures accurately represent the underlying data to prevent any confusion or incorrect interpretation of the results.

7. Figures 3C and 3E present what appears to be overlapping data regarding *Irx3* KO ME3 cells, yet the results showed inconsistency, particularly for day 0. This discrepancy raises questions about the suitability of the methods in these experiments. Additionally, it is unclear whether the treatment with ML-792 leads to a reduction in the global SUMOylome pattern in *Irx3*KO cells. Clarification on this point would be valuable, as it could significantly impact the interpretation of the ML-792 treatment effects in the context of *Irx3* knockout.

8. I recommend the authors specify the cell types used in each experiment when describing the results. Since various cell lines, including ME3, 3T3-L1, and primary preadipocytes, were utilized and compared within the study, explicitly identifying the cell type in each instance will aid in avoiding confusion and will enhance the clarity and accuracy of the interpretations made from the data.

(Remarks to the Author)

In the present article by Bjune et al. revealed a newfound connection between IRX3 and the sumoylation pathway. IRX3 was identified as a direct controller of the SUMO machinery and chromatin remodelers. Inhibiting sumoylation was demonstrated to rescue adipogenesis in cells where IRX3 was knocked out, highlighting IRX3's inhibitory role on sumoylation during the early stages of cell differentiation. The study also illustrated that IRX3 and sumoylation have overlapping target genes, particularly those associated with the development of adipocytes and osteoblasts, indicating a complex interplay between these elements in determining the fate of cells.

From a functional perspective, the work by June et al. uncovered that cells lacking IRX3 exhibited a modified epigenetic landscape that favored osteogenic differentiation over adipogenesis. This shift towards enhanced osteogenesis was linked to a more permissive chromatin state in the absence of IRX3. Crucially, the study demonstrated that the promotion of osteogenesis resulting from IRX3 ablation was, in part, mediated by heightened levels of sumoylation. Inhibiting sumoylation was shown to counteract the pro-osteogenic effects of IRX3 ablation. These findings significantly contribute to a deeper comprehension of the intricate regulatory network involving IRX3, sumoylation, and their influence on the differentiation of adipocytes and osteoblasts.

The manuscript is well-written, follows a logic narrative and presents interesting data supporting their hypothesis. I believe the findings will be of interest of the broad readership of nature communications.

My main criticism is even though their work is partially supported by others findings, as it is highlighted in the discussion, without *in vivo* data using floxed mice that could be targeted with adipose or bone specific cre lines needs to be substantiated. See floxed mice in Gaborit N. et al Development 2012 or Cain CJ et al 2016 Bone Reports. I believe tuning down claims in the discussion and highlighting the need of *in vivo* models to support the claims would be appropriate. Otherwise, authors could consider testing some of their hypothesis *in vivo*.

Conceptually, I believe that based on the readership of nature communications I believe that a broader section on the biological impact the findings could have beyond FTO and obesity could be of interest. The numbers of osteoporosis and other bone defects keep rising. Do authors believe their findings could help epigenetically modify some of those detrimental states. A bit of big picture impact would be appropriate.

Experimentally, in the context of the frame provided I believe the work has been done at the highest standards and statistics are both sufficient and appropriate in all figures. With an *in vitro* frames I believe this manuscript is ready for publication and all logic questions are appropriately answered.

Reviewer #4

(Remarks to the Author)

Bjune and coworkers studied *Irx3* target genes during early adipogenesis in wild-type vs. *Irx3*-KO mouse preadipocytes. They firstly identified > 300 "strong" *Irx3*-binding sites in preadipocytes, mainly at promoter regions, with a strong enrichment of genes linked to SUMOylation, histone modifications and chromatin remodeling. Several genes of the SUMOylation machinery were bound by *Irx3* and differentially expressed in *Irx3*-KO; *Sumo1* and *-3* down-regulated, *Sae1* and *Uba2* (E1), *Ube2i* (E2), *Senp1* as well as *-5* up-regulated.

Western blotting analyses suggested that these alterations in the expression of both forward and reverse reaction-catalyzing SUMOylation machinery genes lead to a modestly increased SUMO2/3 conjugation of cellular proteins in the KO cells. *Irx3* ablation compromised Ppar γ -dependent reporter gene activity and adipogenic differentiation in preadipocytes, which could be partly restored by general SUMOylation inhibitor ML-793. Evidence supporting the notion that SUMOylation is playing a role also in the osteogenesis process induced by IRX3 ablation is provided by experiments using ML-792. The authors conclude that IRX3 acts as a novel upstream regulator of sumoylation, and a potent controller of epigenetic regulators, both directly and indirectly via suppressing global sumoylation levels.

This is a potentially interesting but somewhat incomplete study. It could be considerably strengthened by considering the following points:

- Is the expression of other deSUMOylation enzyme genes altered in the *Irx3*-KO cells?
- Is the expression of PIAS and other SUMO E3 ligase genes affected by IRX3 depletion?
- The enhancing effect of IRX3 depletion on cellular SUMO2/3 modification levels looks very modest. Is SUMO1 conjugation different in wild-type vs. *Irx3*-KO mouse preadipocytes?
- As both forward and reverse reactions of SUMOylation are up-regulated in response to IRX3 depletion, how do the authors explain the increased cellular SUMO2/3 modification in *Irx3*-KO cells?
- Are these modest effects on cellular SUMOylation in response to IRX3 ablation also reflected in (perhaps clearer) changes in SUMO2/3 chromatin occupancy as assessed by SUMO2/3 ChIP-seq?

- To genuinely increase our mechanistic understanding of the potential role of SUMOylation downstream of IRX3 action, the authors should aim to identify (proteomic MS analyses) at least some key SUMOylated TFs/chromatin regulators/SUMO targets in *Irx3*-KO cells.

- Minor point: PGC1 α has been shown to be SUMOylated, which may contribute to the results in the reporter gene assays. It

would be pertinent to note this in this report with a citation.

Version 1:

Reviewer comments:

Reviewer #1

(Remarks to the Author)

In this revised version of their manuscript Bjune et al have addressed most of my comments and largely improved the manuscript. In particular, they provide much cleared immunoblotting showing increased SUMOylation in IRX3 KO cells and performed new ChIP-Seq analysis for SUMO and new RNA-Seq, which help better understand the links between IRX3 and SUMOylation.

I however think some points concerning the new experiments need to be addressed. The authors state throughout the manuscript that IRX3 represses SUMOylation and that IRX3KO cells have higher SUMOylation. However, quite surprisingly, the number of genomic locations showing changes in SUMOylation is very limited in the IRX3-KO cells, and there are more locations showing decreased than increased SUMOylation. Is the ChIP-Seq normalization taking into account the possibility of global increase of the mark (here SUMO) using Spike-In normalization instead of sequencing depth, which might prevent the visualization of global, genome-wide increase in SUMOylation. This is not clear from the method part. In any case, this discrepancy between the expected SUMO profile in IRX3 KO and the actual one should be discussed.

Along this line, they chose to highlight a gene from the Wnt pathway where IRX3 KO leads to a decreased SUMOylation, which again complicates the narrative. What happens with Rspo2 expression upon ML792 treatment?

To strengthen the link between SUMOylation on chromatin and IRX3-controlled gene expression, it would be interesting to determine if the genes being down-regulated in IRX3KO and up-regulated in ML792 (Figure 7) show increased SUMOylation on their regulatory regions upon IRX3KO.

Line 1018: « we observed significant enrichment of several cancer types ». Do you mean several cancer-related genes?

Reviewer #2

(Remarks to the Author)

In this revised manuscript, the authors have made substantial revisions to improve the manuscript, and their efforts in addressing previous concerns are appreciated.

This study presents interesting findings on the role of IRX3 in regulating adipogenic potential in adipocyte precursor cells. However, a critical limitation remains unresolved—the cell type specificity of the findings and their physiological relevance. The central claim that IRX3 modulates a SUMOylation-dependent switch between adipogenesis and osteogenesis is largely supported by data from ME3 cells, and there is no in vivo validation that IRX3 influences mesenchymal differentiation under physiological conditions. ME3 cells represent a specific cell type with unique properties, and it remains unclear whether the observed effects translate to other widely used adipogenic and mesenchymal lineage models, such as 3T3-L1 cells, primary preadipocytes, or bone marrow-derived mesenchymal stem cells. Although the authors acknowledged this limitation in the discussion, stating that “while IRX3 represses osteogenesis and promotes adipogenesis in adipose tissue, it appears to have an opposite effect in bones”, it still remains uncertain whether the findings from ME3 cells can be extrapolated to broader physiological contexts.

This reviewer is now more enthusiastic about this study. However, given these limitations, the title of the manuscript should be revised to better reflect the scope of the study. The current title implies a broad conclusion that IRX3 regulates mesenchymal lineage commitment, but as all functional evidence is derived from ME3 cells and previous studies have reported opposing *Irx3* functions in osteogenic tissues, the title should explicitly specify that findings are based on the adipocyte precursor cells. Suggested revisions for the title include:

- IRX3 controls a SUMOylation-dependent fate determination in adipocyte precursor cells
- IRX3 regulates a SUMOylation-dependent differentiation switch in adipocyte precursor cells
- IRX3 modulates SUMOylation-dependent adipogenic differentiation in ME3 cells

Reviewer #3

(Remarks to the Author)

All concerns were satisfactorily addressed.

Reviewer #4

(Remarks to the Author)

The authors have sufficiently dealt with my criticism. I do not have further concerns.

Version 2:

Reviewer comments:

Reviewer #1

(Remarks to the Author)

The authors have successfully addressed my comments.

Reviewer #2

(Remarks to the Author)

All concerns were satisfactorily addressed.

REVIEWER COMMENTS

We thank the reviewers for critically assessing the manuscript and for providing valuable input, comments and suggestions for experiments that have improved the manuscript.

All line numbers refer to the line numbers in the clean revised manuscript or the version with track changes with the “Simple Markup” setting.

Reviewer #1 (Remarks to the Author):

This manuscript investigates the role played by the *Irx3* protein in adipocyte differentiation. The authors present evidence demonstrating its interaction with DNA, influencing chromatin accessibility, and orchestrating changes in gene **expression** throughout the differentiation process. Notably, the authors propose a correlation between these effects and the capacity of *Irx3* to suppress genes associated with the SUMO pathway, resulting in a widespread reduction in post-translational modification levels. The study reveals that *Irx3* knockout (KO) leads to a diminished adipogenic differentiation while concurrently promoting osteogenesis.

In essence, this manuscript contributes new findings regarding the chromatin-related functions of *Irx3* and sheds light on its implications in the intricate process of adipogenic differentiation. However, the claims and conclusions are not always supported by the data. In addition, many of the data used in the manuscript were previously published by the authors, often in different cellular model. Their reanalysis is valid but the use of data coming from biological system different from those studied here limits the relevance and the strength of the conclusions. In general, figures lack sufficient explanations of their content and the conclusions are not supported by the data, at least given the provided information. We have improved the figure legends and modified our conclusions.

The number of biological replicates should be indicated for every experiment in the figure legend. We have added this information to the figure legends where it was missing as well as in the methods section.

Statistical analyses are not always provided
We have added this information to the figure legends where it was missing.

For example, for all ATAC-Seq experiments, I could not find the information on how many biological replicates were performed and I had no access to the raw data on ArrayExpress as the identifiers given don't match to any recorded study. At least 2, ideally 3, replicates should be performed.

- We have added this information to the figure legends (Figure 3A, Figure 8A and Supplementary Figures S1, and S2, as well as in the methods section. All ATAC-seq experiments were performed in duplicates (with technical duplicates each) or triplicates, except for one sample (iWAT d-1) in which only one replicate passed through QC control after sequencing.
- We apologize for not providing reviewer access to the raw data in the initial manuscript. Below is the access to ArrayExpress:
- Please, note that we cannot make these data publicly available until the manuscript is accepted, and before that neither reviewers nor us (submitters) can view or download the raw data because “data files for sequencing-based experiments are brokered to <https://www.ebi.ac.uk/ena>, but that resource does not provide an option similar to BioStudies where not yet public datasets can be shared”, according to EBI helpdesk.
 - **E-MTAB-13520:** <https://www.ebi.ac.uk/biostudies/arrayexpress/studies/E-MTAB-13520?key=dc863df3-b6df-407c-ac3b-ac4c4fb36c34>
 - **E-MTAB-13540:** <https://www.ebi.ac.uk/biostudies/arrayexpress/studies/E-MTAB-13540?key=eaf42902-3293-4791-8c57-227089a93c34>
 - **E-MTAB-13524:** <https://www.ebi.ac.uk/biostudies/arrayexpress/studies/E-MTAB-13524?key=d56e86e0-0c79-499f-8e3e-f08c2c648b7f>

- **E-MTAB-13525:** <https://www.ebi.ac.uk/biostudies/arrayexpress/studies/E-MTAB-13525?key=187ffe1d-522f-4f37-920c-fd2f4c232606>
- **E-MTAB-14723:** <https://www.ebi.ac.uk/biostudies/arrayexpress/studies/E-MTAB-14723?key=507bca73-10e6-491e-93c8-d0acfed63c04>
- Reviewer access to the GEO database is found here:
 - **GSE278972:** GEO Accession viewer (token: arqtsuigdzs1xgn)

Figure 1: it is stated that there is a dramatic increase in Irx3 binding events following initiation of differentiation. When looking at the profiles provided in Fig1E, I don't see much differences in the peak intensities between iWAT (day -1) and iWAT (day 1). For Uba2, the peaks are not considered as peaks in the iWAT (day -1) while it is for the iWAT (day 1) although I don't see a difference on the profile. Are these genes not representative of the peaks gained? Other profiles representative of genes with peaks appearing in the iWAT (day1) condition should be presented.

- We apologize for making an unclear statement. With increase in binding events, we initially meant increase in the number of called peaks, not increase in peak intensities. Although this holds true with the given threshold, we do see that many of the identified peaks in iWAT at day 1 are also present on day -1, although slightly weaker, thus missing the *a priori* set threshold of 10X enrichment. Overall, we agree that the provided data did not support a very distinct differential binding between day -1 and day 1. Since this was not our main message we have removed this paragraph from the manuscript.
- We have provided a heatmap and metaprofile of the unfiltered peak intensities in the new **Figure 1A** and B, respectively. These data show that there is a trend (these types of plots do not provide statistics) towards an increase in peak intensity in iWAT from day -1 to day 1. We have stated this in lines 112-114.
- Regarding the selected profiles (now moved to **Figure 1H**), they were intended to show the high consistency in IRX3 presence at the SUMO genes across the different samples despite variations in the degree of enrichment. As pointed out by the reviewer, the visual representation of the bigwig files clearly shows peaks even in cases where they were not called by the Macs2 software, due to insufficient enrichment (did not pass the *a priori* set threshold of 10X enrichment) in at least one of the parallels.
- As mentioned above, we have downplayed the difference between the different tissues and timepoints and treated the ChIP-seq data as a whole in the revised manuscript.

A heatmap with all the peaks with iWAT (day -1) as a reference should be provided.

- We have included a heatmap that was generated with the union of all the peaks as a reference and show it in the new **Figure 1A**.
- The specific heatmap requested (only loci with a peak in iWAT day -1 is provided at your convenience below, but we believe it is not suitable for the manuscript since all the unique peaks in iWAT day 1 and gWAT are omitted from this heatmap.

A metaprofile for Irx3 peaks around TSS for all conditions should also be provided.

- We have added this plot in the new **Figure 1B** (based on union of peaks)
- The metaprofile using iWAT d-1 as reference is shown below

For the gWAT, only the Day 1 is presented, which makes it difficult to state that there are differences as the basal status of Irx3 binding might be different to iWAT. Moreover, only one replicate was performed for this condition. This questions the robustness of the results and of the conclusions.

- We generally agree with the reviewer on this point. However, these are challenging tissues and a particularly challenging protein to capture on ChIP-seq. Unfortunately, we were only able to produce data from day 1 in gWAT and only one parallel. Thus, we agree that we cannot report on changes in IRX3 binding from day -1 to day 1 in gWAT. We have clarified this in the main text in lines 112-114. While the gWAT sample only has one parallel, the signal-to-noise ratio in this sample is superior to those obtained from iWAT (refer to the IRX3 ChIP-seq tracks). As such, we still find this sample relevant.
- Overall, this particular ChIP-seq experiment must be regarded as a hypothesis-generating explorative analysis and a starting point for exploring IRX3-chromatin interactions.

Figure 1: The enrichment of SUMO pathway genes is not so obvious. There are 3 genes of the SUMO pathway bound by Irx3 in iWAT out of 142 peaks. Moreover, in Supplementary File 1, I could not find any GO term linked to SUMO.

- We thank the reviewer for the comment. We have emphasized in lines 152-154 that SUMO was significantly enriched in iWAT, but not in gWAT.
- Regarding the enrichment in iWAT, even if the number of hits are low (3), the enrichment depends on the total number of genes in the pathway, which for the SUMO pathway is only 13 (out of 15,000 total genes). Enrichment is then calculated as $ie = \frac{\text{observed: } 3/142}{\text{expected: } 13/15000} = 24$. Of note, the actual reported enrichment will vary depending on how many peak ids were actually recognized by Reactome and used for their statistical analyses, which for iWAT day 1 was 83 out of 142 peaks.
- We acknowledge that results from GO analyses depend on their database annotations and will therefore vary between tools. Because the original REACTOME analysis was quite old (several years), we ran it again this August and updated the figure (new **Figure 1E**). This re-analysis actually resulted

in more SUMO hits, reflecting the rapidly evolving research field of sumoylation and updated annotations. On the other hand, the Panther GO analyses did not reveal SUMO, as pointed out. We still included these results to be as open as possible about our data and to let readers with particular interests have an alternative and expanded view of enriched GO terms. Moreover, we have provided the full GO results from the ENCODE GOs in addition to the PANTHER results supplementary file 1. Please also note that the “top” of such lists may vary depending on whether they are sorted according to enrichment or significance. We have provided both versions at your convenience.

Figure 1C: the detailed results of the reactome analysis should be given in the supplementary table. We totally agree and apologize for not including them in the previous version. The results from the Reactome analyses have been included in the supplementary file 1 of the revised manuscript, with separate sheets showing the pathways ranked either according to significance or enrichment. Pathways related to sumoylation, chromatin modifications/epigenetic regulation and adipogenesis have been highlighted in green, blue and yellow, respectively to aid the reader.

Figure S1A: The total number of ATAC seq peaks decreases upon differentiation, suggesting that the accessibility of chromatin globally decreases. From the text, we could indeed assume that it increases. We appreciate the reviewer’s comment. We actually shortened down this part just before submitting the manuscript and we have now realized that too much information was omitted. Thus, we have reintroduced a brief description of the data in lines 178-182: “While the total number of open regions was stable until day 7 of differentiation, followed by a 50% reduction in the mature adipocytes, there were dynamic changes in chromatin accessibility in ~20,000 loci across the different days, which included more open chromatin in regions related to fat cell differentiation, energy homeostasis and a wide range of other processes”. We also replaced “increasingly open” with “open” in the subheading.

FigureS1B: The numbers of the clusters should be indicated (assuming they are the same as those in the rest of the figure).
Cluster numbers have been added to the heatmap.

Figure S1D: specific GO terms were selected but there are many others in the list, which are more enriched than those indicated. The sentence line 166 should be modified to take this into account. Moreover the last sentence is, at least at this stage, an overstatement.
We agree with the reviewer, and have modified the sentence on line 166 to “which included more open chromatin in regions related to fat cell differentiation, energy homeostasis and a wide range of other processes”, in lines 180-182. We also removed the last sentence in this subchapter and generally have rewritten the whole subchapter in lines 175-201.

Figure S1E: it is not clear what can be concluded from this figure. It seems that there is indeed no strong correlation between the opening of chromatin and Irx3 binding.
We agree with the reviewer, and have modified the text in lines 184-185 to “these loci remained open throughout differentiation”

Figure S2: same remarks as SupFig1: the clusters should be indicated on B.
Cluster numbers have been added to the heatmap.

Figure S2I: the ATAC-Seq pic presented on KDM3a seems to decrease with the differentiation. This contradicts the message that Irx3-bound peaks get more accessible during differentiation.
We agree with the reviewer. This locus was used because of its high signal-to-noise ratio and we did not realize that it was one of the exceptions regarding openness. In the revised manuscript, the locus has been replaced with another locus which more clearly represents the increase in chromatin openness, at least in the ME3 cells.

Figure 2: based on the ATAC-Seq results, it is assumed that *Irx3* has the same binding profile in WAT and ME3 cells. I don't think this is necessarily true.

We agree that the binding profiles are likely not identical and realize that the text might be interpreted as if the profiles are the same. We have removed the last sentence on line 176-179 in the original manuscript and altered the text in the next paragraph (lines 204-209). Unfortunately, we do not have ChIP-seq data from IRX3 in ME3 cells and RNA-seq data from the iWAT/gWAT-derived cells. However, with our approach we picked up the most robust or consistent IRX3 target genes across similar, but not identical cell types. We have added a sentence about this in line 208-209 in the revised manuscript.

In addition, we were later able to do ChIP-qPCR in ME3 cells against some of the IRX3 target genes that were identified in WAT. Although these only represent a fraction of IRX3 target genes, it supports that IRX3 is found at numerous genes involved in SUMOylation in ME3 as in WAT.

Figure 2A: how are DEG selected? Is there a fold change selection? This should be indicated in the figure legend

DEGs were filtered as follows: $p_{adj} \leq 0.01$ and $FC \geq 1.2$. We have added this information to **Figure 2** and all other relevant figures.

Figure 2B: *Irx3* binding sites are equally abundant on genes that are up- and down-regulated upon differentiation. This makes it difficult to conclude on the role of *Irx3* in their expression and suggests that the binding of *Irx3* could be independent from its effect on gene expression.

We cannot completely exclude that the IRX3 binding might be independent of the gene expression in some cases, but we find it unlikely that IRX3 should solely repress all target genes. Although *Drosophila* orthologs have mainly been described as transcriptional repressors, IRX3 has been shown to activate target genes (ie PMID: 21825130, 26429810). Thus, we believe the direction of change is likely affected by interactions with other transcription factors and co-regulators that may alter the transcriptional function of IRX3. For example, in the heart, IRX3 has been found to physically interact with *Tbx5* and *Nkx.5* to activate target gene expression (PMID: 26786475). Moreover, for both *Irx2* and *Irx4*, the N- and C-terminal domains have been shown to have opposing roles on the transcriptional effect of the protein, with phosphorylation status acting as a switch between the two states (PMID: 15133517, PMID: 11382777). Thus, we find it likely that similar regulatory mechanisms exist for IRX3 in adipocytes. We have included this topic in the discussion, lines 891-909.

Figure 3: the effect of *Irx3* on SUMO pathway genes binding and expression is not really convincing. I don't understand the sentence line 232-233: the authors state that the SUMO pathway genes are enriched in the « differentially expressed *Irx3*-bound genes » but this is not significant. Then I would conclude they are not enriched in the differentially expressed genes if it is not significant. This contradicts the subsequent claim that these genes are up- or down-regulated (line 240-241) as well as the sentence in the abstract (line 56-57). Considering the low level of changes, I am anyway not convinced this translates in significant changes in protein abundance. This should be checked by immunoblotting, at least for the most dysregulated SUMO pathway proteins.

We agree that the phrasing in line 232-233 was counterintuitive and have removed it in the revised manuscript. We have rewritten this paragraph (lines 253-264). However, the fact that this GO term did not reach significant *enrichment* for the overlap between ChIP-seq and RNA-seq in gWAT does not mean that the binding did not occur or that the genes were not differentially expressed. Rather, it reflects that the SUMOylation genes were diluted when comparing the entire list of IRX3 target genes (including binding in gWAT) to DEGs, which often happens when integrating numerous datasets.

We agree that the changes in gene expression are modest. We analyzed the protein levels for most of the genes in **Figure 3A** by WB (new **Figure 3B-C**: *Senp1*, *Senp5*, *Sae1*, *Uba2*, *Ube2i* (*Ubc9*). (*Ranbp2* was too big at >400 kDa and we could not find an antibody targeting mouse *Usp11*). Overall, we found about the same FC on the protein levels as for the mRNA levels, except for *Senp5*, which has at least 2 isoforms, of which one was upregulated 15-fold, while the other was not changed.

We also checked other well-established SUMO E3 ligases that were not picked up by ChIP-seq in WAT for differential expression in ME3 control and IRX3-KO cells and found particularly Zmiz1 to be both highly expressed and also upregulated 2-fold in KO cells compared to control cells, in addition to slightly upregulated Pias1 and Pias2 expression (New **Figure 3D**). Taken together, we believe these data support functional changes in SUMOylation.

Figure 3C-F: the increase in SUMOylation in the KO is not really convincing. Whilst in Fig3C, I can see an increase, this is not that clear in Fig3E. The experiments in Fig3C, F should be performed at least 3 times, which seems not the case. In Fig3F, no statistical significance is provided, which questions the validity of the presented results. The levels of SUMOylation should also be analyzed in iWAT and gWAT upon induction of differentiation

We have repeated the experiments (3-4 independent experiments in total) and increased the number of biological replicates/wells to n=3 per experiment to allow statistical analyses (New **Figure 3E-F and I-J**). The data show a significant increase in SUMOylation with IRX3-KO. We also analyzed levels of conjugated SUMO1 according to another reviewer's request, and found significant effects (although less marked) (**Figure 3G-H**), although IRX3 overexpression still reduced the SUMOylation levels (**Figure 3K-L**).

There is no Supplementary Figure S4

We apologize for the error in numbering the supplementary figures. We have made rearrangements in the supplementary figure numbering in the revised manuscript, and S4 now includes new data showing IRX3 ChIP-qPCR against genes in the SUMOylation cycle in ME3 cells.

Figure 4: Although the effect of *Irx3*-KO on SUMOylation is not really convincing (Figure 3), the effect of ML792 on bodipy staining is impressive. The effect on the genes is also very clear. It might therefore be that *Irx3* primarily affects the SUMOylation on the chromatin. It would thus be interesting to see if there are changes in the level of SUMO on the adipogenic genes in wt vs *Irx3* KO cells by ChIP-Seq or ChIP-qPCR. We performed SUMO2/3 ChIP-seq in ME3 control and IRX3-KO cells (n=2 replicates per condition), please refer to the new **Figure 6**. We analyzed the data systematically and found reduced SUMO occupancy on genes involved in Wnt and Rho signaling (6 GO terms), which are well-known pathways that promote osteogenesis over adipogenesis, to be among the top 15 most enriched GO terms. We found enriched motifs for KLF and SP family members at these loci, suggesting that they may be SUMOylated and have altered binding upon IRX3 ablation. Moreover, we also found evidence that RUNX2 and PPAR γ (known SUMO targets) overlap the SUMO peak in *Rspo2*, one of the Wnt-signaling genes. Finally, genes known to be activated and repressed by Wnt were up- and downregulated, respectively in the KO cells, supporting increased Wnt signaling in the KO cells, suggesting SUMO had a repressive effect on this pathway.

Figure 5B: it is stated that ML792 increases PPAR γ activity 2-4 fold. It is not really obvious from the graph provided. Moreover, no statistical analysis is provided on the graph. I agree that the effect becomes clear in the presence of Rosiglitazone and PGC1 α but the ligand is not present in the context of the differentiation model, suggesting that the desumoylation of PPAR might not be the main driver of ML792 induced adipogenesis in *Irx3* KO cells

Applying statistics on luciferase data can be challenging, especially with complex setups and large differences in effects, leading to very large differences in SDs (heteroscedasticity), which violate the assumptions of ANOVA. However, after square-root transformation of the data, we were able to conform with these assumptions and we have added the statistics to the revised **Figure 5B**.

We agree that it is not easy to read the precise height of each bar from the split Y-axis. Please refer to the below plot where the FC between selected graphs is indicated:

Please note that the differentiation model does indeed contain Rosiglitazone on day 1 of differentiation. We apologize for making the differentiation procedure hard to check by linking to a previous paper. Here is the information from the linked paper (Bjune et al 2020):

“Adipogenic differentiation was initiated three days post confluency (day 0) by **induction medium** containing 5 µg/mL Insulin (INS) (Sigma), 1 µM Dexamethasone (DEX) (Sigma), 0.5mM isobutylmethylxanthine (IBMX) (Sigma) and **1 µM Rosiglitazone (ROSI)** (Cayman Chemical, Ann Arbor, MI, USA). From day 2 to day 4 only insulin was added to the basal medium and from day 4 to 7 cells were grown in the basal medium.”

Figure 6: I don't think that the comparison of the RNA-Seq in *Irx3*-KO in ME3 to the SLAM-Seq in 3T3L1 is very relevant.

We agree with the reviewer that this comparison can be omitted, and have removed the SLAM-seq data from the manuscript.

Moreover, as around 1/3 of the protein-coding genes expressed in mouse genome are changing expression in *Irx3* KO, it is somehow normal that there is some overlap with ML792 regulated genes. It would be more informative to perform RNA-Seq (or SLAM seq) in ME3 wt and *Irf3* KO +/- ML792.

We agree with the reviewer's comment, and we have performed the requested experiment (new **Figure 7** and new supplementary **Figure S9**). Here we treated ME3 control and *IRX3*-KO cells with DMSO or ML-792 for 24h before (day -2 to day -1) or after (day 0 to day 1) induction of adipogenesis, as well as throughout differentiation (day 0 to 9). This comprises 12 experimental conditions, each in n=3 biological replicates from one experiment. Changes in gene expression was analyzed by RNA-seq. We have moved the comparison between ME3 cells and 3T3-L1 cells from the old Figure 6 to the new Supplementary **Figure S7**).

Figure 6: *Irx3* KO increases SUMOylation so should have an opposite effect to ML792, which decreases SUMOylation. However, when looking at the overlapping DEG, this is not the case. Genes up-regulated in *Irx3* KO are not necessarily down-regulated in ML792 and vice versa. This questions the relevance of the conclusions.

In the new **Figure 7C** we have included correlation plots which show the direction and magnitude of change between overlapping genes in the two datasets, both now derived from ME3 cells, which allows for a direct comparison. While the number of inversely regulated genes between *IRX3*-KO and ML-792 is about 50% on days -1 and 1, this number increases to almost 90% by day 9, clearly demonstrating that *IRX3*-mediated changes in SUMOylation are affecting the expression these genes.

Reviewer #2 (Remarks to the Author):

In this manuscript, Bjune et al demonstrated that IRX3, known to be a causal gene of obesity-associated FTO variants, plays a critical epigenetic role in the regulating adipogenic and osteogenic potentials, particularly via sumoylation. To reach this conclusion, the authors utilized multi-omic approaches, including ChIP-seq, ATAC-seq and RNA-seq, in combination with cell biological experiments. The manuscript is interesting, but below are points that I hope will strengthen the study.

1. One of the novel results in this manuscript is *Irx3* ChIP-seq data in preadipocytes, as its direct targets have not been well studied. However, the manuscript lacks clarity regarding the *Irx3* ChIP-seq experiment. The details about ChIP antibodies against *Irx3* and the control antibodies were not described in the methods section. Please provide information about the antibody (cat#, log#, concentration). We apologize for the missing information. We have included this information and more in the revised method section, lines 1112-1113. Of note, two pooled DNA inputs were used as negative controls instead of IgG. We quote Active Motif who performed this ChIP-seq experiments regarding this matter: "We find there is not much value in IgG controls since in our process, the results are always negative, and the anti-IgG antibody is unrelated to the target antibody and is not a helpful indicator of the sample. Instead, we ran two pooled DNA inputs. These control for false peaks created by things such as sonication conditions, artifacts, etc. We use this in the analysis pipeline instead of IgG."

Especially, a clear validation of the *Irx3* antibody specificity will be required and shown (e.g. IP-WB, cross-reactivity testing etc).

Below are SDS-PAGE followed by WB using this antibody on: cell lysates from 3T3-L1 cells +/- siRNA knockdown of IRX3 (left). The correct band corresponding to endogenous IRX3 is ~55 kDa. A cross-reactivity test was performed by overexpressing IRX3-GFP, *Irx5*-GFP or an empty plasmid in ME3 cells (right). We observed no noticeable cross-reactivity with *Irx5*.

Fig. 6.8 | Western Blot using ChIP-grade antibody against IRX3. IRX3 antibody used was abcam ab25703 (lot GR198517-1). IRX3 is predicted to be 52kDa. 3T3-L1 pre-adipocytes were treated with siCON (lanes 1-2) or siIrx3 (lane 3) before protein was isolated and Western Blot performed. Bottom panel shows GAPDH on same blot (antibody used was anti-GAPDH antibody (ab8245)).

- 15 µg whole lysate loaded in each well on a 4-20% TGX gel
- Transferred to nitrocellulose membrane with BioRad turboblot 7 min
- blocked in 5% dry milk/PBST o/n 4°C + 40 min RT
- anti-Irx3 (ab25703) diluted 1:400 in 3% BSA/PBST for 1h 30 min
- anti-rabbit (31460) diluted 1:7500 in 3% BSA/PBST for 35 min
- detected with femto: luminol:peroxide:H₂O 1:1:1, 2 min incubation

Although IgG is conventionally used as a control in ChIP-seq experiments, I suggest a more specific approach given that the authors have *Irx3*KO ME3 cells. It would be strongly suggested to perform *Irx3*-ChIP using the same anti-*Irx3* antibody in both WT and *Irx3*-KO ME3 cells, and then compare the data to the IgG control ChIP-seq data. This method would help in generating *Irx3* ChIP-seq data free from non-specific binding.

Since IgG was not used as a control in the ChIP-seq experiment, we cannot directly address this comment. However, we have performed ChIP-qPCR on sumoylation genes using the same IRX3 antibody in ME3 control and IRX3-KO cells (**Supplementary Figure S4**). Ratios of % input in control/KO are approximately 1 for the control regions and 2-3 for the genes involved in SUMOylation, showing specific binding to the genes involved in SUMOylation in the control cells.

Additionally, it appears that motif analysis for *Irx3* binding regions was not performed. Did you observe any putative *Irx3* binding sequence?

We have performed motif analysis in the *IRX3* binding regions (new **Figure 1 I-K**). In vitro experiments show that *Irx* family members bind as homo- or heterodimers directly to $ACAN_xTGT$ palindromic sequences, but there is some discrepancy in the literature regarding the number of nucleotides between the two half sites, although $x=2$ seems to be most frequently reported. In addition, binding has also been reported for the inverse motif (TGTnnACA).

In our analyses, we identified the inverse *IRX3* motif (TGTnnACA) (significant p-value), but this finding did not reach a significant E-value. However, two other highly significant motifs were identified, matching ETS family members and other transcription factors, respectively. These data suggest that in the cell type and developmental stage we analyzed, *IRX3* may also bind DNA indirectly via complexes with other TFs.

2. While *Irx3*-ChIP seq was performed in primary preadipocytes (isolated from SVF), most of the other experiments were performed in ME3 cells. Although ME3 cells are known to be useful for studying beige adipocytes, they are fundamentally mouse embryo fibroblasts, which are developmentally different and more potent than fate-determined preadipocytes. Consequently, the observed phenotypes, such as decreased adipogenic potential and increased osteogenic potential, may reflect developmental changes rather than specific adipocyte and adipogenesis mechanisms. This is supported by the ATAC-seq data at day 0, which showed higher open chromatin in *Bmp2* and *Wnt* in *Irx3*-KO ME3 cells, indicating the developmental role of *Irx3* in the fate determination of mesenchymal cells. Conducting *Irx3* ChIP-seq in ME3 cells would further strengthen these findings.

The reviewer is absolutely correct on this matter, we used the ME3 cells because they are indeed developmentally upstream of primary preadipocytes, and thereby closer to mesenchymal stem cells (which is the cell type where human *IRX3* expression was found to be affected by a SNP in the intron of *FTO*). Unfortunately, we have not been able to successfully perform full-scale ChIP-seq in the ME3 cells (they are very hard to lyse when crosslinked, which causes problems when applying the standard commercial pipelines. However, using in-house methods, we were at least able to do ChIP-qPCR against the sumoylation genes in ME3 control and *IRX3*-KO cells, and we found specific binding in the control ME3 cells to these genes (**Supplementary Figure S4**). Thus, we find that *IRX3* consistently binds to sumoylation genes at both developmental stages.

Furthermore, the authors' main claim that *Irx3* controls a sumoylation-dependent epigenetic switch between adipogenesis and osteogenesis seems less relevant when considering cell type variability. If this claim is relevant, it is important to examine whether *Irx3*KO primary preadipocytes or 3T3-L1 cells also demonstrate a similar propensity to differentiate into the osteogenic pathway.

We do not completely share the reviewer's view on the difference between 3T3-L1 and ME3 cells, because they are both derived from mouse embryonic fibroblasts (MEFs) and both readily differentiate into adipocytes using standard differentiation protocols. Moreover, the ME3 cells also appear to be committed to the adipogenic lineage since the control cells do not respond to chondrogenic (PMID: 31751577) or osteogenic (present manuscript) stimulation, while the *IRX3*-KO cells do. Although there are some important differences, such as the ME3 cells being more beige or brown-like than 3T3-L1 cells, both are MEF-derived and both are committed to the adipogenic arm of mesenchymal fate. That being said, it is a good question whether primary preadipocytes would undergo the same switch. However, proper CRISPR-Cas9 knockout of *IRX3* requires generation of stable clones by immortalization of primary preadipocytes, but this would entail a new project and is beyond the scope of the present work.

The findings appear to be specifically relevant to ME3 embryonic fibroblast cells, raising questions about the generalizability of the claim to other cell types and in vivo contexts. In vivo, the role of *Irx3* might be more pertinent to bone health or bone marrow dynamics rather than to adipose tissue function or obesity. Therefore, it is crucial for the authors to explore and clarify how the observed mechanisms and findings in cell culture models translate to in vivo and physiological conditions. Investigating this aspect

would not only enhance the understanding of *Irx3*'s role in different tissues but also provide a clearer picture of its potential impact on obesity and related metabolic conditions.

The reviewer raises an important question regarding the role of IRX3 across different tissues *in vivo*. Previous mouse models have demonstrated that IRX3 exerts vital roles outside adipocytes, for instance during embryogenesis, and in the heart, brain and bone. Whole-body IRX3-KO mice display, apart from reduced adiposity, lower survivability, reduced body size (runts) and osteopenia. However, while adipocyte-specific KO of IRX3 alone is sufficient to recapitulate the effect of global IRX3-KO on body weight and fat storage in adipose depots, the detrimental effects of bone-marrow-specific deletion of IRX3 are mainly only apparent in a background of IRX5-KO (lines 770-806). These data suggest a stronger and more autonomous role of IRX3 in adipose tissues compared to the bone marrow (lines 789-792).

That being said, it is interesting that IRX3 appears to have an opposite effect on adiposity in bone marrow compared to adipose tissues, in which IRX3-KO leads to increased bone marrow adipogenesis (lines 778-793). Interestingly, a recent study found this opposite effect only in males and not in females (lines 796-806). Thus, organ-specific, time-specific and sex-specific ablation of IRX3 all appears to result in different phenotypes. We acknowledge the need for further studies of conditional knockout and knockin models (line 802-806).

3. Due to the reason above, it seems overly speculative to conclude that these findings are directly related to FTO genetic variants-associated obesity. To establish a more concrete connection, *in vivo* or *in vitro* experiments should be conducted using preadipocytes from mice lacking *Irx3* in the context of obesity. This approach would provide more relevant data to understand the role of *Irx3* in the context of FTO variant-associated obesity.

We agree with the reviewer and have toned down the link to the obesity-associated genetic variants. As pointed out, establishing direct mechanisms that connect FTO variants with IRX3, changes in epigenetics and changes in adiposity requires a follow-up study and is beyond the scope of the current manuscript. We have altered the text in lines 926-943 accordingly.

4. In Figure 2A, the criteria used for analyzing the RNA-seq data (*Irx3*KO/CTRL) are not clearly defined and appear to be overly permissive. This is evident from the excessively high number of differentially expressed genes (DEGs), exceeding 11,000. Such a large number of DEGs implies statistically significant risk of overlap, potentially up to a 50% chance for any given gene. Additionally, the differential expression levels of sumoylation genes are relatively modest, with a log₂ fold change of less than 0.6. This indicates a change less than 50%, which may not be substantial enough to draw strong conclusion.

We apologize for not showing the criteria. In our re-analysis of the data, we used $p_{adj} \leq 0.01$, $FC \geq 1.2$ in order to match the criteria of the other RNA-seq analyses that we have included in this paper. We have included this in the legend to **Figure 2A** in the revised manuscript.

We agree that this FC threshold can be considered overly permissive, especially the way RNA-seq were analyzed before. However, common practice now is to use p_{adj} only and not include log₂ FC. The reason for this is that a log₂ FC of 1.5 that is significant for a highly-transcribed gene will result in millions of extra reads while a log₂ FC of 10 for a lowly-transcribed gene will only result in a few more reads, which actually is biologically less likely to be significant. Moreover, even small FC in many genes in the same pathways may provide important biological insights. This appears to be the case for the genes in the SUMOylation cycle, in which modest changes in many genes in the same pathway seems sufficient to lead to a strong phenotype.

Similarly, the logical analysis pipeline focusing on epigenetics and sumoylation through bioinformatic analysis seems unclear. Considering the nature of GO term analysis, which works with lists of genes regardless of the extent of differential expression, the uncertain criteria for fold change and the exceptionally high number of DEGs could generate a GO list with many available functional terms. Therefore, in the GO term and KEGG pathway analysis, the authors should provide the overall list of terms generated and the actual rank of epigenetics and sumoylation-related pathways. This will clarify whether these pathways were identified as high-ranking hits from an unbiased analysis or if they were

specifically selected by the authors based on their interests. Such transparency is crucial to assess the relevance and impact of these pathways in the study.

We agree with the reviewer, and have provided the complete lists of GO REACTOME pathways from the IRX3 ChIP-seq analysis, with pathways ranked both according to significance and according to enrichment (Supplementary file 1). Please also refer to Supplementary Files 5, 6, 8, 11, 12, 14, 15, 16 and 18 for complete lists of GO analyses for the various omics data presented in this manuscript. Of note, we have updated figure legends to clarify whether the shown GOs are from the top of the lists (most of the cases), or whether a few, selected GO terms have been showed to convey a specific message.

5. There seems to be an inconsistency in the data comparison between different sets of omics data. In Figure 2A, the number of DEG on day 1 (Irx3KO/WT ME3) was reported as 11486, while in Figure 6, this number is reduced to 8013. Furthermore, the cutoff criteria for these analyses should be clearly stated. When they are mentioned, as in Figure 6A, the authors have used a fold change of 1.2 (not log₂FC) as the threshold, which could be considered overly lenient for bulk RNA-seq. A more stringent FC threshold is preferred to reduce the likelihood of identifying false positives in such analyses.

- We thank the reviewer for pointing out these issues. We have updated the figure legends with cutoff criteria.
- Regarding **Figure 2**, both datasets have used padj <0.02 and FC>1.2.
- In **Figure 2**, the number of genes was actually supposed to represent the union of both day 1 and day 7. We discovered an error in the reported number and have updated the figure accordingly. The correct number here should be 14272.
- In the old Figure 6 (now **Supplementary Figure S7**), only day 1 is shown, and the reported number (8013) is correct.
- It is more common now in RNA-seq analyses to not include FC thresholds at all actually (see the detailed explanation in our response to point 4 on page 10). In this regard, we use a more stringent criterion than many.
- Finally, while two random datasets with may give false overlap by chance, there is a biological rationale for the overlap in this case since both datasets are describing genes affected by the same protein (IRX3).

6. Figure 4D appears to be confusing as it does not accurately reflect the changes occurring in Irx3-KO ME3 cells, potentially leading to misinterpretation. For example, Fabp4 appeared to be shown as 8-fold higher in Irx3-KO cells treated with ML-792. However, Supplementary Figure S5 presents more direct and comparable data. It would be helpful if the authors could ensure that the figures accurately represent the underlying data to prevent any confusion or incorrect interpretation of the results.

- We have moved the data from Supplementary **Figure S5** to the new **Figure 4E** according to the reviewer's request.
- It should be noted though, that the original **Figure 4D** showed the ML-792/DMSO ratio, as stated in the title of the y-axis. In other words, this figure shows the fold change of gene expression in response to ML-792 in two different cell lines. We believe fold changes are commonly used to show the magnitude of response to treatments. Thus, the figure shows how much these genes change in expression in response to ML-792 in Ctr and IRX3-KO cells, respectively. For example, while ML-792 treatment only increases Fabp4 expression 1.5-fold in control cells (compared to DMSO), it increases Fabp4 expression 8-fold in the control cells (compared to DMSO).
- However, as pointed out by the reviewer, fold changes do not take into account the absolute levels of gene expression (which is often neglected in RNA-seq studies). This is why we originally included the absolute levels in **Supplementary Figure S5B**. But in this figure, it is very hard to tell from the absolute levels how much ie. Fabp4 expression increases in response to ML-792. Overall, we believe both absolute and fold changes are relevant in this case and we have therefore included both in the main figure in the revised manuscript as mentioned above.

- We have also added some clarifying sentences in lines 349-354 to emphasize that the ML-792 treatment does not completely restore the adipogenic gene expression in the KO cells.

7. Figures 3C and 3E present what appears to be overlapping data regarding *Irx3* KO ME3 cells, yet the results showed inconsistency, particularly for day 0. This discrepancy raises questions about the suitability of the methods in these experiments.

We have repeated the experiment several times and with more parallels to assess the robustness of our findings (see figure below). We consistently observe an increase in sumo levels on both days. As it turns out, the low increase on day 0 shown in the original **Figure 3D** appears not to be representative. Thus, we have replaced it with a new figure that shows a more representative (and higher) increase.

All independent SUMO-WB experiments showing baseline levels

Additionally, it is unclear whether the treatment with ML-792 leads to a reduction in the global SUMOylome pattern in *Irx3*KO cells. Clarification on this point would be valuable, as it could significantly impact the interpretation of the ML-792 treatment effects in the context of *Irx3* knockout.

ML-792 is a highly potent inhibitor that will eradicate the SUMO signal almost completely. Please refer to a previously published experiment where we showed this effect in 3T3-L1 cells (Zhao et al, NAR 2022, **Supplementary Figure S1C**, and copied below). As such, we did not find it necessary to repeat this experiment in the ME3 cells.

8. I recommend the authors specify the cell types used in each experiment when describing the results. Since various cell lines, including ME3, 3T3-L1, and primary preadipocytes, were utilized and compared within the study, explicitly identifying the cell type in each instance will aid in avoiding confusion and will enhance the clarity and accuracy of the interpretations made from the data.

We fully agree and have updated the figures and/or figure legends/main text accordingly.

Reviewer #3 (Remarks to the Author):

In the present article by Bjune JI et al. revealed a newfound connection between IRX3 and the sumoylation pathway. IRX3 was identified as a direct controller of the SUMO machinery and chromatin remodelers. Inhibiting sumoylation was demonstrated to rescue adipogenesis in cells where IRX3 was knocked out, highlighting IRX3's inhibitory role on sumoylation during the early stages of cell differentiation. The study also illustrated that IRX3 and sumoylation have overlapping target genes, particularly those associated with the development of adipocytes and osteoblasts, indicating a complex interplay between these elements in determining the fate of cells.

From a functional perspective, the work by June et al. uncovered that cells lacking IRX3 exhibited a modified epigenetic landscape that favored osteogenic differentiation over adipogenesis. This shift towards enhanced osteogenesis was linked to a more permissive chromatin state in the absence of IRX3. Crucially, the study demonstrated that the promotion of osteogenesis resulting from IRX3 ablation was, in part, mediated by heightened levels of sumoylation. Inhibiting sumoylation was shown to counteract the pro-osteogenic effects of IRX3 ablation. These findings significantly contribute to a deeper comprehension of the intricate regulatory network involving IRX3, sumoylation, and their influence on the differentiation of adipocytes and osteoblasts.

The manuscript is well-written, follows a logic narrative and presents interesting data supporting their hypothesis. I believe the findings will be of interest of the broad readership of nature communications.

My main criticism is even though their work is partially supported by others findings, as it is highlighted in the discussion, without in vivo data using floxed mice that could be targeted with adipose or bone specific care lines needs to be substantiated. See floxed mice in Gaborit N. et al Development 2012 or Cain CJ et al 2016 Bone Reports. I believe tuning down claims in the discussion and highlighting the need of in vivo models to support the claims would be appropriate. Otherwise, authors could consider testing some of their hypothesis in vivo.

We agree with the reviewer and have toned down claims in the discussion and expanded upon findings of others regarding the effect of IRX3 in bone cells (lines 770-806), as well as highlighted the need for in vivo experiments using tissue-specific manipulations of IRX3 (lines 802-806). There is an interesting opposite effect of IRX3 on adipogenesis and osteogenesis in adipose and bone tissues, but this effect *may* be sex-specific (see our answer on page 10).

Conceptually, I believe that based on the readership of nature communications I believe that a broader section on the biological impact the innings could have beyond FTO and obesity could be of interest. The numbers of osteoporosis and other bone defects keep rising. Do authors believe their findings could help epigenetically modify some of those detrimental states. A bit of big picture impact would be appropriate. We thank the reviewer for this input. As mentioned above, we have included a section regarding IRX3 in bones (lines 770-806), and discuss implications in other tissues where IRX3 is known to play a role (lines 808-825). Future studies should investigate whether IRX3 ablation affects epigenetics in bone marrow cells before we can consider epigenetic modifications in these cells.

Experimentally, in the context of the frame provided I believe the work has been done at the highest standards and statistics are both sufficient and appropriate in all figures. With an in vitro frames I believe this manuscript is ready for publication and all logic questions are appropriately answered. We thank the reviewer for this comment.

Reviewer #4 (Remarks to the Author):

Bjune and coworkers studied *Irx3* target genes during early adipogenesis in wild-type vs. *Irx3*-KO mouse preadipocytes. They firstly identified > 300 “strong” *Irx3*-binding sites in preadipocytes, mainly at promoter regions, with a strong enrichment of genes linked to SUMOylation, histone modifications and chromatin remodeling. Several genes of the SUMOylation machinery were bound by *Irx3* and differentially expressed in *Irx3*-KO; *Sumo1* and -3 down-regulated, *Sae1* and *Uba2* (E1), *Ube2i* (E2), *Senp1* as well as -5 up-regulated.

Western blotting analyses suggested that these alterations in the expression of both forward and reverse reaction-catalyzing SUMOylation machinery genes lead to a modestly increased SUMO2/3 conjugation of cellular proteins in the KO cells. *Irx3* ablation compromised Ppar γ -dependent reporter gene activity and adipogenic differentiation in preadipocytes, which could be partly restored by general SUMOylation inhibitor ML-793. Evidence supporting the notion that SUMOylation is playing a role also in the osteogenesis process induced by *IRX3* ablation is provided by experiments using ML-792. The authors conclude that *IRX3* acts as a novel upstream regulator of sumoylation, and a potent controller of epigenetic regulators, both directly and indirectly via suppressing global sumoylation levels.

This is a potentially interesting but somewhat incomplete study. It could be considerably strengthened by considering the following points:

- Is the expression of other deSUMOylation enzyme genes altered in the *Irx3*-KO cells?
- Is the expression of PIAS and other SUMO E3 ligase genes affected by *IRX3* depletion?

We thank the reviewer for the question/suggestion. We checked for both additional E3 ligases and desumoylases (New **Figure 3D**) and found several E3 ligase to be upregulated in the *IRX3*-KO cells. Among these, *Zmiz1* was found to be both highly expressed and also most upregulated in KO compared to control cells, in addition to moderate increases in *Pias1* and *Pias2* expression (New **Figure 3D**).

However, we also found the desumoylases *Senp2*, *Senp3* (minimal change, but significant) and *Senp6* to be upregulated to some degree in the KO cells while *Senp7*, *Desi1* and *Desi2* were not significantly changed. The upregulation of *SEN6* will primarily affect poly-SUMO2/3 chains. Taken together, these data support increased maturation and ligation of SUMOylation in the KO cells, but possibly also increased de-sumoylation. The SUMO-WB shows that the net result is a modest increase.

- The enhancing effect of *IRX3* depletion on cellular SUMO2/3 modification levels looks very modest. Is SUMO1 conjugation different in wild-type vs. *Irx3*-KO mouse preadipocytes?

We analyzed SUMO1 conjugation (new **Figure 3 G-H and K-L**) and found again a modest effect on day 1, but a stronger effect on day 0.

- As both forward and reverse reactions of SUMOylation are up-regulated in response to *IRX3* depletion, how do the authors explain the increased cellular SUMO2/3 modification in *Irx3*-KO cells?

When looking at the absolute levels, it is clear from **Figure 3D** that the SUMO E3 ligase *Zmiz1* is much more highly expressed than any of the desumoylases. This also holds true for the other sumo-genes in **Figure 3A**, in which the activating, conjugating and ligating enzymes are more highly expressed than the deSUMOylating genes (which also serve a positive role by maturing newly synthesized sumo). These data support a net increase in sumo conjugation.

Gene	Average signal in control cells	Average signal in KO cells
Senp1	2469	2994
Senp5	1171	1547
Sae1	4938	5370
Uba2	5062	6306
Ube2i	3729	3786
Ranbp2	6315	8593
Uspl1	1313	994

- Are these modest effects on cellular SUMOylation in response to IRX3 ablation also reflected in (perhaps clearer) changes in SUMO2/3 chromatin occupancy as assessed by SUMO2/3 ChIP-seq?

We performed SUMO2/3 ChIP-seq in ME3 control and IRX3-KO cells (n=2 replicates per condition) as requested by the reviewer, please refer to the new **Figure 6**. As seen in **Figure 6A**, we do observe a trend of massive rearrangement of SUMO2/3 occupancy at the chromatin. However, when applying statistics, a smaller number of differential SUMO peaks were significant. Still, the most significant changes were found in genes regulating the Wnt signaling pathway, which is a key determinant in adipogenic vs osteogenic lineage determination.

- To genuinely increase our mechanistic understanding of the potential role of SUMOylation downstream of IRX3 action, the authors should aim to identify (proteomic MS analyses) at least some key SUMOylated TFs/chromatin regulators/SUMO targets in *Irx3*-KO cells.

While we agree that such a study may be relevant for unraveling and validating underlying molecular mechanisms by which SUMO exerts its effects, we would like to point out that gaining such molecular details lies outside the focus of our story.

Even though we are experienced with such an approach, performing follow-up MS experiments would be very time-consuming. In addition, these experiments would yield at least 1000 targets according to our previous work in differentiating adipocytes (PMID: 35100417) and other works (PMID: 32937131, PMID: 18408734, PMID: 27435506). Therefore, processing these data to obtain more insightful information than just a list of proteins would constitute a separate project and the basis of a follow-up study, involving the construction of clonal mutant cell lines followed by in-depth characterization of SUMO target candidates in adipogenesis and osteogenesis. For these reasons, this “follow-up” strategy was used in (Theurillat et al, Cell Rep 2020, PMID: 32937131) to follow up on the important study (Cossec et al, Cell stem cell, 2018, PMID: 30401455).

More pragmatically, these experiments, to be performed at the highest standards and yield a biologically relevant result, require a very specific expertise only owned by the group of Michael Nilsen (Hendriks et al, Nat Commun 2018), with whom we collaborated before (Zhao et al, NAR 2022). Unfortunately, this group ceased to exist during the course of these revisions, and it will take time before we have access to SUMO MS again via sufficiently competent collaborations.

- Minor point: PGC1a has been shown to be SUMOylated, which may contribute to the results in the reporter gene assays. It would be pertinent to note this in this report with a citation.

We thank the reviewer for highlighting this information. We have included this in the description of the luciferase in lines 415-420.

Revision 2 (NCOMMS-23-49769A)

Reviewer #1 (Remarks to the Author):

In this revised version of their manuscript Bjune et al have addressed most of my comments and largely improved the manuscript. In particular, they provide much cleared immunoblotting showing increased SUMOylation in IRX3 KO cells and performed new CHIP-Seq analysis for SUMO and new RNA-Seq, which help better understand the links between IRX3 and SUMOylation.

I however think some points concerning the new experiments need to be addressed. The authors state throughout the manuscript that IRX3 represses SUMOylation and that IRX3KO cells have higher SUMOylation. However, quite surprisingly, the number of genomic locations showing changes in SUMOylation is very limited in the IRX3-KO cells, and there are more locations showing decreased than increased SUMOylation.

Is the CHIP-Seq normalization taking into account the possibility of global increase of the mark (here SUMO) using Spike-In normalization instead of sequencing depth, which might prevent the visualization of global, genome-wide increase in SUMOylation. This is not clear from the method part. In any case, this discrepancy between the expected SUMO profile in IRX3 KO and the actual one should be discussed.

The reviewer raises an important question. With respect to normalization, we did not use spike-in controls, but relied on normalization to sequencing depth and input DNA. We have added this information in the methods section in lines 1250-1251. While sequencing depth alone would dilute a systemic increase in global SUMO levels, this should be at least partially corrected by also normalizing to input DNA, which is not affected by differences in total SUMOylation levels.

Regarding the divergence between global and chromatin-bound SUMO levels, the reduced presence of SUMOylated proteins at chromatin in Irx3 knockout (KO) cells may result from a combination of the following factors:

- **Divergence between global SUMOylation levels and chromatin-associated SUMOylation:** As observed with other post-translational modifications, an increase in

global SUMOylation detected by Western blot does not necessarily correspond to increased SUMOylation at chromatin [1]. For example, we previously demonstrated that while global SUMOylation increases during adipogenesis in WB and ChIP-seq experiments using 3T3-L1 cells, specific chromatin regions exhibit either stable or transient decreases in SUMOylation. Furthermore, the presence or absence of SUMO at target genes can correlate positively or negatively with gene expression, depending on gene subsets, differentiation stage, or cell type (pre-adipocyte vs. mature adipocyte) [1,2]. This suggests that multiple dynamic mechanisms regulate SUMOylation simultaneously to fine-tune chromatin-based processes. Since IRX3 may not be the sole regulator of SUMOylation, other repressors or activators could compensate or become more prominent in its absence too.

- **The indirect effect of *Irx3* KO on SUMOylation at the chromatin:** IRX3 regulates genes beyond those directly involved in SUMOylation. Thus, we can't exclude that although the identity shift we observed is clear, some reductions in SUMOylation at the chromatin could be a secondary consequence of dysregulation in other *Irx3* target genes rather than a direct effect on the SUMOylation pathway itself.
- **Potential compensatory mechanisms:** Cells may activate compensatory pathways to counteract excessive chromatin-associated SUMOylation in *Irx3* KO cells [3]. Similar compensatory mechanisms have been described in transcriptional regulation, where the knockout of general transcription factors (GTFs) such as SAGA paradoxically leads to increased mRNA levels, counterbalancing otherwise severe transcriptional reductions [4].
- **SUMOylation patterns observed may simply reflect cell identity conversion:** The heterogeneous SUMOylation landscape observed in Figure 6 (regions of both higher and lower SUMOylation) may be a direct consequence of the stable conversion of beige cells into osteoblasts. Similar SUMOylation shifts have been reported in embryonic stem cells (ESCs) versus 2C-like cells and mouse embryonic fibroblasts (MEFs) [5,6], and during pre-adipocyte-to-mature-adipocyte transitions[1].

To further clarify the effects of *Irx3* loss on SUMOylation at the chromatin, future studies could employ an acute degradation system (e.g., AID degron) instead of a stable *Irx3* KO. This approach would allow for a more precise examination of the early consequences of *Irx3*

depletion on SUMOylation levels on chromatin and help delineate how increased SUMOylation promotes the switch from osteogenesis to adipogenesis.

We have included this discussion in lines 936-971.

Along this line, they chose to highlight a gene from the Wnt pathway where IRX3 KO leads to a decreased SUMOylation, which again complicates the narrative.

We acknowledge the potential confusion surrounding this finding and recognize the need for a clearer explanation of our data interpretation. As detailed above, our results demonstrate that *Irx3* knockout (KO) directly affects SUMOylation levels while also exerting indirect effects that contribute to cellular identity conversion into osteoblasts.

SUMOylation is primarily known as a transcriptional repressor [2]. Therefore, the osteoblast-specific SUMOylation profile observed in *Irx3* KO cells, characterized by reduced SUMOylation at Wnt gene loci, may suggest active transcription of these genes. These lower SUMOylation levels likely indicate that the Wnt pathway is actively maintaining the newly acquired osteoblastic identity of *Irx3* KO cells.

What happens with *Rspo2* expression upon ML792 treatment?

Rspo2 is downregulated with ML-792 treatment (-1.6 fold on day 1 and -2.8 fold on day 9). Conversely, it is upregulated with IRX3-KO (1.7-fold on day 1 and 9-fold on day 9). These data support increased Wnt signaling (*Rspo2* is a Wnt activator) in IRX3-KO cells and reduced Wnt signaling with ML-792 treatment, which fits the narrative. We have added these data to a new panel H in Figure 6, as well as updated the text on lines 466, 473-479 and 511-515.

To strengthen the link between SUMOylation on chromatin and IRX3-controlled gene expression, it would be interesting to determine if the genes being down-regulated in IRX3KO and up-regulated in ML792 (Figure 7) show increased SUMOylation on their regulatory regions upon IRX3KO.

The reviewer highlights a relevant question. In line with the limited changes of SUMO occupancy, most of the inversely regulated genes did not show altered SUMO occupancy (taking promoters and proximal, but not distal (e.g MB ranges), enhancers into account). However, we did observe a significant increase in SUMO occupancy at the promoter of *Fdx1*, a gene regulating sterol synthesis and protein lipoylation (including enzymes in the TCA

cycle) [7]. This gene is part of the GO term “electron transport chain” (quadrant v), whose genes were downregulated with IRX3-KO and upregulated by ML-792. We have included this data in a new Figure 7F, as well as in lines 600-604 and 625-627.

Line 1018: « we observed significant enrichment of several cancer types ». Do you mean several cancer-related genes?

Yes, we mean enrichment of genes related to (by GO annotation) several types of cancer. We have clarified this sentence in line 832 in the revised manuscript, thanks for the comment.

Reviewer #2 (Remarks to the Author):

In this revised manuscript, the authors have made substantial revisions to improve the manuscript, and their efforts in addressing previous concerns are appreciated.

This study presents interesting findings on the role of IRX3 in regulating adipogenic potential in adipocyte precursor cells. However, a critical limitation remains unresolved—the cell type specificity of the findings and their physiological relevance. The central claim that IRX3 modulates a SUMOylation-dependent switch between adipogenesis and osteogenesis is largely supported by data from ME3 cells, and there is no in vivo validation that IRX3 influences mesenchymal differentiation under physiological conditions. ME3 cells represent a specific cell type with unique properties, and it remains unclear whether the observed effects translate to other widely used adipogenic and mesenchymal lineage models, such as 3T3-L1 cells, primary preadipocytes, or bone marrow-derived mesenchymal stem cells. Although the authors acknowledged this limitation in the discussion, stating that “while IRX3 represses osteogenesis and promotes adipogenesis in adipose tissue, it appears to have an opposite effect in bones”, it still remains uncertain whether the findings from ME3 cells can be extrapolated to broader physiological contexts.

This reviewer is now more enthusiastic about this study. However, given these limitations, the title of the manuscript should be revised to better reflect the scope of the study. The current title implies a broad conclusion that IRX3 regulates mesenchymal lineage commitment, but as all functional evidence is derived from ME3 cells and previous studies have reported opposing Irx3 functions in osteogenic tissues, the title should explicitly specify

that findings are based on the adipocyte precursor cells. Suggested revisions for the title include:

- IRX3 controls a SUMOylation-dependent fate determination in adipocyte precursor cells
- IRX3 regulates a SUMOylation-dependent differentiation switch in adipocyte precursor cells
- IRX3 modulates SUMOylation-dependent adipogenic differentiation in ME3 cells

We acknowledge this limitation and have modified the title of the manuscript to “IRX3 controls a SUMOylation-dependent differentiation switch in adipocyte precursor cells” to emphasize that the scope of this study focused on adipose-derived cells. Additionally, we already highlighted the fact that we used SVF-derived primary cells and ME3 cells, and not bona fide mesenchymal stem cells, in the “limitation” paragraph of the discussion (lines 976-977).

Reviewer #3 (Remarks to the Author):

All concerns were satisfactorily addressed.

Reviewer #4 (Remarks to the Author):

The authors have sufficiently dealt with my criticism. I do not have further concerns.

References:

- [1] Zhao, X., Hendriks, I.A., Le Gras, S., Ye, T., Ramos-Alonso, L., Aurí, A., et al., 2022. Waves of sumoylation support transcription dynamics during adipocyte differentiation. *Nucleic Acids Research* 50(3): 1351–69.
- [2] Chymkowitch, P., Nguéa P, A., Enserink, J.M., 2015. SUMO-regulated transcription: challenging the dogma. *BioEssays : News and Reviews in Molecular, Cellular and Developmental Biology* 37(10): 1095–105.
- [3] Drisaldi, B., Colnaghi, L., Fioriti, L., Rao, N., Myers, C., Snyder, A.M., et al., 2015. SUMOylation Is an Inhibitory Constraint that Regulates the Prion-like Aggregation and Activity of CPEB3. *Cell Reports* 11(11): 1694–702.
- [4] Baptista, T., Grünberg, S., Minoungou, N., Koster, M.J.E., Timmers, H.T.M., Hahn, S., et al., 2017. SAGA Is a General Cofactor for RNA Polymerase II Transcription. *Molecular Cell* 68(1): 130-143.e5.
- [5] Theurillat, I., Hendriks, I.A., Cossec, J.C., Andrieux, A., Nielsen, M.L., Dejean, A., 2020. Extensive SUMO Modification of Repressive Chromatin Factors Distinguishes Pluripotent from Somatic Cells. *Cell Reports* 32(11).
- [6] Cossec, J.C., Theurillat, I., Chica, C., Búa Agúin, S., Gaume, X., Andrieux, A., et al., 2018. SUMO Safeguards Somatic and Pluripotent Cell Identities by Enforcing Distinct Chromatin States. *Cell Stem Cell* 23(5): 742-757.e8.
- [7] Mohibi, S., Zhang, Y., Perng, V., Chen, M., Zhang, J., Chen, X., 2024. Ferredoxin 1 is essential for embryonic development and lipid homeostasis. *ELife* 13.

REVIEWERS' COMMENTS

Reviewer #1 (Remarks to the Author):

The authors have successfully addressed my comments.

We thank the reviewer for his/her contributions to improve the manuscript.

Reviewer #2 (Remarks to the Author):

All concerns were satisfactorily addressed.

We thank the reviewer for his/her contributions to improve the manuscript.